# Breaking the Order Barrier: Off-Policy Evaluation for Confounded POMDPs

**Qi Kuang**
School of Statistics and Data Science
Jiangxi University of Finance and Economics

**Jiayi Wang**
Department of Mathematical Sciences
University of Texas at Dallas

**Fan Zhou** *
School of Statistics and Data Science
MoE Key Laboratory of Interdisciplinary Research of Computation and Economics
Shanghai University of Finance and Economics

**Zhengling Qi** *
School of Business
George Washington University

## Abstract

We consider off-policy evaluation (OPE) in Partially Observable Markov Decision Processes (POMDPs) with unobserved confounding. Recent advances have introduced bridge-function to circumvent unmeasured confounding and develop estimators for the policy value, yet the statistical error bounds of them related to the length of horizon $T$ and the size of the state-action space $|\mathcal{O}||\mathcal{A}|$ remain largely unexplored. In this paper, we systematically investigate the finite-sample error bounds of OPE estimators in finite-horizon tabular confounded POMDPs. Specifically, we show that under certain rank conditions, the estimation error for policy value can achieve a rate of $\mathcal{O}(T^{1.5}/\sqrt{n})$, excluding the cardinality of the observation space $|\mathcal{O}|$ and the action space $|\mathcal{A}|$. With an additional mild condition on the concentrability coefficients in confounded POMDPs, the rate of estimation error can be improved to $\mathcal{O}(T/\sqrt{n})$. We also show that for a *fully history-dependent policy*, the estimation error scales as $\mathcal{O}\big(T/\sqrt{n}(|\mathcal{O}||\mathcal{A}|)^{\frac{T}{2}}\big)$, highlighting the exponential error dependence introduced by history-based proxies to infer hidden states. Furthermore, when the target policy is *memoryless policy*, the error bound improves to $\mathcal{O}\big(T/\sqrt{n}\sqrt{|\mathcal{O}||\mathcal{A}|}\big)$, which matches the optimal rate known for tabular MDPs. To the best of our knowledge, this is the first work to provide a comprehensive finite-sample analysis of OPE in confounded POMDPs.

## 1 Introduction

Partially Observable Markov Decision Processes (POMDPs) (Monahan, 1982) have become a practical framework for modeling decision-making under uncertainty across a wide range of applications (Albright, 1979; Monahan, 1982; Singh et al., 1994; Cassandra, 1998; Young et al., 2013; Bravo et al., 2019). In POMDPs, the agent must make decisions based on partial observations rather than full access to the underlying system state. Such partial observability often arises in real-world settings, making standard Markov Decision Processes (MDPs) inadequate for modeling the underlying data-generating processes. For example, in healthcare, clinical decision-making is frequently based on

---

*Corresponding authors: `zhoufan@mail.shufe.edu.cn`; `qizhengling@email.gwu.edu`

39th Conference on Neural Information Processing Systems (NeurIPS 2025).

partial information, while important latent factors like disease progression or genetic predispositions remain hidden or difficult to quantify.

A recent line of research has focused on off-policy evaluation (OPE) within the framework of confounded POMDPs (Tennenholtz et al., 2020; Nair and Jiang, 2021; Shi et al., 2022; Miao et al., 2022; Bennett and Kallus, 2024). Since confounded POMDPs inherently violate the Markov assumption and meanwhile encounter unmeasured confounders, existing approaches draw inspiration from proximal causal inference (Miao et al., 2018; Kallus et al., 2021; Cui et al., 2024), leveraging partial observations as a proxy to infer the hidden state, which enables the identification of the policy value. However, treating histories as proxy states poses fundamental hardness for POMDPs, as these histories can entail an exponentially large number of possibilities, thereby demanding a potentially exponential number of samples for accurate evaluation (Liu and Jin, 2022).

To avoid an explicit exponential dependence of the error on the cardinality of the observation space, recent work (Tennenholtz et al., 2020; Nair and Jiang, 2021) uses importance sampling (IS) but introduces an exponential-in-horizon quantity due to using the cumulative importance weights. To address the history-induced "curse of horizon"(Liu et al., 2018), recent work (Shi et al., 2022; Uehara et al., 2023; Zhang and Jiang, 2025a) employ function approximation under realizability assumptions and specific coverage assumptions. However, in tabular confounded POMDPs, despite the inevitable dependence on the observation and action sizes, the explicit dependence on the cardinality $|\mathcal{O}||\mathcal{A}|$ for history-dependent policies remains unspecified.

On the other hand, excluding the influence of cardinality, the optimal error rate achievable by OPE estimators in confounded POMDPs remains unclear. Recent work (Zhang and Jiang, 2025a,b) demonstrate an error rate of $\mathcal{O}(T^2/\sqrt{n})$, yet these pertain to *unconfounded* POMDPs, where the behavior policy is solely dependent on observed variables. In contrast, it has been shown that OPE estimators can achieve an error rate of $\mathcal{O}(T/\sqrt{n})$ in MDPs, which is known to be minimax optimal in both sample size and horizon length. For instance, Yin and Wang (2020) shows that marginal importance sampling (MIS) estimator can attain such error bounds for tabular setting, while Wang et al. (2024) further proves that nonparametric Fitted-Q Evaluation (FQE) estimators can achieve similar sharp dependence under the realizability assumption on the ratio functions. Given these insights, one naturally wonders if the successful analysis can be leveraged to better understand the horizon dependence for the convergence rate of the OPE estimator in the presence of unobserved confounding.

Specifically, drawing from the aforementioned literature, we seek to address three questions:

**Q1:** *How does the history-dependent policy explicitly influence the dependency of error bounds in confounded POMDPs?* **Q2:** *How does the error bound depend on the horizon $T$? Can the sharp linear rate, known for fully observable settings, be achieved in confounded POMDPs?* **Q3:** *What is the role of the concentrability coefficients (the ratio functions defined in Assumption 3) in improving the convergence rate of OPE estimators in confounded POMDPs?*

In this paper, we investigate the problem of off-policy evaluation under the confounded POMDPs, with a focus on addressing the three key questions. Compared to existing work on OPE in confounded POMDPs, our contributions are as follows: Firstly, we establish a two-step model-based approach to estimate the policy value, which relies on certain matrix invertibility conditions. We demonstrate the estimation error for fully history-dependent policy scale as $\mathcal{O}(T^{1.5}/\sqrt{n}(|\mathcal{O}||\mathcal{A}|)^{\frac{T}{2}})$, while the higher-order terms of error bound exhibit a stronger dependence on the size of observation-action space, scaling as $(|\mathcal{O}||\mathcal{A}|)^{\frac{3T}{2}}$. Secondly, excluding the influence of cardinality, as $T$ grows, the first-order term of the error bound demonstrates a $T^{1.5}$ dependence, while the higher-order terms show a stronger $T^3$ dependence. Thirdly, by assuming the boundedness of certain concentrability coefficients in confounded POMDPs, without additionally estimating these probability ratio functions, we show that the first-order term of the error bound can be improved to $\mathcal{O}(T/\sqrt{n}(|\mathcal{O}||\mathcal{A}|)^{\frac{T}{2}})$. Lastly, when the target policy is reduced to a memoryless policy, the error bound improves to $\mathcal{O}(T/\sqrt{n}\sqrt{|\mathcal{O}||\mathcal{A}|})$, matching the sharpest rate of convergence known in the tabular MDPs setting.

## 2 Related Work

**OPE in confounded POMDPs.** A growing line of work has studied OPE in confounded POMDPs by leveraging proxy variables to identify policy value in the presence of unobserved confounders

(Zhang and Bareinboim, 2016; Shi et al., 2022; Lu et al., 2023; Hong et al., 2024b,a). While these methods have primarily focused on settings involving function approximation, theoretical analysis in tabular settings remains relatively limited. Besides, a line of research investigates the *unconfounded* setting (Uehara et al., 2023; Hu and Wager, 2023; Zhang and Jiang, 2025a,b). Among these, Zhang and Jiang (2025b) demonstrates that history-dependent policies can achieve polynomial sample complexity for OPE under certain coverage assumptions and further highlights the necessity of model-based approaches in this context. However, the unconfounded setting is less challenging than ours, as it fails to capture the complexities introduced by unobserved confounding. Crucially, none of the above works investigate whether optimal convergence rates can be achieved in confounded POMDPs.

**OPE in MDPs.** OPE in MDPs has been extensively studied, including importance sampling (IS) approaches (Precup, 2000) and their doubly robust variants (Dudík et al., 2011; Jiang and Li, 2016; Thomas and Brunskill, 2016), as well as marginalized importance sampling (MIS) methods (Xie et al., 2019; Kallus and Uehara, 2020; Yin and Wang, 2020). While IS-based estimators are broadly applicable, they suffer from high variance, leading to exponential dependence on the horizon length. Additionally, MIS-based estimators rely on the Markov property, which limits their direct application to confounded POMDPs. Yin and Wang (2020) demonstrate that MIS methods can achieve a rate of $\mathcal{O}(T/\sqrt{n})$. However, there has been no work investigating the potential for improving the dependence on horizon length for OPE in confounded POMDPs.

## 3 Preliminaries

**POMDP Setup.** We consider a finite-horizon episodic POMDP denoted by $\mathcal{M} := (\mathcal{S}, \mathcal{O}, \mathcal{A}, T, \nu_1, \{P_t\}_{t=1}^T, \{\mathcal{T}_t\}_{t=1}^T, \{r_t\}_{t=1}^T)$, where $\mathcal{S}$, $\mathcal{O}$ and $\mathcal{A}$ denote the state space, the observation space, and the action space respectively. In this paper, all $\mathcal{S}$, $\mathcal{O}$ and $\mathcal{A}$ are finite. The integer $T$ is the total length of the horizon. We use $\nu_1 \in \Delta(\mathcal{S})$ to denote the distribution of the initial state, where $\Delta(\Omega)$ is a class of all probability distributions over the space $\Omega$. Denote $\{P_t\}_{t=1}^T$ to be the collection of state transition kernels over $\mathcal{S} \times \mathcal{A}$ to $\mathcal{S}$, and $\{\mathcal{T}_t\}_{t=1}^T$ to be the collection of observation emission kernels over $\mathcal{S}$ to $\mathcal{O}$. We use $\{r_t\}_{t=1}^T$ denote the collection of reward functions, i.e., $r_t : \mathcal{S} \times \mathcal{A} \to [-1, 1]$ at each time step $t$. Finally, we let $O_0$ denote the prior observation, which provides prior information about the initial state $S_1$.

In a standard POMDP, at each time step $t$, given the current (hidden) state $S_t$, an observation $O_t \sim \mathcal{T}_t(\cdot \mid S_t)$ is observed. The agent then selects an action $A_t$ according to a certain policy, receives a reward $R_t$ with $\mathbb{E}[R_t \mid S_t = s_t, A_t = a_t] = r_t(s_t, a_t)$ for every $(s_t, a_t)$, and the environment transitions to the next state $S_{t+1}$ according to $P_t(\cdot \mid S_t, A_t)$.

**Off-policy Evaluation (OPE) under Unmeasured Confounding.** This paper aims to estimate the value of a potentially history-dependent policy for POMDPs using offline data. Define the observed history by $H_t := (O_1, A_1, ..., O_t, A_t) \in \mathcal{H}_t$, where $\mathcal{H}_t := \prod_{j=1}^t (\mathcal{O} \times \mathcal{A})$ is the space of observable history up to time $t$. The *target policy* to be evaluated is denoted by $\{\pi_t\}_{t=1}^T$, where $\pi_t : \mathcal{O} \times \mathcal{H}_{t-1} \to \Delta(\mathcal{A})$ is *history-dependent*, illustrated by the green arrows in Figure 1. Given the target policy $\pi := \{\pi_t\}_{t=1}^T$, the policy value is defined as

$$\mathcal{V}(\pi) := \mathbb{E}_{S_1 \sim \nu_1}\Big[\mathbb{E}^\pi\Big[\sum_{t=1}^T R_t \mid S_1\Big]\Big],$$

where $\mathbb{E}^\pi$ is taken with respect to the distribution induced by the policy $\pi$.

In the offline setting, an agent cannot interact with the environment but only has access to a pre-collected dataset generated by some *behavior policy* $\{\pi_t^b\}_{t=1}^T$. We assume that the behavior policy depends on the unobserved state $S_t$, i.e., $\pi_t^b : \mathcal{S} \to \Delta(\mathcal{A})$ for each $t$. Unlike previous work (Uehara et al., 2023; Zhang and Jiang, 2025a,b), which restricts the behavior policy to be *memoryless* and dependent only on the current observation (i.e., $a_t \sim \pi_t^b(\cdot|o_t)$), our approach allows the behavior policy to depend on the latent state, making it more aligned with real-world scenarios and thus introducing additional complexity in the analysis. We use $\mathcal{P}$ to denote the offline data distribution under the behavior policy and summarize the data as $\mathcal{D} := \{o_0^i, (o_t^i, a_t^i, r_t^i)_{t=1}^T\}_{i=1}^n$, which consists of $n$ i.i.d. samples drawn from $\mathcal{P}$. The full data generating process in so-called confounded POMDPs is depicted in Figure 1.

**Notations.** Throughout this paper, we assume that $\mathbb{E}$ is taken with respect to the offline distribution. We use uppercase letters such as $(S_t, O_t, A_t, R_t, H_t)$ to denote random variables and lowercase letters such as $(s_t, o_t, a_t, r_t, h_t)$ to denote their realizations, unless stated otherwise. We use the notation $X \perp\!\!\!\perp Y \mid Z$ when $X$ and $Y$ are conditionally independent given $Z$ under the offline distribution. For random variables $X, Y$ and $Z$ taking values on $\{x_1, \ldots, x_m\}$, $\{y_1, \ldots, y_n\}$, and $\{z_1, \ldots, z_q\}$, respectively, $\mathbb{P}(X)$ denotes a $n$ length vector with entries $\mathbb{P}_i(X) := p(X = x_i)$ and $\mathbb{P}(X \mid Y)$ denotes a $n \times m$ matrix with entries $\mathbb{P}_{i,j}(X \mid Y) := p(X = x_j \mid Y = y_i)$. Similarly, $\mathbb{P}(X = x \mid Y)$ denotes the $n$ length vector with entries $\mathbb{P}_i(X \mid Y) := p(X = x \mid Y = y_i)$, and $\mathbb{P}(X, Z = z \mid Y)$ denotes a $n \times m$ matrix with entries $\mathbb{P}_{i,j}(X, Z = z \mid Y) := p(X = x_j, Z = z \mid Y = y_i)$. For brevity, we sometimes abbreviate these as $\mathbb{P}(x \mid Y)$ and $\mathbb{P}(X, z \mid Y)$. For any two sequences $\{a_n\}_{n=1}^\infty$, $\{b_n\}_{n=1}^\infty$, $a_n \lesssim b_n$ denotes $a_n \leq C b_n$ for some $N, C > 0$ and every $n > N$. For a matrix $A$, we denote its Moore-Penrose inverse by $A^\dagger$ and its smallest singular value by $\sigma_{min}(A)$.

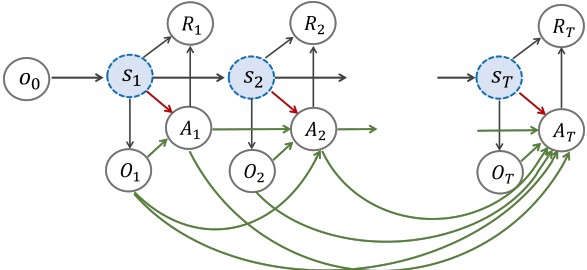

Figure 1: The directed acyclic graph of the data-generating process in confounded POMDPs, where states $S_t$ are not observed. Red arrows indicate the generation of actions via the behavior policy, while green arrows indicate the generation through a target history-dependent policy. Under the offline distribution $\mathcal{P}$, we have the conditional independency $O_0 \perp\!\!\!\perp (O_t, O_{t+1}, R_t) \mid S_t, A_t, H_{t-1}$ for any $t \in [T]$.

## 4 Methods

In this section, we introduce the proposed off-policy evaluation method for confounded POMDPs. Before considering how to estimate $\mathcal{V}(\pi)$, we first consider the problem of identification.

### 4.1 Identification

In general, identification is not possible for OPE in the presence of unobserved confounders. The fundamental challenge arises from two sources: (i) **partial observability**, which breaks the conditional independence (Markov property) essential for classical OPE estimators; and (ii) **unmeasured confounding**, which renders direct estimation of the policy value $\mathcal{V}(\pi)$ intractable. Crucially, failure to account for such confounding leads to biased estimates of the policy value (Shi et al., 2022), as the hidden state simultaneously influences both the action and future rewards and transitions.

To address these issues, a natural strategy is to use the observed history to infer information about the hidden state (Shi et al., 2022; Hong et al., 2024a,b). This approach is motivated by the insight that the entire history contains rich information that can help reconstruct a proxy for the unobserved state. To enable the observed history for identifying the policy value, we impose rank conditions.

**Assumption 1** (Invertibility). *For each $t \in [T]$ and $a_t \in \mathcal{A}$, assume $\mathbb{P}(O_t \mid A_t = a_t, S_t) \in \mathbb{R}^{|\mathcal{S}| \times |\mathcal{O}|}$ has full row rank and $\mathbb{P}(O_t \mid A_t = a_t, O_0, H_{t-1}) \in \mathbb{R}^{|\mathcal{O}||\mathcal{H}_{t-1}| \times |\mathcal{O}|}$ has full column rank.*

A necessary condition for the rank condition is that $|\mathcal{O}| > |\mathcal{S}|$. Invertibility of $\mathbb{P}(O_t \mid A_t = a_t, S_t)$ guaranties that sufficient information about the hidden states is encoded in the observations across time steps. The invertibility of $\mathbb{P}(O_t \mid A_t = a_t, O_0, H_{t-1})$ further ensures the recovery of information about the hidden states $S_t$ from the observable history $\{O_0, H_{t-1}\}$. Notably, leveraging the entire history as a proxy relaxes the standard rank condition that requires $\mathbb{P}(O_t \mid A_t = a_t, O_{t-1})$, using one-step observation as a proxy, to be invertible (Tennenholtz et al., 2020; Shi et al., 2022). This is particularly beneficial in real-world applications where the observation space is often larger than the hidden state space.

**Theorem 1.** *Under Assumptions 1, $\mathcal{V}(\pi)$ is identified as*

$$\mathcal{V}(\pi) = \sum_{t=1}^{T} \sum_{r_t} \sum_{h_t} r_t \Big( \prod_{i=1}^{t} \pi_i(a_i \mid o_i, h_{i-1}) \Big) \mathbb{P}(O_0) \mathbb{P}(O_1 | O_0) \mathbb{P}^{\dagger}(O_1 \mid A_1 = a_1, O_0)$$

$$\Big( \prod_{i=1}^{t-1} \mathbb{P}(O_{i+1}, O_i = o_i \mid A_i = a_i, O_0, H_{i-1}) \mathbb{P}^{\dagger}(O_{i+1} \mid A_{i+1} = a_{i+1}, O_0, H_i) \Big)$$

$$\mathbb{P}(R_t = r_t, O_t = o_t \mid A_t = a_t, O_0, H_{t-1}).$$

Theorem 1 states that the policy value can be expressed entirely in terms of observable variables. This expression is derived by decomposing the marginal distribution $p^{\pi}(r_t)$ using the transition dynamics, reward, and target policy, based on the identity $\mathcal{V}(\pi) = \mathbb{E}^{\pi}[\sum_{t=1}^{T} R_t] = \sum_{t=1}^{T} \sum_{r_t} r_t p^{\pi}(r_t)$. Similar results were initially introduced by Tennenholtz et al. (2020). Specifically, the invertibility of the action-conditioned probability matrices $\mathbb{P}(O_{t+1} \mid A_t = a_t, O_0, H_t)$ allows us to algebraically reconstruct the distribution of observations and rewards under the target policy using offline data collected by the behavior policy. Intuitively, conditioning on the action blocks the confounding path through the unobserved states, isolating the confounding effect of the action. Consequently, these action-conditioned probabilities, as proxy functions, can effectively correct the confounding influence, thus enabling us to identify the policy value. In addition, we extend the identification results beyond the tabular setting (see Theorem 5 in the Appendix), showing that the completeness condition (Assumption 5 in the Appendix), which is a generalization of the rank conditions in Assumption 1, is sufficient for the identification theorem to hold in more general settings.

## 4.2 Estimation via Value Functions

According to Theorem 1, we can directly perform OPE by estimating those conditional matrices relying solely on the offline data. However, the presence of multiple matrix inverses in the product is computationally expensive and may lead to instability in the estimation. To address this, we are motivated by the structure of the identification in Theorem 1 and propose to solve a Bellman-type equation to estimate the policy value as a more stable and computationally efficient approach.

**Bellman-like recursions in confounded POMDPs.** Under the invertibility assumptions stated in Assumption 1, the proxy $\{O_0, H_{t-1}\}$ is sufficient to construct a Bellman-like recursion based entirely on observable variables (see Appendix B.1 for further details). Specifically, we can formulate a system of linear integral equations (1), whose solution defines a sequence of proxy value functions $\{b_{V,t}^{\pi} : \mathcal{A} \times \mathcal{O} \times \mathcal{H}_{t-1} \to \mathbb{R}\}_{t=1}^{T}$.

$$\mathbb{E}\Big[b_{V,t}^{\pi}(A_t, O_t, H_{t-1}) \mid O_0, A_t, H_{t-1}\Big]$$
$$= \mathbb{E}\Big[R_t \pi_t(A_t \mid O_t, H_{t-1}) + \sum_{a' \in \mathcal{A}} b_{V,t+1}^{\pi}(a', O_{t+1}, H_t) \pi_t(A_t \mid O_t, H_{t-1}) \mid O_0, A_t, H_{t-1}\Big], \quad (1)$$

where $b_{V,T+1}^{\pi} \equiv 0$. Note that the invertibility of $\mathbb{P}(O_t \mid A_t = a_t, O_0, H_{t-1})$ ensures that the value functions $\{b_{V,t}^{\pi}\}_{t=1}^{T}$ are uniquely defined solutions to this system. In the context of confounded POMDPs, these proxy functions play an analogous role to the value functions in MDPs, enabling the estimation of policy value through these functions. This naturally suggests a two-step procedure for OPE in confounded POMDPs: (i) estimate the conditional probabilities, and (ii) compute the value function from (1) using the estimated probabilities.

To illustrate the model-based estimation in the tabular setting, we first define two key conditional probability matrices for any given $a_t \in \mathcal{A}$, $r_t$, and $o_{t+1} \in \mathcal{O}$,

$$\mathbf{P}_{a_t} := \mathbb{P}(O_t \mid O_0, A_t = a_t, H_{t-1}) \in \mathbb{R}^{|\mathcal{O}||\mathcal{H}_{t-1}| \times |\mathcal{O}|},$$

$$\mathbf{P}_{a_t, r_t, o_{t+1}} := \mathbb{P}(O_{t+1} = o_{t+1}, R_t = r_t, O_t \mid O_0, A_t = a_t, H_{t-1}) \in \mathbb{R}^{|\mathcal{O}||\mathcal{H}_{t-1}| \times |\mathcal{O}|}.$$

Then, we can rewrite the value recursion from equation (1) in matrix form as:

$$\mathbf{P}_{a_t} \mathbf{B}_t = \sum_{r_t, o_{t+1}} \mathbf{P}_{a_t, r_t, o_{t+1}} \Big[ r_t \cdot \pi_t(a_t | \mathbf{o}_t, h_{t-1}) + \sum_{a' \in \mathcal{A}} b_{V,t+1}^{\pi}(a', o_{t+1}, h_{t-1}, a_t, \mathbf{o}_t) \odot \pi_t \Big], \quad (2)$$

where $\mathbf{B}_t := b_{V,t}^\pi(a_t, \mathbf{o}_t, h_{t-1}) \in \mathbb{R}^{|\mathcal{O}|}$, $\mathbf{P}_{a_t}, \mathbf{P}_{a_t, r_t, o_{t+1}} \in \mathbb{R}^{|\mathcal{O}||\mathcal{H}_{t-1}| \times |\mathcal{O}|}$ are conditional probability matrices defined earlier, $\odot$ denotes the element-wise product, and $r_t \pi_t(a_t | \mathbf{o}_t, h_{t-1}) + \pi_t(a_t | \mathbf{o}_t, h_{t-1}) \odot \sum_{a'} b_{V,t+1}^\pi(a', o_{t+1}, h_{t-1}, a_t, \mathbf{o}_t) \in \mathbb{R}^{|\mathcal{O}|}$ is a $|\mathcal{O}|$ length vector. To ensure a unique solution of $\mathbf{B}_t$, the matrix $\mathbf{P}_{a_t}$ must be full column rank, i.e. $\text{rank}(\mathbf{P}_{a_t}) = |\mathcal{O}|$. This is a mild and typically reasonable assumption in practice, since $|\mathcal{O}| \ll |\mathcal{O}||\mathcal{H}_{t-1}|$ as $t$ increases. Then, the solution to the linear system (2) is given by

$$\mathbf{B}_t = \mathbf{P}_{a_t}^\dagger \sum_{r_t, o_{t+1}} \mathbf{P}_{a_t, r_t, o_{t+1}} \big[ r_t \cdot \pi_t(a_t | \mathbf{o}_t, h_{t-1}) + \sum_{a' \in \mathcal{A}} b_{V,t+1}^\pi(a', o_{t+1}, h_{t-1}, a_t, \mathbf{o}_t) \odot \pi_t \big],$$

where $\mathbf{P}_{a_t}^\dagger = \left( \mathbf{P}_{a_t}^\top \mathbf{P}_{a_t} \right)^{-1} \mathbf{P}_{a_t}^\top \in \mathbb{R}^{|\mathcal{O}| \times |\mathcal{O}||\mathcal{H}_{t-1}|}$ is the Moore–Penrose inverse of $\mathbf{P}_{a_t}$. Given $b_{V,T+1}^\pi \equiv 0$, the value functions $\{b_{V,t}^\pi\}_{t=1}^T$ can be solved iteratively, starting from $t = T$ and proceeding backward. Specifically, for any $a_t \in \mathcal{A}, o_t \in \mathcal{O}, h_{t-1} \in \mathcal{H}_t$, the update rule gives

$$b_{V,t}^\pi(a_t, o_t, h_{t-1}) = \pi_t(a_t | o_t, h_{t-1}) \psi_t^\top \mathbf{P}_{a_t}^\dagger \sum_{r_t, o'} \mathbf{P}_{a_t, r_t, o_{t+1}} \psi_t \big( r_t + \sum_{a' \in \mathcal{A}} b_{V,t+1}^\pi(a', o', h_t) \big), \quad (3)$$

where $\psi_t(o_t) \in \mathbb{R}^{|\mathcal{O}|}$ is a one-hot encoding vector for the observation $o_t$.

**Two-stage estimation.** To enable the estimation of value functions, we estimate the conditional probability matrices $\widehat{\mathbf{P}}_{a_t}, \widehat{\mathbf{P}}_{a_t, r_t, o_{t+1}}$ from the dataset $\mathcal{D}$ at time step $t$. Each matrix is constructed entry-wise as follows:

$$\widehat{p}(o_t \mid o_0, a_t, h_{t-1}) = \frac{\sum_{i=1}^n \mathbb{1}\big\{ (o_t^i, h_{t-1}^i, o_0^i, a_t^i) = (o_t, h_{t-1}, o_0, a_t) \big\}}{n_{o_0, h_{t-1}, a_t}},$$

$$\widehat{p}(o_{t+1}, o_t, r_t \mid o_0, a_t, h_{t-1}) = \frac{\sum_{i=1}^n \mathbb{1}\big\{ (o_{t+1}^i, o_t^i, r_t^i, o_0^i, h_{t-1}^i, a_t^i) = (o_{t+1}, o_t, r_t, o_0, h_{t-1}, a_t) \big\}}{n_{o_0, h_{t-1}, a_t}},$$

(4)

where $n_{o_0, h_{t-1}, a_t} = \sum_{i=1}^n \mathbb{1}\{(o_0^i, h_{t-1}^i, a_t^i) = (o_0, h_{t-1}, a_t)\}$ denotes the number of the triplet $(o_0, h_{t-1}, a_t)$ being visited among $n$ independent episodes.

Then, the sequence of estimated value functions $\{\widehat{b}_{V,t}\}_{t=1}^T$ can be solved iteratively by

$$\widehat{b}_{V,t}(a_t, o_t, h_{t-1}) = \pi_t(a_t | o_t, h_{t-1}) \psi_t^\top \widehat{\mathbf{P}}_{a_t}^\dagger \sum_{r_t, o'} \widehat{\mathbf{P}}_{a_t, r_t, o_{t+1}} \psi_t \big( r_t + \sum_{a' \in \mathcal{A}} \widehat{b}_{V,t+1}(a', o', h_t) \big), \quad (5)$$

where $\widehat{b}_{V,T+1} = 0$, $\widehat{\mathbf{P}}_{a_t}^\dagger = \left( \widehat{\mathbf{P}}_{a_t}^\top \widehat{\mathbf{P}}_{a_t} \right)^{-1} \widehat{\mathbf{P}}_{a_t}^\top \in \mathbb{R}^{|\mathcal{O}| \times |\mathcal{O}||\mathcal{H}_{t-1}|}$. By Lemma 12, these empirical estimators are unbiased, i.e. $\mathbb{E}[\widehat{p}(o_t \mid o_0, a_t, h_{t-1})] = p(o_t \mid o_0, a_t, h_{t-1})$, which implies that $\mathbb{E}[\widehat{\mathbf{P}}_{a_t}] = \mathbf{P}_{a_t}$. Consequently, the recursion in (5) ultimately yields asymptotically unbiased estimates of the true value functions. Moreover, the $\widehat{\mathbf{P}}_{a_t}$ must be invertible for the recursion to proceed, which implicitly requires that each tuple $(o_0, h_{t-1}, a_t)$ must be observed sufficiently many times. To ensure numerical stability and avoid division by zero, we assume each $(o_0, h_{t-1}, a_t)$ in the data is sufficiently sampled. If the triple $(o_0, h_{t-1}, a_t)$ is not collected in the offline data, we set $\widehat{p}(\cdot | o_0, a_t, h_{t-1}) = 0$.

**Policy value estimation.** After computing the value functions via (5) for $T$ iterations, we plug the estimated value function $\widehat{b}_{V,1}$ into $\mathcal{V}(\pi)$ to obtain the empirical estimator as

$$\widehat{\mathcal{V}}(\pi) = \frac{1}{n} \sum_{i=1}^n \Big[ \sum_{a \in \mathcal{A}} \widehat{b}_{V,1}(a, o_1^i) \Big].$$

Algorithm 1 summarizes the proposed algorithm for OPE in confounded POMDPs.

## 5 Theoretical Results

In this section, we study the theoretical properties of our method under certain technical assumptions. Our primary goal is to establish a finite-sample error upper bound for $\mathcal{V}(\pi) - \widehat{\mathcal{V}}(\pi)$. Specifically, this upper bound will depend on several factors, including the sample size $n$, the horizon length $T$, and the size of the observation space $|\mathcal{O}|$ and action space $|\mathcal{A}|$. To begin with, we impose the following key assumptions that are used in the theoretical analysis.

---

**Algorithm 1** Tabular Off-Policy Evaluation for Confounded POMDPs

---
**Input:** Dataset $\mathcal{D}$, the target policy $\{\pi_t\}_{t=1}^T$, and initialize $\widehat{b}_{V,T+1} = 0$.
**for** $t = T, \ldots, 1$ **do**
    **Estimation of conditional probability**: obtain $\widehat{\mathbf{P}}_{a_t}$ and $\widehat{\mathbf{P}}_{a_t,r_t,o_{t+1}}$ by (4)
    **Estimation of value functions:** obtain $\widehat{b}_{V,t}$ by (5)
**end for**
**Output:** obtain estimated policy value $\widehat{\mathcal{V}}(\pi)$ by $\widehat{b}_{V,1}$.

---

**Assumption 2.** *The following conditions hold.*

*(a) (Coverage) For each* $t \in [T]$, $\mathbb{E}\Big[ \prod_{t'=1}^{t} \Big( \frac{\pi_{t'}(A_{t'}|O_{t'}, H_{t'-1})}{\pi_{t'}^b(A_{t'}|S_{t'})} \Big)^2 \Big] \leq C_{\pi^b} < \infty$;

*(b) (Invertibility) For each* $t \in [T]$, $a_t \in \mathcal{A}$, $\operatorname{rank}(\mathbf{P}_{a_t}) = \operatorname{rank}(\widehat{\mathbf{P}}_{a_t}) = |\mathcal{O}|$, *and the smallest singular value* $\sigma_{min}(\mathbf{P}_{a_t})$ *satisfies* $\sigma_{min}(\mathbf{P}_{a_t}) \geq \frac{C_P^{-1}}{\sqrt{|\mathcal{O}||\mathcal{H}_{t-1}|}}$, *where* $0 < C_P < \infty$ *is a constant;*

*(c)(Sufficient visitation of observations) For each* $t \in [T]$, *the samples used to construct each entry of* $\widehat{\mathbf{P}}_{a_t,r_t,o_{t+1}}$ *satisfy* $n_{o_0,h_{t-1},a_t} \geq np_t^{\pi^b}(o_0, h_{t-1}, a_t)(1 - \theta_{t,ij})$, *where* $0 < \theta_{t,ij} < 1$ *with* $\sum_{i,j} \theta_{t,ij} = 1$, *and we denote* $\theta^* := \min_{t,i,j} \theta_{t,ij}$.

Assumption 2(a) imposes a bounded second moment condition on the cumulative importance ratio (concentrability coefficients), which is milder compared to directly bounding the importance weight $\pi_t(a_t|o_t, h_{t-1})/\pi_t^b(a_t|s_t)$, a common assumption in the OPE literature. The invertibility condition (b) requires $\mathbf{P}_{a_t}$ and $\widehat{\mathbf{P}}_{a_t}$ to be well-conditioned matrices, which is necessary for the uniqueness of the solution to the iteration equations (3) and (5). Furthermore, we require the lower bound of the smallest singular value of $\mathbf{P}_{a_t}$ to decay in proportion to $1/\sqrt{|\mathcal{O}||\mathcal{H}_{t-1}|}$. This is informed by random matrix theory, where for an $M \times N$ random matrix ($M$ fixed, $N \to \infty$), the smallest singular value decays at a rate of $\mathcal{O}(1/\sqrt{N})$ (Rudelson and Vershynin, 2009). Assumption 2(c) requires a sufficient number of samples for each triple $(o_0, h_{t-1}, a_t)$, ensuring consistent estimation of the conditional probability matrices. Specifically, this requires the sample size $n \geq \frac{\operatorname{polylog}(|\mathcal{O}|^T, |\mathcal{A}|^T, T)}{\min_{t,o_0,h_{t-1},a_t} p_t^{\pi^b}(o_0, h_{t-1}, a_t)}$, and further details can be found in Appendix C. Assumption 2(c) is introduced to simplify the proof and can be relaxed using a truncation argument, similar to the approach in Yin and Wang (2020).

We now present the main theorem that provides the upper bound for $|\mathcal{V}(\pi) - \widehat{\mathcal{V}}(\pi)|$.

**Theorem 2.** *Under Assumptions 1 and 2. Then, with probability at least* $1 - \delta$, *it holds that*

$$
\begin{aligned}
\big| \mathcal{V}(\pi) - \widehat{\mathcal{V}}(\pi) \big| \lesssim & \frac{T^{1.5}}{\sqrt{n}} (1 - \theta^*)^{-\frac{1}{2}} C_{\pi^b}^{\frac{1}{2}} C_P |\mathcal{O}|^{\frac{T}{2}} |\mathcal{A}|^{\frac{T}{2}} \\
& + \frac{T^{1.5}}{n} (1 - \theta^*)^{-1} C_{\pi^b}^{\frac{1}{2}} C_P^2 |\mathcal{O}|^{\frac{3T}{2}} |\mathcal{A}|^{\frac{3T}{2}} \sqrt{\log(T^2|\mathcal{O}|^T|\mathcal{A}|^T/\delta)} \\
& + \frac{T^3}{n^{\frac{3}{2}}} (1 - \theta^*)^{-\frac{3}{2}} C_{\pi^b}^{\frac{1}{2}} C_P^3 |\mathcal{O}|^{\frac{5T}{2}} |\mathcal{A}|^{\frac{5T}{2}} \log(T^2|\mathcal{O}|^T|\mathcal{A}|^T/\delta).
\end{aligned}
\tag{6}
$$

In Theorem 2, the first term in (6) scales as $\mathcal{O}(T^{1.5}/\sqrt{n})$, which we refer to as the first-order term. The remaining terms, which are higher-order terms, exhibit a stronger dependence on $T$ but converge more quickly due to the faster rates of the sample size $n$. These results respond to **Q2**. For clarity, we omit the detailed form of the higher-order terms here, and the full specifications are discussed in Appendix A. Compared to the order of $\mathcal{O}(T^2/\sqrt{n})$ obtained in both unconfounded POMDPs (Uehara et al., 2023; Zhang and Jiang, 2025a,b) and confounded POMDPs (Bennett and Kallus, 2024), we achieve a sharper dependence on the horizon by leveraging the fact that the variance of the first-order term can be decomposed into a sum of $T$ individual expectations of the conditional variance. Moreover, the first-order term in the upper bound (6) exhibits an exponential dependence of order $(|\mathcal{O}||\mathcal{A}|)^{\frac{T}{2}}$ on the observation and action space, while the higher-order term shows a strong dependence of order $(|\mathcal{O}||\mathcal{A}|)^{\frac{\beta T}{2}}$ with $\beta \geq 3$. These results respond to **Q1**. The increased complexity arises from the fully history-dependent policy, leading to challenges in evaluating policies as the effective policy domain expands over time with the growth of the history space $|\mathcal{H}_t|$. This complexity highlights the inherent statistical challenges of evaluating fully history-dependent policies in confounded POMDPs.

**Corollary 1.** *Under the conditions in Theorem 2, for the memoryless policy dependent on the current observation, i.e. $\pi : \mathcal{O} \to \Delta(\mathcal{A})$, with high probability, it holds that*

$$\left| \mathcal{V}(\pi) - \widehat{\mathcal{V}}(\pi) \right| = \mathcal{O}\left( \frac{T^{1.5}}{\sqrt{n}} (1 - \theta^*)^{-\frac{1}{2}} C_{\pi^b}^{\frac{1}{2}} C_P |\mathcal{O}|^{\frac{1}{2}} |\mathcal{A}|^{\frac{1}{2}} \right). \tag{7}$$

As shown in Corollary 1, when the target policy reduces to a memoryless policy, the exponential dependence on the observation and action spaces is no longer present. The upper bound in (7) instead exhibits a polynomial dependence on the horizon $T$, as well as on the cardinalities of the observation-action space. We now turn to a refined analysis of the upper bound in (6). Before proceeding, we assume the existence of the following sequence of ratio functions.

**Assumption 3.** *We assume the existence of real-valued functions $\{b_{W,t}^\pi : \mathcal{A} \times \mathcal{O} \times \mathcal{H}_{t-1} \to \mathbb{R}\}_{t=1}^T$ that satisfy the following conditional moment restrictions,*

$$\mathbb{E}\left[ b_{W,t}^\pi (A_t, H_{t-1}, O_0) \mid S_t, A_t, H_{t-1} \right] = \frac{\omega_t(S_t, H_{t-1})}{\pi_t^b(A_t \mid S_t)}, \tag{8}$$

*where $\omega_t(S_t, H_{t-1}) = p_t^\pi(S_t, H_{t-1}) / p_t^{\pi^b}(S_t, H_{t-1})$. We denote $\rho_t(o_t, s_t, a_t, h_{t-1}) := \mathbb{E}\left[ b_{W,t}^\pi(a_t, h_{t-1}, O_0) \pi_t(a_t \mid o_t, h_{t-1}) | o_t, s_t, a_t, h_{t-1} \right]$. We further assume there exists a constant $0 < C_W < \infty$ such that*

$$\sup_{t, o_t, s_t, a_t, h_{t-1}} \rho_t(o_t, s_t, a_t, h_{t-1}) \leq C_W. \tag{9}$$

These types of weight functions $\{b_{W,t}^\pi\}_{t=1}^T$ are also utilized in OPE for confounded POMDPs (Shi et al., 2022; Bennett and Kallus, 2024), which allows for the adjustment of the observed data distribution to account for the influence of unobserved confounders. The ratio $\rho_t$ plays the role of a concentrability coefficient in confounded POMDPs. Specifically, the following identity holds:

$$\rho_t(o_t, s_t, a_t, h_{t-1}) = \frac{p_t^\pi(o_t, s_t, a_t, h_{t-1})}{p_t^{\pi^b}(o_t, s_t, a_t, h_{t-1})} = \frac{p_t^\pi(s_t, h_{t-1}) \pi_t(a_t \mid o_t, h_{t-1}) p(o_t \mid s_t)}{p_t^{\pi^b}(s_t, h_{t-1}) \pi_t^b(a_t \mid s_t) p(o_t \mid s_t)}.$$

In the MDPs, $\rho_t$ reduces to the ratio of the state-action marginal density, which quantifies the mismatch between the marginal state-action distributions of the target and behavior policies. Analogous to the fully observable case, we impose the bounded ratio in (9), which ensures that the offline data distribution $\mathcal{P}$ can calibrate the distribution induced by the target policy $\pi$.

**Lemma 1.** *For the target policy $\pi : \mathcal{O} \times \mathcal{H}_{t-1} \to \Delta(\mathcal{A})$, we have*

$$\sum_{t=1}^T \mathbb{E}^\pi \left[ Var \left[ R_t + \sum_{a \in \mathcal{A}} b_{V,t+1}^\pi(a, O_{t+1}, H_t) \,\Big|\, O_t, S_t, A_t, H_{t-1} \right] \right] \leq Var^\pi \left[ \sum_{t=1}^T R_t \right],$$

*where $Var^\pi$ is taken with respect to the distribution induced by $\pi$.*

Lemma 1 establishes a variance decomposition bound that is analogous to Lemma 3.4 in Yin and Wang (2020), which focuses on the OPE in MDPs. Notably, it shows that the sum of the conditional variances across time steps is upper bounded by the variance of the cumulative reward, which is on the order of $T^2$, since $Var^\pi[\sum_{t=1}^T R_t] \lesssim T^2$. This result can be derived by iteratively applying the law of total variance. The complete derivation is provided in Appendix C.1.2.

**Theorem 3.** *Under Assumptions 1, 2, and 3, we have the following results:*

$(a)$ *for history-dependent policy, i.e. $\pi_t : \mathcal{O} \times \mathcal{H}_{t-1} \to \Delta(\mathcal{A})$, with high probability it holds that*

$$\left| \mathcal{V}(\pi) - \widehat{\mathcal{V}}(\pi) \right| = \mathcal{O}\left( \frac{T}{\sqrt{n}} (1 - \theta^*)^{-\frac{1}{2}} C_W^{\frac{1}{2}} C_P |\mathcal{O}|^{\frac{T}{2}} |\mathcal{A}|^{\frac{T}{2}} \right); \tag{10}$$

$(b)$ *for memoryless policy, i.e. $\pi_t : \mathcal{O} \to \Delta(\mathcal{A})$, with high probability it holds that*

$$\left| \mathcal{V}(\pi) - \widehat{\mathcal{V}}(\pi) \right| = \mathcal{O}\left( \frac{T}{\sqrt{n}} (1 - \theta^*)^{-\frac{1}{2}} C_W^{\frac{1}{2}} C_P |\mathcal{O}|^{\frac{1}{2}} |\mathcal{A}|^{\frac{1}{2}} \right). \tag{11}$$

Based on Lemma 1, we derive Theorem 3. We omit the higher-order terms here, as they remain identical to those in Theorem 2. Notably, the constant of the coverage condition $C_{\pi^b}$ is replaced by the constant of the bounded ratio $C_W$. Compared to the results in Theorem 2, the upper bound in (10) demonstrates an improvement in the dependence on the horizon length $T$ in the first-order term, reducing it from $T^{1.5}$ to $T$, which is the sharpest known dependence of the horizon for tabular POMDPs. The bounded ratio function in Assumption 3 here provides an enhanced method for deriving the bound on the first-order term, thereby achieving faster convergence. These results answer the question **Q3**. In the memoryless case, the upper bound in (11) matches the optimal order of $\mathcal{O}(T/\sqrt{n}\sqrt{|\mathcal{O}||\mathcal{A}|})$ established for tabular MDPs (Yin and Wang, 2020). This finding highlights that a linear sample complexity in $T$ is sufficient for evaluating the memoryless policy in confounded POMDPs under these conditions. In addition, it is crucial to emphasize that our approach does not require the estimation of weight functions $\{b_{W,t}^{\pi}\}_{t=1}^{T}$, focusing solely on the estimation of value functions $\{b_{V,t}^{\pi}\}_{t=1}^{T}$. This distinguishes our approach from prior work such as Shi et al. (2022), which necessitates the estimation of both components.

## 6 Simulation Results

We conduct a simulation study to examine the behavior of the error $|\mathcal{V}(\pi) - \widehat{\mathcal{V}}(\pi)|$ with respect to sample size $n$ and horizon $T$. The primary objective is to provide empirical validation for our theoretical results. To this end, we use a relatively simple simulation setup that ensures clarity in demonstration. Specifically, we evaluate our approach in a simulated POMDP environment characterized by a one-dimensional discrete state/observation space, a discrete reward space, and binary actions. Concretely, we set $\mathcal{A} = \{0, 1\}, \mathcal{S} = \mathcal{O} = \{0, 1, 2\}$ for all $t$. The initial observation is given by $O_0 \sim \text{Unif}(\{0, 1, 2\})$ and $S_t \sim \text{Unif}(\{0, 1, 2\})$, and the transition dynamic is given by $O_t \sim P_t(\cdot|S_t)$, where $P_t(O_t|S_t) = \mathbb{1}\{O_t = S_t\}(1 - 3\epsilon/2) + \epsilon/2$. The immediate reward is set to be $R_t = 2/\{1 + \exp(-2S_t A_t - 3) - 1\}$. We collected offline data using a time-homogeneous behavioral policy $\pi_t^b(1|S_t) = 1/\{1 + \exp(-0.6S_t + 1)\} = 1 - \pi_t^b(0|S_t)$. For experimental details, we set $\epsilon = 0.2$, initialize $\widehat{b}_{V,T+1} = 0$, estimate conditional probability matrices as described in (4), and iteratively compute the value function $\widehat{b}_{V,t}$ over $T$ steps according to (5).

We evaluate two target policies. (1) For the memoryless target policy $\pi_t(1|O_t) = 1/\{1 + \exp(-0.8O_t + 1)\} = 1 - \pi_t(0|O_t)$, the conditional probability $\mathbf{P}_{a_t}$ and $\mathbf{P}_{a_t,r_t,o_{t+1}}$ are conditioned on $O_0$. We evaluate its value using sample sizes $n = 200, 400, \ldots, 1000$, and horizon lengths $T = 20, 60, 100, 140$. The results, shown in Figure 2(a), reveal a nearly linear relationship between $|\mathcal{V}(\pi) - \widehat{\mathcal{V}}(\pi)|$ and $T$, which aligns with our theoretical results as shown in Theorem 3. (2) For the fully history-dependent target policy setting, to simplify computation, we fix the action space to a single action, $\mathcal{A} = \{1\}$. In this case, $\pi_t(1|O_t, H_{t-1}) = 1$ and the historical information $H_{t-1}$ is only used to estimate the conditional probability matrices. We evaluate the policy value using sample sizes $n = 1000, 4000, 7000, 10000$, and horizon lengths $T = 2, 4, 6$. Figure 2(b) presents the logarithm of $|\mathcal{V}(\pi) - \widehat{\mathcal{V}}(\pi)|$ versus $T$. For $n = 1000$ setting, we observe noticeable fluctuations due to the increased size of the conditional probability matrices as $T$ grows, which requires more samples to estimate each entry accurately. Nonetheless, across different $n$, we observe an approximately linear relationship between the logarithmic error and the horizon $T$. These experimental results are consistent with the theoretical findings presented in Theorem 3.

## 7 Conclusion

In this paper, we study the problem of OPE in confounded POMDPs, where both partial observability and unobserved confounding pose significant challenges to policy evaluation. To address these challenges, we propose a model-based framework that leverages observable history as a proxy for hidden states. Under suitable invertibility conditions, we establish identification results for history-dependent policies. Our theoretical analysis demonstrates that the proposed method achieves a convergence rate of $\mathcal{O}(\frac{T}{\sqrt{n}}|\mathcal{O}|^{\frac{T}{2}}|\mathcal{A}|^{\frac{T}{2}})$. An important direction for future research is to investigate the minimax-optimal rate for off-policy evaluation in confounded POMDPs and extend the framework to continuous state and action spaces.

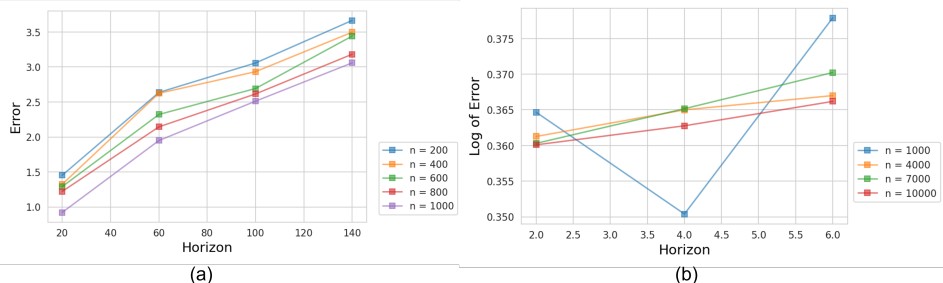

Figure 2: (a) Results for $|\mathcal{V}(\pi) - \widehat{\mathcal{V}}(\pi)|$ when the target policy is memoryless. (b) Results for $\log(|\mathcal{V}(\pi) - \widehat{\mathcal{V}}(\pi)|)$ when the target policy is fully history-dependent.

## Acknowledgment

The authors would like to thank the anonymous reviewers for their valuable comments and constructive suggestions, which have significantly improved the quality of this paper. Qi Kuang acknowledges funding from the Early-Career Young Scientists and Technologists Project of Jiangxi Province (No. 20252BEJ730126) and the National Natural Science Foundation of China (No. 12571286). Fan Zhou acknowledges support from the Shanghai Research Center for Data Science and Decision Technology.

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

The Appendix is organized as follows:

Section A presents the detailed statements for Theorem 2. Section B presents the detailed proof of Theorem 1. Section C presents the detailed proof of Theorem 2. Section D outlines the technical lemmas essential for the proofs. The code to implement the simulation is available at `https://github.com/kuangqi927/Confoundedpomdp`.

## A    Detailed Statements of Theorem 2

We begin by presenting the detailed statements for Theorem 2. We omit the full proof for Theorem 3, which differs from Theorem 2 only in the first-order term.

**Theorem 4.** *Under Assumptions 1 and 2, suppose the sample size $n$ is sufficiently large, then, with probability at least $1 - \delta$, it holds that*

$$
\begin{aligned}
\left| \mathcal{V}(\pi) - \widehat{\mathcal{V}}(\pi) \right| \lesssim & \frac{T^{1.5}}{\sqrt{n}} (1 - \theta^*)^{-\frac{1}{2}} C_{\pi^b}^{\frac{1}{2}} C_P |\mathcal{O}|^{\frac{T}{2}} |\mathcal{A}|^{\frac{T}{2}} \\
& + \frac{T^{1.5}}{n} (1 - \theta^*)^{-1} C_{\pi^b}^{\frac{1}{2}} C_P^2 |\mathcal{O}|^{\frac{3T}{2}} |\mathcal{A}|^{\frac{3T-1}{2}} \sqrt{\log(T^2 |\mathcal{O}|^T |\mathcal{A}|^T / \delta)} \\
& + \frac{T^3}{n} (1 - \theta^*)^{-1} C_{\pi^b}^{\frac{1}{2}} C_P^2 |\mathcal{O}|^{\frac{5T}{2}} |\mathcal{A}|^{\frac{5T}{2}} \sqrt{\log(T^2 |\mathcal{O}|^T |\mathcal{A}|^T / \delta)} \qquad (12) \\
& + \frac{T^3}{n^{\frac{3}{2}}} (1 - \theta^*)^{-\frac{3}{2}} C_{\pi^b}^{\frac{1}{2}} C_P^3 |\mathcal{O}|^{\frac{5T}{2}} |\mathcal{A}|^{\frac{5T}{2}} \log(T^2 |\mathcal{O}|^T |\mathcal{A}|^T / \delta) \\
& + \frac{T}{\sqrt{n}} \log(T^2 |\mathcal{A}| / \delta).
\end{aligned}
$$

The first four terms are related to sub-optimality terms. The proof of the first-order term is provided in Appendix C.1.1, and the proof of the higher-order terms is given in Appendix C.1.5. The proof of the last term related to the empirical process is presented in Appendix C.2.

## B    Proof of Theorem 1

For convenience, we omit the uppercase letters and abbreviate $\mathbb{P}(X, Z = z \mid Y)$ as $\mathbb{P}(X, z \mid Y)$.

*Proof.* By the definition of policy value, we have

$$
\begin{aligned}
\mathcal{V}(\pi) &= \sum_{t=1}^{T} \sum_{r_t} r_t p^{\pi}(r_t) \\
&= \sum_{t=1}^{T} \sum_{r_t} r_t \sum_{o_0, s_1, a_1, o_1, \ldots, s_t, a_t, o_t} p(r_t \mid o_0, s_1, a_1, o_1, \ldots, s_t, a_t, o_t) p(o_0, s_1, a_1, o_1, \ldots, s_t, a_t, o_t) \\
&= \sum_{t=1}^{T} \sum_{r_t} r_t \sum_{o_0, s_1, a_1, o_1, \ldots, s_t, a_t, o_t} p(r_t \mid s_t, a_t) p(o_0, s_1, a_1, o_1, \ldots, s_t, a_t, o_t).
\end{aligned}
$$

We now recursively decompose the joint probability $p(o_0, s_1, a_1, o_1, \ldots, s_t, a_t, o_t)$ as follows,

$$
\begin{aligned}
& p(o_0, s_1, a_1, o_1, \ldots, s_t, a_t, o_t) \\
&= p(a_t \mid o_0, s_1, a_1, o_1, \ldots, s_t, o_t) p(o_0, s_1, a_1, o_1, \ldots, s_t, o_t) \\
&= \pi_t(a_t \mid o_t, h_{t-1}) p(o_t \mid o_0, s_1, a_1, o_1, \ldots, s_t) p(o_0, s_1, a_1, o_1, \ldots, s_t) \\
&= \pi_t(a_t \mid o_t, h_{t-1}) p(o_t \mid s_t) p(o_0, s_1, a_1, o_1, \ldots, s_t) \\
&= \pi_t(a_t \mid o_t, h_{t-1}) p(o_t \mid s_t) p(s_t \mid o_0, s_1, a_1, o_1, \ldots, o_{t-1}) p(o_0, s_1, a_1, o_1, \ldots, o_{t-1}) \\
&= \pi_t(a_t \mid o_t, h_{t-1}) p(o_t \mid s_t) p(s_t \mid s_{t-1}, a_{t-1}) p(o_0, s_1, a_1, o_1, \ldots, o_{t-1}).
\end{aligned}
$$

Hence, by induction, we obtain that

$$
p(o_0, s_1, a_1, o_1, \ldots, s_t, a_t, o_t) = \left( \prod_{i=1}^{t} \pi_i(a_i \mid o_i, h_{i-1}) p(o_i \mid s_i) \right) \left( \prod_{i=1}^{t-1} p(s_{i+1} \mid s_i, a_i) \right) p(s_1 | o_0) p(o_0).
$$

Note that $O_t \perp\!\!\!\perp A_t, A_{t-1} \mid S_t$, we can rewrite as

$$p(r_t \mid s_t, a_t)p(o_0, s_1, a_1, o_1, \ldots, s_t, a_t, o_t)$$

$$= p(r_t, o_t \mid s_t, a_t)\left(\prod_{i=1}^{t} \pi_i(a_i \mid o_i, h_{i-1})\right)\left(\prod_{i=1}^{t-1} p(s_{i+1}, o_i \mid s_i, a_i)\right) p(s_1|o_0)p(o_0).$$

We now rewrite the policy value in the vector form as

$$\mathcal{V}(\pi) = \sum_{t=1}^{T} \sum_{r_t} r_t \sum_{a_1, o_1, \ldots, a_t, o_t} \left(\prod_{i=1}^{t} \pi_i(a_i \mid o_i, h_{i-1})\right)$$

$$\mathbb{P}(O_0)\mathbb{P}(S_1|O_0)\left(\prod_{i=1}^{t-1} \mathbb{P}(S_{i+1}, o_i \mid a_i, S_i)\right)\mathbb{P}(r_t, o_t \mid a_t, S_t). \tag{13}$$

Thus, the summation uses only observable variables.

We now aim to eliminate the hidden states in (13). We invoke Lemma 2 of Tchetgen et al. (2020), and the difference lies in that we use the full history $\{O_0, H_{t-1}\}$ as a proxy to infer hidden state $S_t$ instead of one-step observation $O_{t-1}$. This yields the following key causal structure.

$$\mathbb{P}(S_t, o_{t-1} \mid a_{t-1}, S_{t-1})\mathbb{P}(S_{t+1}, o_t \mid a_t, S_t)$$

$$=\mathbb{P}(O_{t-1} \mid a_{t-1}, S_{t-1})\mathbb{P}^{\dagger}(O_{t-1} \mid a_{t-1}, O_0, H_{t-2})\mathbb{P}(O_t, o_{t-1} \mid a_{t-1}, O_0, H_{t-2}) \tag{14}$$

$$\mathbb{P}^{\dagger}(O_t \mid a_t, O_0, H_{t-1})\mathbb{P}(S_{t+1}, o_t \mid a_t, O_0, H_{t-1}),$$

and

$$\mathbb{P}(S_t, o_{t-1} \mid a_{t-1}, S_{t-1})\mathbb{P}(r_t, o_t \mid a_t, S_t)$$

$$=\mathbb{P}(O_{t-1} \mid a_{t-1}, S_{t-1})\mathbb{P}^{\dagger}(O_{t-1} \mid a_{t-1}, O_0, H_{t-2})\mathbb{P}(O_{t-1} \mid a_{t-1}, O_0, H_{t-2}) \tag{15}$$

$$\mathbb{P}^{\dagger}(O_t \mid a_t, O_0, H_{t-1})\mathbb{P}(r_t, o_t \mid a_t, O_0, H_{t-1}),$$

and

$$\mathbb{P}(S_t, o_{t-1} \mid a_{t-1}, O_0, H_{t-2})\mathbb{P}(O_t \mid a_t, S_t) = \mathbb{P}(O_t, o_{t-1} \mid a_{t-1}, O_0, H_{t-2}). \tag{16}$$

Combining these three equations (14), (15), and (16), we eliminate the latent states from (13) and yield

$$\mathbb{P}(O_0)\mathbb{P}(S_1|O_0)\left(\prod_{i=1}^{t-1} \mathbb{P}(S_{i+1}, o_i \mid a_i, S_i)\right)\mathbb{P}(r_t, o_t \mid a_t, S_t)$$

$$=\mathbb{P}(O_0)\mathbb{P}(O_1|O_0)$$

$$\mathbb{P}^{\dagger}(O_1 \mid a_1, O_0)\mathbb{P}(O_2, o_1 \mid a_1, O_0)\mathbb{P}^{\dagger}(O_2 \mid a_2, O_0, H_1)\mathbb{P}(O_3, o_2 \mid a_2, O_0, H_1)$$

$$\mathbb{P}^{\dagger}(O_3 \mid a_3, O_0, H_2)\mathbb{P}(O_4, o_3 \mid a_3, O_0, H_2)\mathbb{P}^{\dagger}(O_4 \mid a_4, O_0, H_3)\mathbb{P}(O_5, o_4 \mid a_4, O_0, H_3)$$

$$\cdots$$

$$\mathbb{P}^{\dagger}(O_{t-1} \mid a_{t-1}, O_0, H_{t-2})\mathbb{P}(O_{t-1} \mid a_{t-1}, O_0, H_{t-2})\mathbb{P}^{\dagger}(O_t \mid a_t, O_0, H_{t-1})\mathbb{P}(r_t, o_t \mid a_t, O_0, H_{t-1})$$

$$=\mathbb{P}(O_0)\mathbb{P}(O_1|O_0)\mathbb{P}^{\dagger}(O_1 \mid a_1, O_0)\left(\prod_{i=1}^{t-1} \mathbb{P}(O_{i+1}, o_i \mid a_i, O_0, H_{i-1})\mathbb{P}^{\dagger}(O_{i+1} \mid a_{i+1}, O_0, H_i)\right)$$

$$\mathbb{P}(r_t, o_t \mid a_t, O_0, H_{t-1}).$$

Here, we require $\mathbb{P}(O_t \mid a_t, S_t)$ to be invertible.

Putting all together gives the conclusion, which expresses the policy value entirely in terms of observable variables. $\qquad\square$

## B.1 Identification via value function

In this section, we present a complete proof of the identification results beyond the tabular setting.

**Theorem 5** (Identification). *Under Assumptions 4 and 5, the policy value for $\pi$ can be identified as*

$$\mathcal{V}(\pi) = \mathbb{E}\left[\sum_{a \in \mathcal{A}} b_{V,1}^{\pi}(a, O_1)\right]. \tag{17}$$

*Proof.*

$$\mathcal{V}(\pi) = \mathbb{E}^{\pi}\Big[\sum_{t=1}^{T} R_t\Big] \quad \text{(by definition of policy value)}$$

$$= \mathbb{E}_{S_1 \sim \nu_1} \sum_{t=1}^{T} \mathbb{E}^{\pi}\Big[R_t \,\Big|\, S_1\Big] \quad \text{(by the law of total expectation)}$$

$$= \mathbb{E}_{S_1 \sim \nu_1} \sum_{t=1}^{T} \mathbb{E}^{\pi}\Big[R_t \,\Big|\, S_1, H_0\Big] \quad \text{(by } H_0 = \emptyset)$$

$$= \mathbb{E}_{S_1 \sim \nu_1}\Big[\mathbb{E}\Big[\sum_{a} b_{V,1}^{\pi}(a, O_1, H_0) \,\Big|\, S_1, H_0\Big]\Big] \quad \text{(by Lemma 2)}$$

$$= \mathbb{E}_{S_1 \sim \nu_1}\Big[\mathbb{E}\Big[\sum_{a} b_{V,1}^{\pi}(a, O_1) \,\Big|\, S_1\Big]\Big] \quad \text{(by } H_0 = \emptyset)$$

$$= \mathbb{E}\Big[\sum_{a} b_{V,1}^{\pi}(a, O_1)\Big].$$

Consequently, we complete the proof. $\qquad\square$

**Assumption 1 $\Rightarrow$ Assumption 4, 5.** In tabular settings, the rank conditions are sufficient conditions for the completeness condition to hold. $\mathbb{P}(O_t \mid a_t, S_t)$ being invertible means different states map to distinguishable observation distributions. Similarly, invertibility of $\mathbb{P}(O_t \mid a_t, O_0, H_{t-1})$ ensures the history can span the full observation space. These together imply that the observable history $\{O_0, H_{t-1}\}$ contains enough information to separate functions of the hidden state $S_t$, which is exactly what completeness requires. Besides, the invertibility of $\mathbb{P}(O_t \mid a_t, O_0, H_{t-1})$ ensures the existence of value functions in Assumption 4.

**Assumption 4.** *There exist real-valued functions $\{b_{V,t}^{\pi} : \mathcal{A} \times \mathcal{O} \times \mathcal{H}_{t-1} \to \mathbb{R}\}_{t=1}^{T}$ that satisfy the following conditional moment restrictions:*

$$\mathbb{E}\Big[b_{V,t}^{\pi}(A_t, O_t, H_{t-1}) \,\big|\, O_0, A_t, H_{t-1}\Big]$$
$$= \mathbb{E}\Big[R_t \pi_t(A_t \mid O_t, H_{t-1}) + \sum_{a'} b_{V,t+1}^{\pi}(a', O_{t+1}, H_t)\,\pi_t(A_t \mid O_t, H_{t-1}) \,\big|\, O_0, A_t, H_{t-1}\Big].$$

Beyond the tabular setting, the existence of these value functions is justified by some mild regularity conditions utilizing tools from singular value decomposition in functional analysis (Kress et al., 1989).

**Assumption 5** (Completeness). *For any measurable function $g_t : \mathcal{S} \times \mathcal{A} \times \mathcal{H}_{t-1} \to \mathbb{R}$ and any $1 \le t \le T$,*

$$\mathbb{E}\big[g_t(S_t, A_t, H_{t-1}) \mid A_t, H_{t-1}, O_0\big] = 0$$

*almost surely if and only if $g_t(S_t, A_t, H_{t-1}) = 0$ almost surely.*

The completeness assumption is widely used in statistics and econometrics. For example, it plays a crucial role in instrumental variable estimation, where the identification of structural functions often depends on the completeness (Newey and Powell, 2003; Hu and Shiu, 2018).

Under Assumptions 4 and 5, we obtain another sequence of conditional moment restrictions (18) that are projected onto $(S_t, A_t, H_{t-1})$.

$$\mathbb{E}\Big[b_{V,t}^{\pi}(A_t, H_{t-1}, O_t) \,\big|\, A_t, H_{t-1}, S_t\Big] = \mathbb{E}\Big[R_t \pi_t(A_t \mid O_t, H_{t-1})$$
$$+ \sum_{a' \in \mathcal{A}} b_{V,t+1}^{\pi}(a', H_t, O_{t+1})\pi_t(A_t \mid O_t, H_{t-1}) \,\big|\, A_t, H_{t-1}, S_t\Big]. \tag{18}$$

**Lemma 2.** *Under Assumptions 4 and 5, for all $t = 1, ..., T$, it holds that*

$$\mathbb{E}\Big[\sum_{a} b_{V,t}^{\pi}(a, O_t, H_{t-1}) \mid S_t, H_{t-1}\Big] = \sum_{j=t}^{T} \mathbb{E}^{\pi}\Big[R_t \mid S_t, H_{t-1}\Big]. \tag{19}$$

## B.2 Proof of Lemma 2

*Proof.* We prove it by induction. At the step $t = T$, we have

$$\mathbb{E}^\pi \left[ R_T \mid S_T, H_{T-1} \right]$$

$$= \mathbb{E} \left[ \mathbb{E}^\pi \left[ \mathbb{E} \left[ R_T \mid S_T, H_{T-1}, O_T, A_T \right] \mid S_T, H_{T-1}, O_T \right] \mid S_T, H_{T-1} \right]$$

(by law of total expectation)

$$= \mathbb{E}^\pi \left[ \mathbb{E} \left[ \sum_a \mathbb{E} \left[ R_T \mid S_T, H_{T-1}, O_T, A_T = a \right] \pi_T(a \mid O_T, H_{T-1}) \mid S_T, H_{T-1}, O_T \right] \mid S_T, H_{T-1} \right]$$

$$= \mathbb{E} \left[ \sum_a \mathbb{E} \left[ R_T \mid S_T, H_{T-1}, O_T, A_T = a \right] \pi_T(a \mid O_T, H_{T-1}) \mid S_T, H_{T-1} \right]$$

$$= \mathbb{E} \left[ \sum_a \mathbb{E} \left[ R_T \mid S_T, H_{T-1}, A_T = a \right] \pi_T(a \mid O_T, H_{T-1}) \mid S_T, H_{T-1} \right]$$

(by $R_T \perp\!\!\!\perp O_T \mid S_T, A_T, H_{T-1}$)

$$= \sum_a \mathbb{E} \left[ R_T \mid S_T, H_{T-1}, A_T = a \right] \mathbb{E} \left[ \pi_T(a \mid O_T, H_{T-1}) \mid S_T, H_{T-1} \right]$$

$$= \sum_a \mathbb{E} \left[ R_T \mid S_T, H_{T-1}, A_T = a \right] \mathbb{E} \left[ \pi_T(a \mid O_T, H_{T-1}) \mid S_T, H_{T-1}, A_T = a \right]$$

(by $O_T \perp\!\!\!\perp A_T \mid S_T, H_{T-1}$)

$$= \sum_a \mathbb{E} \left[ R_T \pi_T(a \mid O_T, H_{T-1}) \mid S_T, H_{T-1}, A_T = a \right] \quad \text{(by } O_T \perp\!\!\!\perp R_T \mid S_T, A_T, H_{T-1})$$

$$= \sum_a \mathbb{E} \left[ b_{V,T}^\pi(O_T, H_{T-1}, a) \mid S_T, H_{T-1}, A_T = a \right] \quad \text{(by Equation 18)}$$

$$= \sum_a \mathbb{E} \left[ b_{V,T}^\pi(O_T, H_{T-1}, a) \mid S_T, H_{T-1} \right] \quad \text{(by } O_T \perp\!\!\!\perp A_T \mid S_T, H_{T-1})$$

$$= \mathbb{E} \left[ \sum_a b_{V,T}^\pi(a, O_T, H_{T-1}) \mid S_T, H_{T-1} \right]$$

According to the above derivation, we have shown $\mathbb{E} \left[ \sum_a b_{V,j}^\pi(a, O_j, H_{j-1}) \mid S_j, H_{j-1} \right] = \mathbb{E}^\pi \left[ \sum_{t=j}^T R_t \mid S_j, H_{j-1} \right]$ when $j = T$. We proceed with the derivation by induction. Assume that $\mathbb{E} \left[ \sum_a b_{V,j}^\pi(a, O_j, H_{j-1}) \mid S_j, H_{j-1} \right] = \mathbb{E}^\pi \left[ \sum_{t=j}^T R_t \mid S_j, H_{j-1} \right]$ holds for $j = k + 1$, we will show that it also holds for $j = k$.

For $j = k$, we first notice that

$$\mathbb{E}^\pi \left[ \sum_{t=k}^T R_t \mid S_k, H_{k-1} \right] = \mathbb{E}^\pi \left[ R_k \mid S_k, H_{k-1} \right] + \mathbb{E}^\pi \left[ \sum_{t=k+1}^T R_t \mid S_k, H_{k-1} \right].$$

Next, we analyze these two terms separately. Analyzing the first term is the same as $\mathbb{E}^\pi[R_T \mid S_T, H_{T-1}]$ by replacing $T$ with $k$.

$$\mathbb{E}^\pi\left[R_k \mid S_k, H_{k-1}\right]$$

$$=\mathbb{E}\left[\mathbb{E}^\pi\left[\mathbb{E}\left[R_k \mid S_k, H_{k-1}, O_k, A_k\right] \Big| S_k, H_{k-1}, O_k\right] \Big| S_k, H_{k-1}\right]$$

$$\text{(by law of total expectation)}$$

$$=\mathbb{E}^\pi\left[\mathbb{E}\left[\sum_a \mathbb{E}\left[R_k \mid S_k, H_{k-1}, O_k, A_k = a\right]\pi_k(a \mid O_k, H_{k-1}) \Big| S_k, H_{k-1}, O_k\right] \Big| S_k, H_{k-1}\right]$$

$$=\mathbb{E}\left[\sum_a \mathbb{E}\left[R_k \mid S_k, H_{k-1}, O_k, A_k = a\right]\pi_k(a \mid O_k, H_{k-1}) \Big| S_k, H_{k-1}\right]$$

$$=\mathbb{E}\left[\sum_a \mathbb{E}\left[R_k \mid S_k, H_{k-1}, A_k = a\right]\pi_k(a \mid O_k, H_{k-1}) \Big| S_k, H_{k-1}\right]$$

$$\text{(by } R_k \perp\!\!\!\perp O_k \mid S_k, A_k, H_{k-1})$$

$$=\sum_a \mathbb{E}\left[R_k \mid S_k, H_{k-1}, A_k = a\right]\mathbb{E}\left[\pi_k(a \mid O_k, H_{k-1}) \Big| S_k, H_{k-1}\right]$$

$$=\sum_a \mathbb{E}\left[R_k \mid S_k, H_{k-1}, A_k = a\right]\mathbb{E}\left[\pi_k(a \mid O_k, H_{k-1}) \Big| S_k, H_{k-1}, A_k = a\right]$$

$$\text{(by } O_k \perp\!\!\!\perp A_k \mid S_k, H_{k-1})$$

$$=\sum_a \mathbb{E}\left[R_k\pi_k(a \mid O_k, H_{k-1}) \Big| S_k, H_{k-1}, A_k = a\right]$$

$$\text{(by } O_k \perp\!\!\!\perp R_k \mid S_k, A_k, H_{k-1})$$

$$=\sum_a \mathbb{E}\left[R_k\pi_k(a \mid O_k, H_{k-1}) \Big| S_k, H_{k-1}, A_k = a\right]$$

$$\tag{20}$$

For the second term, we have

$$\mathbb{E}^\pi\left[\sum_{t=k+1}^{T} R_t \;\middle|\; S_k, H_{k-1}\right]$$

$$=\mathbb{E}^\pi\left[\mathbb{E}^\pi\left[\sum_{t=k+1}^{T} R_t \;\middle|\; S_{k+1}, H_k, S_k\right] \;\middle|\; S_k, H_{k-1}\right] \text{ (by law of total expectation)}$$

$$=\mathbb{E}^\pi\left[\mathbb{E}^\pi\left[\sum_{t=k+1}^{T} R_t \;\middle|\; S_{k+1}, H_k\right] \;\middle|\; S_k, H_{k-1}\right] \text{ (by } \{R_t\}_{t=k+1}^{T} \perp\!\!\!\perp S_k \mid S_{k+1}, H_k)$$

$$=\mathbb{E}^\pi\left[\mathbb{E}\left[\sum_{a'} b_{V,k+1}^\pi(a', O_{k+1}, H_k) \;\middle|\; S_{k+1}, H_k\right] \;\middle|\; S_k, H_{k-1}\right]$$

$$=\mathbb{E}^\pi\left[\mathbb{E}\left[\sum_{a'} b_{V,k+1}^\pi(a', O_{k+1}, H_k) \;\middle|\; S_{k+1}, H_k, S_k\right] \;\middle|\; S_k, H_{k-1}\right] \;(O_{k+1} \perp\!\!\!\perp S_k \mid S_{k+1}, H_k)$$

$$=\mathbb{E}^\pi\left[\sum_{a'} b_{V,k+1}^\pi(a', O_{k+1}, H_k) \;\middle|\; S_k, H_{k-1}\right] \text{ (by law of total expectation)}$$

$$=\mathbb{E}^\pi\left[\mathbb{E}\left[\sum_{a'} b_{V,k+1}^\pi(a', O_{k+1}, H_k) \;\middle|\; S_k, H_{k-1}, O_k, A_k\right] \;\middle|\; S_k, H_{k-1}\right] \text{ (by law of total expectation)}$$

$$=\mathbb{E}\left[\sum_{a} \mathbb{E}\left[\sum_{a'} b_{V,k+1}^\pi(a', O_{k+1}, H_k) \;\middle|\; S_k, H_{k-1}, O_k, A_k = a\right] \pi_k(a \mid O_k, H_{k-1}) \;\middle|\; S_k, H_{k-1}\right]$$

$$=\mathbb{E}\left[\sum_{a} \mathbb{E}\left[\sum_{a'} b_{V,k+1}^\pi(a', O_{k+1}, H_k)\pi_k(a \mid O_k, H_{k-1}) \;\middle|\; S_k, H_{k-1}, O_k, A_k = a\right] \;\middle|\; S_k, H_{k-1}\right]$$

$$=\sum_{a}\mathbb{E}\left[\mathbb{E}\left[\sum_{a'} b_{V,k+1}^\pi(a', O_{k+1}, H_k)\pi_k(a \mid O_k, H_{k-1}) \;\middle|\; S_k, H_{k-1}, O_k, A_k = a\right] \;\middle|\; S_k, H_{k-1}\right]$$

$$=\sum_{a}\mathbb{E}\left[\mathbb{E}\left[\sum_{a'} b_{V,k+1}^\pi(a', O_{k+1}, H_k)\pi_k(a \mid O_k, H_{k-1}) \;\middle|\; S_k, H_{k-1}, O_k, A_k = a\right] \;\middle|\; S_k, H_{k-1}, A_k = a\right]$$

$$\text{(by } O_k \perp\!\!\!\perp A_k \mid S_k)$$

$$=\sum_{a}\mathbb{E}\left[\sum_{a'} b_{V,k+1}^\pi(a', O_{k+1}, H_k)\pi_k(a \mid O_k, H_{k-1}) \;\middle|\; S_k, H_{k-1}, A_k = a\right] \text{ (by law of total expectation)}$$

$$\tag{21}$$

Combining Equations (20) and (21), we have

$$\mathbb{E}^\pi\left[\sum_{t=k}^{T} R_t \;\middle|\; S_k, H_{k-1}\right]$$

$$=\mathbb{E}^\pi\left[R_k \;\middle|\; S_k, H_{k-1}\right] + \mathbb{E}^\pi\left[\sum_{t=k+1}^{T} R_t \;\middle|\; S_k, H_{k-1}\right]$$

$$=\sum_{a}\mathbb{E}\left[R_k\pi_k(a \mid O_k, H_{k-1}) \;\middle|\; S_k, H_{k-1}, A_k = a\right]$$

$$+\sum_{a}\mathbb{E}\left[\sum_{a'} b_{V,k+1}^\pi(a', O_{k+1}, H_k)\pi_k(a \mid O_k, H_{k-1}) \;\middle|\; S_k, H_{k-1}, A_k = a\right]$$

$$\text{(by Equations (20) and (21))}$$

$$=\sum_{a}\mathbb{E}\left[R_k\pi_k(a \mid O_k, H_{k-1}) + \sum_{a'} b_{V,k+1}^\pi(a', O_{k+1}, H_k)\pi_k(a \mid O_k, H_{k-1}) \;\middle|\; S_k, H_{k-1}, A_k = a\right]$$

$$= \sum_a \mathbb{E}\left[ b_{V,k}^\pi(a, O_k, H_{k-1}) \,\Big|\, S_k, H_{k-1}, A_k = a \right] \quad \text{(by Equation 18)}$$

$$= \sum_a \mathbb{E}\left[ b_{V,k}^\pi(a, O_k, H_{k-1}) \,\Big|\, S_k, H_{k-1} \right] \quad \text{(by } O_k \perp\!\!\!\perp A_k \,\Big|\, S_k, H_{k-1})$$

$$= \mathbb{E}\left[ \sum_a b_{V,k}^\pi(a, O_k, H_{k-1}) \,\Big|\, S_k, H_{k-1} \right]$$

Therefore, $\mathbb{E}^\pi\left[ \sum_{t=k}^T R_t \,\Big|\, S_k, H_{k-1} \right] = \mathbb{E}\left[ b_{V,k}^\pi(O_k, H_{k-1}) \,\Big|\, S_k, H_{k-1} \right]$ also holds for $j = k$, if it holds for $j = k + 1$. By the induction argument, the proof is completed. $\qquad\square$

### B.3 Identification via weight functions

We provide an alternative identification formula with weight functions $\{b_{W,t}^\pi\}_{t=1}^T$.

**Theorem 6** (Identification with weight functions). *Under Assumptions 4 and 5, the policy value can be identified by weight functions as*

$$\mathcal{V}(\pi) = \mathbb{E}\Big[ \sum_{t=1}^T R_t \pi_t(A_t \mid O_t, H_{t-1}) b_{W,t}^\pi(A_t, H_{t-1}, O_0) \Big].$$

*Proof.*

$$\mathcal{V}(\pi) = \sum_{t=1}^T \mathbb{E}^\pi\left[ R_t(S_t, A_t) \right]$$

$$= \sum_{t=1}^T \mathbb{E}^\pi\left[ \mathbb{E}^\pi\left[ R_t(S_t, A_t) \mid O_t, S_t, H_{t-1} \right] \right]$$

$$= \sum_{t=1}^T \mathbb{E}^\pi\left[ \sum_a R_t(S_t, a) \pi_t(a \mid O_t, H_{t-1}) \right]$$

$$= \sum_{t=1}^T \mathbb{E}^\pi\left[ \mathbb{E}^\pi\left[ \sum_a R_t(S_t, a) \pi_t(a | O_t, H_{t-1}) \,\Big|\, S_t, H_{t-1} \right] \right]$$

$$= \sum_{t=1}^T \mathbb{E}\left[ \omega_t(S_t, H_{t-1}) \cdot \mathbb{E}^\pi\left[ \sum_a R_t(S_t, a) \pi_t(a \mid O_t, H_{t-1}) \,\Big|\, S_t, H_{t-1} \right] \right]$$

$$= \sum_{t=1}^T \mathbb{E}\left[ \omega_t(S_t, H_{t-1}) \sum_{a \in \mathcal{A}} R_t(S_t, a) \pi_t(a | O_t, H_{t-1}) \right]$$

$$= \sum_{t=1}^T \mathbb{E}\left[ \omega_t(S_t, H_{t-1}) \sum_{a \in \mathcal{A}} R_t(S_t, a) \pi_t^b(a | S_t) \frac{\pi_t(a | O_t, H_{t-1})}{\pi_t^b(a | S_t)} \right]$$

$$= \sum_{t=1}^T \mathbb{E}\left[ \mathbb{E}\left[ \omega_t(S_t, H_{t-1}) R_t(S_t, A_t) \frac{\pi_t(A_t | O_t, H_{t-1})}{\pi_t^b(A_t \mid S_t)} \,\Big|\, S_t, H_{t-1}, O_t \right] \right] \quad \text{(by } A_t \sim \pi_t^b(\cdot | S_t))$$

$$= \sum_{t=1}^T \mathbb{E}\left[ R_t(S_t, A_t) \pi_t(A_t \mid O_t, H_{t-1}) \frac{\omega_t(S_t, H_{t-1})}{\pi_t^b(A_t \mid S_t)} \right]$$

$$= \sum_{t=1}^T \mathbb{E}\left[ R_t(S_t, A_t) \pi_t(A_t \mid O_t, H_{t-1}) \mathbb{E}\left[ b_{W,t}^\pi(A_t, H_{t-1}, O_0) \,\Big|\, S_t, H_{t-1}, A_t \right] \right] \quad \text{(by Equation 18)}$$

$$= \sum_{t=1}^T \mathbb{E}\left[ \mathbb{E}\left[ R_t(S_t, A_t) \pi_t(A_t \mid O_t, H_{t-1}) b_{W,t}^\pi(A_t, H_{t-1}, O_0) \,\Big|\, S_t, H_{t-1}, A_t \right] \right]$$

$$(O_0 \perp\!\!\!\perp (O_t, R_t) \mid S_t, A_t, H_{t-1})$$

$$= \sum_{t=1}^{T} \mathbb{E}\left[ R_t(S_t, A_t)\pi_t(A_t|O_t, H_{t-1})b_{W,t}^{\pi}(A_t, H_{t-1}, O_0) \right].$$

$\square$

## C  Proof of Theorem 2 and 3

In this section, we provide the proof for Theorems 2 and 3 in the main text. We only show the proof for the fully history-dependent policy, as the memoryless case can be derived straightforwardly by excluding the history $H_{t-1}$ from the conditional probabilities $\mathbf{P}_{a_t}$ and $\mathbf{P}_{a_t, r_t, o_{t+1}}$. Before proceeding with the analysis, we first recall and introduce some notations.

**Additional Notations.**  For $t = 1, \ldots, T$, let $\mathcal{D}_t$ represent the collection of historical data up to time step $t$, i.e., $\mathcal{D}_t = \{o_0^i, (o_{t'}^i, a_{t'}^i, r_{t'}^i)_{t'=1}^t\}_{i=1}^n$ with $\mathcal{D}_0 = \{o_0^i\}_{i=1}^n$. For simplicity, we omit the superscript $\pi^b$ when referring to the expectation, variance, or probability under the distribution induced by $\pi^b$. To distinguish between different sources of randomness, we use $\mathcal{E}$ to denote expectations over random variables (capital letters) and $\mathbb{E}$ to represent expectations over offline data, when both $\mathcal{E}$ and $\mathbb{E}$ appear simultaneously. We use $\mathbf{I}$ to represent the identity matrix, with its dimension being clear from the context. If a non-negative random variable $X$ satisfies $P(X \leq c\Xi(n, T)) \to 0$ as $c \to 0$ for any $n, T$, we write $X = \mathcal{O}_P(\Xi(n, T))$ with high probability.

Note that we assume a sufficiently large $n$ in Assumption 2(c), where we require a sufficient number of samples for each triple $(o_0, h_{t-1}, a_t)$, ensuring consistent estimation of the conditional probability matrices. Specifically, we require $n_{o_0, h_{t-1}, a_t} \geq np_t^{\pi^b}(o_0, h_{t-1}, a_t)(1 - \theta_{t,ij})$, where $n_{o_0, h_{t-1}, a_t}$ represents the count of the triple $(o_0, h_{t-1}, a_t)$ in the data, and $p_t^{\pi^b}(o_0, h_{t-1}, a_t)$ is the probability density under the behavior policy $\pi^b$. Define the event $E := \{\exists\, t, o_0, h_{t-1}, a_t \text{ s.t. } n_{o_0, h_{t-1}, a_t} \geq np_t^{\pi^b}(o_0, h_{t-1}, a_t)(1 - \theta_{t,ij})\}$. Then, combining the multiplicative Chernoff bound and a union bound over each $t, o_0, h_{t-1}$, and $a_t$, we have

$$\mathbb{P}[E^c] \leq \sum_t \sum_{o_0} \sum_{h_{t-1}} \sum_{a_t} \mathbb{P}\left[ n_{o_0, h_{t-1}, a_t} < np_t^{\pi^b}(o_0, h_{t-1}, a_t)(1 - \theta_{t,ij}) \right]$$

$$\leq T|\mathcal{O}|^T|\mathcal{A}|^T e^{-\frac{\theta * 2 n \min_{t, o_0, h_{t-1}, a_t} p_t^{\pi^b}(o_0, h_{t-1}, a_t)}{2}}.$$

To ensure the number of samples is sufficiently large, the $\mathbb{P}[E^c]$ should be sufficiently small. This requires the sample size satisfying $n \geq \frac{\mathrm{polylog}(|\mathcal{O}|^T, |\mathcal{A}|^T, T)}{\min_{t, o_0, h_{t-1}, a_t} p_t^{\pi^b}(o_0, h_{t-1}, a_t)}$.

Then, we consider the following decomposition of the error

$$\mathcal{V}(\pi) - \widehat{\mathcal{V}}(\pi) = \mathbb{E}\left[ \sum_a b_{V,1}^{\pi}(a, O_1) \right] - \widehat{\mathbb{E}}\left[ \sum_a \widehat{b}_{V,1}(a, O_1) \right]$$

$$= \mathbb{E}\left[ \sum_a b_{V,1}^{\pi}(a, O_1) \right] - \mathbb{E}\left[ \sum_a \widehat{b}_{V,1}(a, O_1) \right]$$

$$+ \mathbb{E}\left[ \sum_a \widehat{b}_{V,1}(a, O_1) \right] - \widehat{\mathbb{E}}\left[ \sum_a \widehat{b}_{V,1}(a, O_1) \right].$$

We begin by analyzing the sub-optimality term $\mathbb{E}[\sum_a b_{V,1}^{\pi}(a, O_1)] - \mathbb{E}[\sum_a \widehat{b}_{V,1}(a, O_1)]$. The second term $\mathbb{E}[\sum_a \widehat{b}_{V,1}(a, O_1)] - \widehat{\mathbb{E}}[\sum_a \widehat{b}_{V,1}(a, O_1)]$ can be upper bounded by the uniform law of large numbers according to the empirical processes.

### C.1  Bounding $\mathbb{E}[\sum_a b_{V,1}^{\pi}(a, O_1)] - \mathbb{E}[\sum_a \widehat{b}_{V,1}(a, O_1)]$

We first decompose the sub-optimality term into three parts. Note that

$$\mathbb{E}\left[ \sum_a b_{V,1}^{\pi}(a, O_1) - \sum_a \widehat{b}_{V,1}(a, O_1) \right]$$

$$=\mathcal{E}\left[\sum_{a_1}b_{V,1}^\pi(a_1,O_1)-\sum_{a_1}\pi_1(a_1|O_1)\psi_1^\top(O_1)\widehat{\mathbf{P}}_{a_1}^\dagger\sum_{r_1,o_2}\widehat{\mathbf{P}}_{a_1,r_1,o_2}\psi_1(O_1)\Big(r_1+\sum_{a_2}\widehat{b}_{V,2}\big(a_2,o_2,(O_1,a_1)\big)\Big)\right]$$

$$=\mathcal{E}\left[\sum_{a_1}b_{V,1}^\pi(a_1,O_1)-\sum_{a_1}\pi_1(a_1|O_1)\psi_1^\top(O_1)\widehat{\mathbf{P}}_{a_1}^\dagger\sum_{r_1,o_2}\widehat{\mathbf{P}}_{a_1,r_1,o_2}\psi_1(O_1)\Big(r_1+\sum_{a_2}b_{V,2}^\pi\big(a_2,o_2,(O_1,a_1)\big)\Big)\right.$$

$$\left.+\sum_{a_1}\pi_1(a_1|O_1)\psi_1^\top(O_1)\widehat{\mathbf{P}}_{a_1}^\dagger\sum_{r_1,o_2}\widehat{\mathbf{P}}_{a_1,r_1,o_2}\psi_1(O_1)\Big(\sum_{a_2}b_{V,2}^\pi\big(a_2,o_2,(O_1,a_1)\big)-\sum_{a_2}\widehat{b}_{V,2}\big(a_2,o_2,(O_1,a_1)\big)\Big)\right]$$

$$=\mathcal{E}\left[\sum_{a_1}b_{V,1}^\pi(a_1,O_1)-\sum_{a_1}\pi_1(a_1|O_1)\psi_1^\top(O_1)\widehat{\mathbf{P}}_{a_1}^\dagger\sum_{r_1,o_2}\widehat{\mathbf{P}}_{a_1,r_1,o_2}\psi_1(O_1)\Big(r_1+\sum_{a_2}b_{V,2}^\pi\big(a_2,o_2,(O_1,a_1)\big)\Big)\right.$$

$$+\sum_{a_1}\pi_1(a_1|O_1)\psi_1^\top(O_1)\widehat{\mathbf{P}}_{a_1}^\dagger\sum_{r_1,o_2}\widehat{\mathbf{P}}_{a_1,r_1,o_2}\psi_1(O_1)\Big(\sum_{a_2}b_{V,2}^\pi\big(a_2,o_2,(O_1,a_1)\big)-\sum_{a_2}\pi_2\big(a_2|o_2,(O_1,a_1)\big)$$

$$\left.\psi_2^\top(o_2)\widehat{\mathbf{P}}_{a_2}^\dagger\sum_{r_2,o_3}\widehat{\mathbf{P}}_{a_2,r_2,o_3}\psi_2(o_2)\Big(r_2+\sum_{a_3}\widehat{b}_{V,3}\big(a_3,o_3,(O_1,a_1,o_2,a_2)\big)\Big)\Big)\right]$$

$$=\cdots$$

$$=\mathcal{E}\big(E_1\big)+\mathcal{E}\big(E_2\big)+\mathcal{E}\big(E_3\big).$$

We replace the data dependent terms $\widehat{\mathbf{P}}_{a_t}^\dagger$ and $\widehat{\mathbf{P}}_{a_t,r_t,o_{t+1}}^\dagger$ with their population counterparts $\mathbf{P}_{a_t}^\dagger$ and $\mathbf{P}_{a_t,r_t,o_{t+1}}^\dagger$, respectively. This error decomposition is then the sum of $\mathcal{E}(E_1)$, $\mathcal{E}(E_2)$ and $\mathcal{E}(E_3)$, where

$$E_1=\sum_{a_1}\pi_1\psi_1^\top\mathbf{P}_{a_1}^\dagger\sum_{r_1,o_2}\widehat{\mathbf{P}}_{a_1,r_1,o_2}\psi_1\left[\sum_{a_1}b_{V,1}^\pi(a_1,O_1)-\Big(r_1+\sum_{a_2}b_{V,2}^\pi\big(a_2,o_2,(O_1,a_1)\big)\Big)\right]$$

$$+\sum_{a_1}\pi_1\psi_1^\top\mathbf{P}_{a_1}^\dagger\sum_{r_1,o_2}\mathbf{P}_{a_1,r_1,o_2}\psi_1\sum_{a_2}\pi_2\psi_2^\top\mathbf{P}_{a_2}^\dagger\sum_{r_2,o_3}\widehat{\mathbf{P}}_{a_2,r_2,o_3}\psi_2$$

$$\left[\sum_{a_2}b_{V,2}^\pi\big(a_2,o_2,(O_1,a_1)\big)-\Big(r_2+\sum_{a_2}b_{V,3}^\pi\big(a_3,o_3,(O_1,a_1,o_2,a_2)\big)\Big)\right]$$

$$+\cdots$$

$$+\sum_{a_1}\pi_1\psi_1^\top\mathbf{P}_{a_1}^\dagger\sum_{r_1,o_2}\mathbf{P}_{a_1,r_1,o_2}\psi_1\sum_{a_2}\pi_2\psi_2^\top\mathbf{P}_{a_2}^\dagger\sum_{r_2,o_3}\mathbf{P}_{a_2,r_2,o_3}\psi_2\cdots\sum_{a_T}\pi_T\psi_T^\top\mathbf{P}_{a_T}^\dagger\sum_{r_T}\widehat{\mathbf{P}}_{a_T,r_T}\psi_T$$

$$\left[\sum_{a_T}b_{V,T}^\pi\big(a_T,o_T,(O_1,a_1,o_2,\cdots,a_{T-1})\big)-r_T\right],$$

$$E_2=\left(\sum_{a_1}\pi_1\psi_1^\top\widehat{\mathbf{P}}_{a_1}^\dagger\sum_{r_1,o_2}\widehat{\mathbf{P}}_{a_1,r_1,o_2}\psi_1-\sum_{a_1}\pi_1\psi_1^\top\mathbf{P}_{a_1}^\dagger\sum_{r_1,o_2}\widehat{\mathbf{P}}_{a_1,r_1,o_2}\psi_1\right)$$

$$\left[\sum_{a_1}b_{V,1}^\pi(a_1,O_1)-\Big(r_1+\sum_{a_2}b_{V,2}^\pi\big(a_2,o_2,(O_1,a_1)\big)\Big)\right],$$

$$+\left(\sum_{a_1}\pi_1\psi_1^\top\widehat{\mathbf{P}}_{a_1}^\dagger\sum_{r_1,o_2}\widehat{\mathbf{P}}_{a_1,r_1,o_2}\psi_1\sum_{a_2}\pi_2\psi_2^\top\widehat{\mathbf{P}}_{a_2}^\dagger\sum_{r_2,o_3}\widehat{\mathbf{P}}_{a_2,r_2,o_3}\psi_2\right.$$

$$\left.-\sum_{a_1}\pi_1\psi_1^\top\mathbf{P}_{a_1}^\dagger\sum_{r_1,o_2}\mathbf{P}_{a_1,r_1,o_2}\psi_1\sum_{a_2}\pi_2\psi_2^\top\mathbf{P}_{a_2}^\dagger\sum_{r_2,o_3}\widehat{\mathbf{P}}_{a_2,r_2,o_3}\psi_2\right)$$

$$\left[\sum_{a_2}b_{V,2}^\pi\big(a_2,o_2,(O_1,a_1)\big)-\Big(r_2+\sum_{a_2}b_{V,3}^\pi\big(a_3,o_3,(O_1,a_1,o_2,a_2)\big)\Big)\right]$$

$$\cdots$$

$$+\left(\sum_{a_1}\pi_1\psi_1^\top\widehat{\mathbf{P}}_{a_1}^\dagger\sum_{r_1,o_2}\widehat{\mathbf{P}}_{a_1,r_1,o_2}\psi_1\sum_{a_2}\pi_2\psi_2^\top\widehat{\mathbf{P}}_{a_2}^\dagger\sum_{r_2,o_3}\widehat{\mathbf{P}}_{a_2,r_2,o_3}\psi_2\cdots\sum_{a_T}\pi_T\psi_T^\top\widehat{\mathbf{P}}_{a_T}^\dagger\sum_{r_T}\widehat{\mathbf{P}}_{a_T,r_T}\psi_T\right.$$

$$-\sum_{a_1}\pi_1\psi_1^\top\mathbf{P}^\dagger_{a_1}\sum_{r_1,o_2}\mathbf{P}_{a_1,r_1,o_2}\psi_1\sum_{a_2}\pi_2\psi_2^\top\mathbf{P}^\dagger_{a_2}\sum_{r_2,o_3}\mathbf{P}_{a_2,r_2,o_3}\psi_2\cdots\sum_{a_T}\pi_T\psi_T^\top\mathbf{P}^\dagger_{a_T}\sum_{r_T}\widehat{\mathbf{P}}_{a_T,r_T}\psi_T\bigg)$$

$$\bigg[\sum_{a_T}b^\pi_{V,T}\big(a_T,o_T,(O_1,a_1,o_2,\cdots,a_{T-1})\big)-r_T\bigg],$$

$$E_3=\sum_{a_1}b^\pi_{V,1}(a_1,O_1)-\sum_{a_1}\pi_1\psi_1^\top\widehat{\mathbf{P}}^\dagger_{a_1}\sum_{r_1,o_2}\widehat{\mathbf{P}}_{a_1,r_1,o_2}\psi_1\sum_{a_1}b^\pi_{V,1}(a_1,O_1)$$

$$+\sum_{a_1}\pi_1\psi_1^\top\widehat{\mathbf{P}}^\dagger_{a_1}\sum_{r_1,o_2}\widehat{\mathbf{P}}_{a_1,r_1,o_2}\psi_1\bigg[\sum_{a_2}b^\pi_{V,2}-\sum_{a_2}\pi_2\psi_2^\top\widehat{\mathbf{P}}^\dagger_{a_2}\sum_{r_2,o_3}\widehat{\mathbf{P}}_{a_2,r_2,o_3}\psi_2\sum_{a_2}b^\pi_{V,2}\bigg]$$

$$+\cdots$$

$$+\sum_{a_1}\pi_1\psi_1^\top\widehat{\mathbf{P}}^\dagger_{a_1}\sum_{r_1,o_2}\widehat{\mathbf{P}}_{a_1,r_1,o_2}\psi_1\cdots\sum_{a_{T-1}}\pi_T\psi_{T-1}^\top\widehat{\mathbf{P}}^\dagger_{a_{T-1}}\sum_{r_{T-1},o_T}\widehat{\mathbf{P}}_{a_{T-1},r_{T-1},o_T}\psi_{T-1}$$

$$\bigg[\sum_{a_T}b^\pi_{V,T}-\sum_{a_T}\pi_T\psi_T^\top\widehat{\mathbf{P}}^\dagger_{a_T}\sum_{r_T}\widehat{\mathbf{P}}_{a_T,r_T}\psi_T\sum_{a_T}b^\pi_{V,T}\bigg].$$

Note that these terms ultimately decompose into a sum over all possible observable history trajectories. Simplifying these terms, we have

$$E_1=\sum_{t=1}^T\bigg(\sum_{a_1}\pi_1\psi_1^\top\mathbf{P}^\dagger_{a_1}\cdots\sum_{r_t,o_{t+1}}\widehat{\mathbf{P}}_{a_t,r_t,o_{t+1}}\psi_t\bigg)\bigg[\sum_{a_t}b^\pi_{V,t}-\Big(r_t+\sum_{a_{t+1}}b^\pi_{V,t+1}\Big)\bigg]$$

$$E_2=\sum_{t=1}^T\bigg(\sum_{a_1}\pi_1\psi_1^\top\widehat{\mathbf{P}}^\dagger_{a_1}\cdots\sum_{r_t,o_{t+1}}\widehat{\mathbf{P}}_{a_t,r_t,o_{t+1}}\psi_t-\sum_{a_1}\pi_1\psi_1^\top\mathbf{P}^\dagger_{a_1}\cdots\sum_{r_t,o_{t+1}}\widehat{\mathbf{P}}_{a_t,r_t,o_{t+1}}\psi_t\bigg)$$

$$\bigg[\sum_{a_t}b^\pi_{V,t}-\Big(r_t+\sum_{a_{t+1}}b^\pi_{V,t+1}\Big)\bigg]$$

$$E_3=\sum_{t=1}^T\bigg(\sum_{a_1}\pi_1\psi_1^\top\widehat{\mathbf{P}}^\dagger_{a_1}\cdots\sum_{r_{t-1},o_t}\widehat{\mathbf{P}}_{a_{t-1},r_{t-1},o_t}\psi_{t-1}\bigg[\sum_{a_t}b^\pi_{V,t}-\sum_{a_t}\pi_t\psi_t^\top\widehat{\mathbf{P}}^\dagger_{a_t}\sum_{r_t,o_{t+1}}\widehat{\mathbf{P}}_{a_t,r_t,o_{t+1}}\psi_t\sum_{a_t}b^\pi_{V,t}\bigg]\bigg).$$

It is straightforward to show $E_3=0$ by noticing that

$$\sum_{a_t}\pi_t(a_t|o_t,h_{t-1})\psi_t^\top(o_t)\widehat{\mathbf{P}}^\dagger_{a_t}\sum_{r_t,o_{t+1}}\widehat{\mathbf{P}}_{a_t,r_t,o_{t+1}}\psi_t(o_t)\sum_{a_t}b^\pi_{V,t}(a_t,o_t,h_{t-1})$$

$$=\sum_{a_t}\pi_t(a_t|o_t,h_{t-1})\psi_t^\top(o_t)\widehat{\mathbf{P}}^\dagger_{a_t}\widehat{\mathbf{P}}_{a_t}\psi_t(o_t)\sum_{a_t}b^\pi_{V,t}(a_t,o_t,h_{t-1})$$

$$=\psi_t^\top(o_t)\psi_t(o_t)\sum_{a_t}b^\pi_{V,t}(a_t,o_t,h_{t-1})$$

$$=\sum_{a_t}b^\pi_{V,t}(a_t,o_t,h_{t-1}).$$

Throughout the proof we use the trick that

$$\sum_{a_t}b^\pi_{V,t}(a_t,o_t,h_{t-1})-\sum_{a_t}\pi_t(a_t|o_t,h_{t-1})\psi_t^\top\widehat{\mathbf{P}}^\dagger_{a_t}\sum_{r_t,o_{t+1}}\widehat{\mathbf{P}}_{a_t,r_t,o_{t+1}}\psi_t\big(r_t+\sum_{a_{t+1}}b^\pi_{V,t+1}(a_{t+1},o_{t+1},h_t)\big)$$

$$=\sum_{a_t}\pi_t(a_t|o_t,h_{t-1})\psi_t^\top\widehat{\mathbf{P}}^\dagger_{a_t}\sum_{r_t,o_{t+1}}\widehat{\mathbf{P}}_{a_t,r_t,o_{t+1}}\psi_t\bigg[\sum_{a_t}b^\pi_{V,t}(a_t,o_t,h_{t-1})-\big(r_t+\sum_{a_{t+1}}b^\pi_{V,t+1}(a_{t+1},o_{t+1},h_t)\big)\bigg].$$

Thus, it suffices to bound $\mathcal{E}(E_1)$ and $\mathcal{E}(E_2)$ separately.

### C.1.1 Bounding $\mathcal{E}(E_1)$

Note that $\mathcal{E}(E_1):=\sum_{t=1}^T\mathcal{E}(I_t)$ as the sum of $T$ terms where

$$\mathcal{E}(I_1)=\mathcal{E}\bigg\{\sum_{a_1}\frac{\pi_1(a_1|O_1)}{\pi_1^b(a_1|S_1)}\pi_1^b(a_1|S_1)\psi_1^\top\mathbf{P}^\dagger_{a_1}\sum_{r_1,o_2}\widehat{\mathbf{P}}_{a_1,r_1,o_2}\psi_1\bigg[\sum_{a_1}b^\pi_{V,1}(a_1,O_1)-\Big(r_1+\sum_{a_2}b^\pi_{V,2}\big(a_2,o_2,(O_1,a_1)\big)\Big)\bigg]\bigg\},$$

$$=\mathcal{E}\left\{\frac{\pi_1(A_1|O_1)}{\pi_1^b(A_1|S_1)}\psi_1^\top \mathbf{P}_{A_1}^\dagger \sum_{r_1,o_2}\widehat{\mathbf{P}}_{A_1,r_1,o_2}\psi_1\left[\sum_{a_1}b_{V,1}^\pi(a_1,O_1)-\left(r_1+\sum_{a_2}b_{V,2}^\pi(a_2,o_2,H_1)\right)\right]\right\},$$

$$\mathcal{E}(I_2)=\mathcal{E}\left\{\sum_{a_1}\frac{\pi_1(a_1|O_1)}{\pi_1^b(a_1|S_1)}\pi_1^b(a_1|S_1)\psi_1^\top\mathbf{P}_{a_1}^\dagger\sum_{r_1,o_2}\mathbf{P}_{a_1,r_1,o_2}\psi_1\left(\sum_{a_2}\frac{\pi_2(a_2|o_2,O_1,a_1)}{\pi_2^b(a_2|S_2)}\pi_2^b(a_2|S_2)\right.\right.$$

$$\left.\left.\psi_2^\top\mathbf{P}_{a_2}^\dagger\sum_{r_2,o_3}\widehat{\mathbf{P}}_{a_2,r_2,o_3}\psi_2\left[\sum_{a_2}b_{V,2}^\pi\big(a_2,o_2,(O_1,a_1)\big)-\left(r_2+\sum_{a_2}b_{V,3}^\pi\big(a_3,o_3,(O_1,a_1,o_2,a_2)\big)\right)\right]\right)\right\}$$

$$=\mathcal{E}\left\{\frac{\pi_1}{\pi_1^b}\psi_1^\top\mathbf{P}_{A_1}^\dagger\mathbf{P}_{A_1}\psi_1\left(\frac{\pi_2}{\pi_2^b}\psi_2^\top\mathbf{P}_{A_2}^\dagger\sum_{r_2,o_3}\widehat{\mathbf{P}}_{A_2,r_2,o_3}\psi_2\left[\sum_{a_2}b_{V,2}^\pi\big(a_2,O_2,H_1\big)-\left(r_2+\sum_{a_2}b_{V,3}^\pi\big(a_3,o_3,H_2\big)\right)\right]\right)\right\}$$

$$=\mathcal{E}\left\{\frac{\pi_1(A_1|O_1)\pi_2(A_2|O_2,H_1)}{\pi_1^b(A_1|S_1)\pi_2^b(A_2|S_2)}\psi_2^\top\mathbf{P}_{A_2}^\dagger\sum_{r_2,o_3}\widehat{\mathbf{P}}_{A_2,r_2,o_3}\psi_2\left[\sum_{a_2}b_{V,2}^\pi\big(a_2,O_2,H_1\big)-\left(r_2+\sum_{a_3}b_{V,3}^\pi\big(a_3,o_3,H_2\big)\right)\right]\right\},$$

$$\cdots$$

$$\mathcal{E}(I_T)=\mathcal{E}\left\{\left(\prod_{t=1}^T\frac{\pi_t}{\pi_t^b}\right)\psi_T^\top\mathbf{P}_{A_T}^\dagger\sum_{r_T}\widehat{\mathbf{P}}_{A_T,r_T}\psi_T\,r_T\right\}.$$

Here, we simplify the expression by rewriting the summation over $a_t, o_t$ as the expectation of $A_t, O_t$, and use the fact that $A_t \perp\!\!\!\perp O_t \mid S_t$.

First, we need to verify that the expectation $\mathbb{E}[\mathcal{E}(E_1)]=0$.

For $\mathcal{E}(I_1)$, it suffices to show $\mathbb{E}[\widehat{\mathbf{P}}_{a_1,r_1,o_2}]=\mathbf{P}_{a_1,r_1,o_2}$. Note that for each entry of $\widehat{\mathbf{P}}_{a_1,r_1,o_2}$, the empirical transition probability is an unbiased estimator of its population transition probability (Lemma 12).

$$\mathbb{E}\left[\frac{\sum_{i=1}^n\mathbb{1}\{(o_2^i,o_1^i,r_1^i,o_0^i,a_1^i)=(o_2,o_1,r_1,o_0,a_1)\}}{\sum_{i=1}^n\mathbb{1}\{(o_0^i,a_1^i)=(o_0,a_1)\}}\right]$$

$$=\mathbb{E}\left[\mathbb{E}\left[\frac{\sum_{i=1}^n\mathbb{1}\{(o_2^i,o_1^i,r_1^i,o_0^i,a_1^i)=(o_2,o_1,r_1,o_0,a_1)\}}{\sum_{i=1}^n\mathbb{1}\{(o_0^i,a_1^i)=(o_0,a_1)\}}\ \Big|\ \{o_0^{(i)},a_1^{(i)}\}_{i=1}^n\right]\right]$$

$$=\mathbb{E}\left[\frac{n_{o_0,a_1}p(o_2,o_1,r_1\mid o_0,a_1)}{n_{o_0,a_1}}\ \Big|\ \{o_0^{(i)},a_1^{(i)}\}_{i=1}^n\right]$$

$$=p(o_2,o_1,r_1\mid o_0,a_1).$$

Thus, $\mathbb{E}[\widehat{\mathbf{P}}_{a_1,r_1,o_2}]=\mathbf{P}_{a_1,r_1,o_2}$, which yields

$$\mathbb{E}\big[\mathcal{E}(I_1)\big]$$

$$=\mathbb{E}\left\{\mathcal{E}\left(\frac{\pi_1}{\pi_1^b}\psi_1^\top\mathbf{P}_{A_1}^\dagger\sum_{r_1,o_2}\widehat{\mathbf{P}}_{A_1,r_1,o_2}\psi_1\left[\sum_{a_1}b_{V,1}^\pi(a_1,O_1)-\left(r_1+\sum_{a_2}b_{V,2}^\pi\big(a_2,o_2,H_1\big)\right)\right]\right)\right\}$$

$$=\mathbb{E}\left\{\frac{\pi_1}{\pi_1^b}\psi_1^\top\mathbf{P}_{A_1}^\dagger\sum_{r_1,o_2}\mathbf{P}_{A_1,r_1,o_2}\psi_1\left[\sum_{a_1}b_{V,1}^\pi(a_1,O_1)-\left(r_1+\sum_{a_2}b_{V,2}^\pi\big(a_2,o_2,H_1\big)\right)\right]\right\}$$

$$=\mathbb{E}\left\{\sum_{a_1}\pi_1(a_1|O_1)\psi_1^\top\mathbf{P}_{a_1}^\dagger\sum_{r_1,o_2}\mathbf{P}_{a_1,r_1,o_2}\psi_1\left[\sum_{a_1}b_{V,1}^\pi(a_1,O_1)-\left(r_1+\sum_{a_2}b_{V,2}^\pi\big(a_2,o_2,a_1,O_1\big)\right)\right]\right\}$$

$$=\mathbb{E}\left\{\sum_{a_1}b_{V,1}^\pi(a_1,O_1)-\sum_{a_1}b_{V,1}^\pi(a_1,O_1)\right\}=0.$$

For $\mathcal{E}(I_t), t>1$, we can show the similar results that $\mathbb{E}[\widehat{\mathbf{P}}_{a_t,r_t,o_{t+1}}]=\mathbf{P}_{a_t,r_t,o_{t+1}}$. Thus, $\mathbb{E}[\mathcal{E}(E_1)]=0$.

Then, it suffices to derive the bound for the variance of $\mathcal{E}(E_1)$. By applying the law of total variance (Lemma 3), we have

$$\mathrm{Var}\Big[\mathcal{E}(E_1)\Big]$$

$$=\mathrm{Var}\left\{\sum_{t=1}^{T}\mathcal{E}\left(\prod_{t'=1}^{t}\frac{\pi_{t'}}{\pi_{t'}^{b}}\psi_{t}^{\top}\mathbf{P}_{A_t}^{\dagger}\sum_{r_t,o_{t+1}}\widehat{\mathbf{P}}_{A_t,r_t,o_{t+1}}\psi_t\Big[\sum_{a_t}b_{V,t}^{\pi}-\big(r_t+\sum_{a_{t+1}}b_{V,t+1}^{\pi}\big)\Big]\right)\right\}$$

$$=\sum_{t=1}^{T}\mathbb{E}\left\{\mathrm{Var}\left[\mathcal{E}\left(\prod_{t'=1}^{t}\frac{\pi_{t'}}{\pi_{t'}^{b}}\psi_{t}^{\top}\mathbf{P}_{A_t}^{\dagger}\sum_{r_t,o_{t+1}}\widehat{\mathbf{P}}_{A_t,r_t,o_{t+1}}\psi_t\Big[\sum_{a_t}b_{V,t}^{\pi}-\big(r_t+\sum_{a_{t+1}}b_{V,t+1}^{\pi}\big)\Big]\right)\,\Big|\,\mathcal{D}_t\right]\right\}.$$

Next we consider bounding $\mathrm{Var}\big[\mathcal{E}(E_1)\big]$ under different conditions.

**(1) Bounding $\mathrm{Var}\big[\mathcal{E}(E_1)\big]$ without considering weight function**  Note that

$$\mathrm{Var}\left[\mathcal{E}(E_1)\right]$$

$$=\sum_{t=1}^{T}\mathbb{E}\left\{\mathrm{Var}\left[\mathcal{E}\left(\prod_{t'=1}^{t}\frac{\pi_{t'}}{\pi_{t'}^{b}}\psi_{t}^{\top}\mathbf{P}_{A_t}^{\dagger}\sum_{r_t,o_{t+1}}\widehat{\mathbf{P}}_{A_t,r_t,o_{t+1}}\psi_t\Big[\sum_{a_t}b_{V,t}^{\pi}-\big(r_t+\sum_{a_{t+1}}b_{V,t+1}^{\pi}\big)\Big]\right)\,\Big|\,\mathcal{D}_t\right]\right\}$$

$$\lesssim T^2\sum_{t=1}^{T}\mathbb{E}\left\{\mathrm{Var}\left[\mathcal{E}\left(\prod_{t'=1}^{t}\frac{\pi_{t'}}{\pi_{t'}^{b}}\psi_{t}^{\top}\mathbf{P}_{A_t}^{\dagger}\sum_{r_t,o_{t+1}}\widehat{\mathbf{P}}_{A_t,r_t,o_{t+1}}\psi_t\right)\,\Big|\,\mathcal{D}_t\right]\right\}$$

$$=T^2\sum_{t=1}^{T}\mathbb{E}\left\{\mathcal{E}\left(\prod_{t'=1}^{t}\Big(\frac{\pi_{t'}}{\pi_{t'}^{b}}\Big)^2\mathrm{Var}\left[\psi_{t}^{\top}\mathbf{P}_{A_t}^{\dagger}\sum_{r_t,o_{t+1}}\widehat{\mathbf{P}}_{A_t,r_t,o_{t+1}}\psi_t\,\Big|\,\mathcal{D}_t\right]\right)\right\}$$

$$\overset{(i)}{\leq}T^2\sum_{t=1}^{T}\mathbb{E}\left\{\mathcal{E}\left(\prod_{t'=1}^{t}\Big(\frac{\pi_{t'}}{\pi_{t'}^{b}}\Big)^2\psi_{t}^{\top}\mathbf{P}_{A_t}^{\dagger}\sum_{r_t,o_{t+1}}\mathrm{Cov}\left[\widehat{\mathbf{P}}_{A_t,r_t,o_{t+1}}\psi_t\,\Big|\,\mathcal{D}_t\right](\mathbf{P}_{A_t}^{\dagger})^{\top}\psi_t\right)\right\}$$

$$=T^2\sum_{t=1}^{T}\mathbb{E}\left\{\mathcal{E}\left(\prod_{t'=1}^{t}\Big(\frac{\pi_{t'}}{\pi_{t'}^{b}}\Big)^2\psi_{t}^{\top}\mathbf{P}_{A_t}^{\dagger}\sum_{r_t,o_{t+1}}\mathrm{diag}(\mathbf{1}\{E_{o_0,h_{t-1}}\})\mathrm{Cov}\left[\widehat{\mathbf{P}}_{A_t,r_t,o_{t+1}}\psi_t\,\Big|\,\mathcal{D}_t\right](\mathbf{P}_{A_t}^{\dagger})^{\top}\psi_t\right)\right\}$$

$$\overset{(ii)}{\leq}T^2\sum_{t=1}^{T}\mathcal{E}\left(\prod_{t'=1}^{t}\Big(\frac{\pi_{t'}}{\pi_{t'}^{b}}\Big)^2\Big|\psi_{t}^{\top}\mathbf{P}_{A_t}^{\dagger}\mathbb{E}\Big[\mathrm{diag}(\mathbf{1}\{E_{o_0,h_{t-1}}\}/n_{o_0,A_t,h_{t-1}})\mathrm{diag}(\mathbf{P}_{A_t}\psi_t)\Big](\mathbf{P}_{A_t}^{\dagger})^{\top}\psi_t\Big|\right)\right\}$$

$$\leq\frac{T^2}{n}(1-\theta)^{-1}\sum_{t=1}^{T}\mathbb{E}\left\{\prod_{t'=1}^{t}\Big(\frac{\pi_{t'}}{\pi_{t'}^{b}}\Big)^2\Big|\psi_{t}^{\top}\mathbf{P}_{A_t}^{\dagger}\mathrm{diag}(\mathbf{P}_{A_t}\psi_t)(\mathbf{P}_{A_t}^{\dagger})^{\top}\psi_t\Big|\right\}$$

$$\leq\frac{T^2}{n}(1-\theta^*)^{-1}C_{\pi^b}\sum_{t=1}^{T}|\mathcal{A}|\|\psi_t\|_2^2\,\|\mathbf{P}_{a_t}^{\dagger}\|_2^2\,\|\mathrm{diag}(\mathbf{P}_{a_t}\psi_t)\|_2$$

$$\overset{(iii)}{\leq}\frac{T^2}{n}(1-\theta^*)^{-1}C_{\pi^b}C_P^2\sum_{t=1}^{T}|\mathcal{A}||\mathcal{O}||\mathcal{H}_{t-1}|$$

$$=\frac{T^2}{n}(1-\theta^*)^{-1}C_P^2 C_{\pi^b}\sum_{t=1}^{T}|\mathcal{O}|^t\,|\mathcal{A}|^t$$

$$\leq\frac{T^3}{n}(1-\theta^*)^{-1}C_P^2 C_{\pi^b}|\mathcal{O}|^T\,|\mathcal{A}|^T.$$

The inequality $(i)$ follows from the independence of each $\widehat{\mathbf{P}}_{A_t,r_t,o_{t+1}}$, since $\{r_t^{(i)},o_{t+1}^{(i)}\}_{i=1}^{n}$ partitions the $n$ episodes into disjoint sets according to all combinations of $r_t$ and $o_{t+1}$. We define $E_{o_0,h_{t-1}}=\{n_{o_0,A_t,h_{t-1}}>np_t^{\pi^b}(1-\theta_{t,ij})\}$, and $\mathbf{1}\{E_{o_0,h_{t-1}}\}$ is a vector of length $|\mathcal{O}||\mathcal{H}_{t-1}|$, where each element corresponds to $E_{o_0,h_{t-1}}$. The inequality $(ii)$ utilizes the fact that for any $a_t\in\mathcal{A}$

$$\mathrm{Cov}\left[\widehat{\mathbf{P}}_{a_t,r_t,o_{t+1}}\psi_t\mid\mathcal{D}_t\right]$$
$$\leq\mathrm{diag}(\mathbf{P}_{a_t,r_t,o_{t+1}}\psi_t/n_{o_0,a_t,h_{t-1}})-\mathbf{P}_{a_t,r_t,o_{t+1}}\psi_t\psi_t^{\top}\mathbf{P}_{a_t,r_t,o_{t+1}}^{\top}/n_{o_0,a_t,h_{t-1}} \tag{22}$$
$$\preceq\mathrm{diag}(\mathbf{P}_{a_t,r_t,o_{t+1}}\psi_t/n_{o_0,a_t,h_{t-1}}).$$

The inequality $(iii)$ follows from that

$$\left\|\text{diag}(\mathbf{P}_{a_t}\psi_t)\right\|_2 \le 1, \text{ and } \left\|\mathbf{P}^\dagger_{a_t}\right\|_2 \le \frac{1}{\sigma_{min}(\mathbf{P}_{a_t})} \le C_P\sqrt{|\mathcal{O}||\mathcal{H}_{t-1}|}.$$

Therefore, we obtain the upper bound of $\mathcal{E}(E_1)$

$$\mathcal{E}(E_1) = \mathcal{O}_P\left(\frac{T^{1.5}}{\sqrt{n}}(1-\theta^*)^{-1/2}C_P C_{\pi^b}^{\frac{1}{2}}|\mathcal{O}|^{\frac{T}{2}}|\mathcal{A}|^{\frac{T}{2}}\right)$$

**(2) Bounding $\mathcal{E}(E_1)$ with the weight function**

$$\text{Var}\left[\mathcal{E}(E_1)\right]$$

$$=\sum_{t=1}^{T}\mathbb{E}\left\{\text{Var}\left[\mathcal{E}\left(\prod_{t'=1}^{t}\frac{\pi_{t'}}{\pi^b_{t'}}\psi_t^\top\mathbf{P}^\dagger_{A_t}\sum_{r_t,o_{t+1}}\widehat{\mathbf{P}}_{A_t,r_t,o_{t+1}}\psi_t\left[\sum_{a_t}b^\pi_{V,t}-\left(r_t+\sum_{a_{t+1}}b^\pi_{V,t+1}\right)\right]\right)\Big|\mathcal{D}_t\right]\right\}$$

$$\le\frac{C_W C_P^2(1-\theta^*)^{-1}}{n}\sum_{t=1}^{T}|\mathcal{O}||\mathcal{H}_{t-1}||\mathcal{A}|$$

$$\mathbb{E}^\pi\left\{\text{Var}\left[\sum_{a_t}b^\pi_{V,t}(a_t,O_t,H_{t-1})-\left(R_t+\sum_{a_{t+1}}b^\pi_{V,t+1}(a_{t+1},O_{t+1},H_t)\right)\Big|O_t,S_t,A_t,H_{t-1}\right]\right\}$$

$$\quad\text{(by Equation (24))}$$

$$=\frac{C_W C_P^2(1-\theta^*)^{-1}}{n}|\mathcal{O}|^T|\mathcal{A}|^T\sum_{t=1}^{T}\mathbb{E}^\pi\left\{\text{Var}\left[R_t+\sum_{a}b^\pi_{V,t+1}(a,O_{t+1},H_t)\Big|O_t,S_t,A_t,H_{t-1}\right]\right\}$$

$$\le\frac{C_W C_P^2(1-\theta^*)^{-1}}{n}|\mathcal{O}|^T|\mathcal{A}|^T\text{Var}^\pi\left[\sum_{t=1}^{T}R_t\right]\quad\text{(by Lemma 1)}$$

$$\le\frac{T^2}{n}C_W C_P^2(1-\theta^*)^{-1}|\mathcal{O}|^T|\mathcal{A}|^T.$$

For the first inequality, we examine the cases for $t=1,2$ separately in the following. The result for $t>2$ follows by a similar argument. Thus, incorporating the ratio function, we obtain

$$\mathcal{E}(E_1) = \mathcal{O}_P\left(\frac{T}{\sqrt{n}}C_W^{\frac{1}{2}}C_P(1-\theta^*)^{-\frac{1}{2}}|\mathcal{O}|^{\frac{T}{2}}|\mathcal{A}|^{\frac{T}{2}}\right)$$

- For $t=1$, we have

$$\mathbb{E}\left\{\text{Var}\left[\mathcal{E}\left(\frac{\pi_1(A_1|O_1)}{\pi^b_1(A_1|S_1)}\psi_1^\top\mathbf{P}^\dagger_{A_1}\sum_{r_1,o_2}\widehat{\mathbf{P}}_{A_1,r_1,o_2}\psi_1\left[\sum_{a_1}b^\pi_{V,1}-\left(r_1+\sum_{a_2}b^\pi_{V,2}\right)\right]\right)\Big|\mathcal{D}_1\right]\right\}$$

$$=\mathbb{E}\left\{\text{Var}\left[\mathcal{E}\left(\mathcal{E}\left(b^\pi_{W,1}(A_1,O_0)\Big|A_1,S_1\right)\pi_1(A_1|O_1)\right.\right.\right.$$

$$\left.\left.\left.\psi_1^\top\mathbf{P}^\dagger_{A_1}\sum_{r_1,o_2}\widehat{\mathbf{P}}_{A_1,r_1,o_2}\psi_1\left[\sum_{a_1}b^\pi_{V,1}(a_1,O_1)-\left(r_1+\sum_{a_2}b^\pi_{V,2}(a_2,o_2,H_1)\right)\right]\right)\Big|\mathcal{D}_1\right]\right\}$$

$$=\mathbb{E}\left\{\text{Var}\left[\mathcal{E}\left(b^\pi_{W,1}(A_1,O_0)\pi_1(A_1|O_1)\right.\right.\right.$$

$$\left.\left.\left.\psi_1^\top\mathbf{P}^\dagger_{A_1}\sum_{r_1,o_2}\widehat{\mathbf{P}}_{A_1,r_1,o_2}\psi_1\left[\sum_{a_1}b^\pi_{V,1}(a_1,O_1)-\left(r_1+\sum_{a_2}b^\pi_{V,2}(a_2,o_2,H_1)\right)\right]\right)\Big|\mathcal{D}_1\right]\right\}$$

$$(\text{by } O_0 \perp\!\!\!\perp O_1 \mid S_1, A_1)$$

$$\leq \mathbb{E}\Bigg\{ \mathcal{E}\Bigg( \Big( b_{W,1}^\pi(A_1, O_0)\pi_1(A_1|O_1) \Big)^2$$

$$\psi_1^\top \mathbf{P}_{A_1}^\dagger \sum_{r_1, o_2} \text{Cov}\Big[ \widehat{\mathbf{P}}_{A_1, r_1, o_2}\psi_1 \,\Big|\, \mathcal{D}_1 \Big]\Big[ \sum_{a_1} b_{V,1}^\pi(a_1, O_1) - \Big( r_1 + \sum_{a_2} b_{V,2}^\pi(a_2, o_2, H_1) \Big) \Big]^2 (\mathbf{P}_{A_1}^\dagger)^\top \psi_1 \Bigg) \Bigg\}$$

$$\overset{(i)}{\leq} \mathcal{E}\Bigg\{ \Big( b_{W,1}^\pi(A_1, O_0)\pi_1(A_1|O_1) \Big)^2$$

$$\psi_1^\top \mathbf{P}_{A_1}^\dagger \mathbb{E}\Big[ \text{diag}\Big(\frac{\mathbf{1}\{E_{o_0, h_{t-1}}\}}{n_{o_0, A_t, h_{t-1}}}\Big)\text{diag}\big(\mathbf{P}_{A_1}\psi_1\big) \Big](\mathbf{P}_{aA_1}^\dagger)^\top \psi_1 \Big[ \sum_{a_1} b_{V,1}^\pi(a_1, O_1) - \Big( R_1 + \sum_{a_2} b_{V,2}^\pi(a_2, O_2, H_1) \Big) \Big]^2 \Bigg\}$$

$$\leq \frac{(1-\theta^*)^{-1}}{n}|\mathcal{A}|\, \mathbb{E}\Bigg\{ \Big( b_{W,1}^\pi(A_1, O_0)\pi_1(A_1|O_1) \Big)^2$$

$$\psi_1^\top \mathbf{P}_{a_1}^\dagger \text{diag}\big(\mathbf{P}_{a_1}\psi_1\big)(\mathbf{P}_{a_1}^\dagger)^\top \psi_1 \Big[ \sum_{a_1} b_{V,1}^\pi(a_1, O_1) - \Big( R_1 + \sum_{a_2} b_{V,2}^\pi(a_2, O_2, H_1) \Big) \Big]^2 \Bigg\}$$

$$\overset{(ii)}{\leq} \frac{C_P^2(1-\theta^*)^{-1}}{n}|\mathcal{O}||\mathcal{A}|$$

$$\mathbb{E}\Bigg\{ b_{W,1}^\pi(A_1, O_0)\pi_1(A_1|O_1)\Big[ \sum_{a_1} b_{V,1}^\pi(a_1, O_1) - \Big( R_1 + \sum_{a_2} b_{V,2}^\pi(a_2, O_2, H_1) \Big) \Big] \Bigg\}^2$$

$$= \frac{C_P^2(1-\theta^*)^{-1}}{n}|\mathcal{O}||\mathcal{A}|$$

$$\mathbb{E}\Bigg\{ \text{Var}\Big[ b_{W,1}^\pi(A_1, O_0)\pi_1(A_1|O_1)\Big[ \sum_{a_1} b_{V,1}^\pi(a_1, O_1) - \Big( R_1 + \sum_{a_2} b_{V,2}^\pi(a_2, O_2, H_1) \Big) \Big] \,\Big|\, O_1, S_1, A_1 \Big] \Bigg\}$$

The inequality $(i)$ holds by applying the same argument as in the previous part that bounding $\text{Var}[\mathcal{E}(E_1)]$ without using the weight function. The inequality $(ii)$ holds since that, for any $t \in [T]$,

$$\Big| \psi_t^\top \mathbf{P}_{a_t}^\dagger \, \text{diag}\big(\mathbf{P}_{a_t}\psi_t\big) \, (\mathbf{P}_{a_t}^\dagger)^\top \psi_t \Big| \leq \|\psi_t\|_2^2 \|\mathbf{P}_{a_t}^\dagger\|_2^2 \|\text{diag}\big(\mathbf{P}_{a_t}\psi_t\big)\|_2 \tag{23}$$
$$\leq C_P^2 |\mathcal{O}||\mathcal{H}_{t-1}|.$$

The last equality follows from the zero mean property that

$$\mathbb{E}\Bigg\{ b_{W,t}^\pi(A_t, O_0, H_{t-1})\pi_t(A_t|O_t, H_{t-1})\Big[ \sum_{a_t} b_{V,1}^\pi(a_t, O_t, H_{t-1}) - \Big( R_t + \sum_{a_{t+1}} b_{V,t+1}^\pi(a_{t+1}, O_{t+1}, H_t) \Big) \Big] \Bigg\}$$

$$= \mathbb{E}\Bigg\{ b_{W,t}^\pi(A_t, O_0, H_{t-1})\mathbb{E}\Big( \pi_t(A_t|O_t, H_{t-1})$$

$$\Big[ \sum_{a_t} b_{V,1}^\pi(a_t, O_t, H_{t-1}) - \Big( R_t + \sum_{a_{t+1}} b_{V,t+1}^\pi(a_{t+1}, O_{t+1}, H_t) \Big) \Big] \,\Big|\, O_0, A_t, H_{t-1} \Big) \Bigg\}$$

$$= \mathbb{E}\Bigg\{ b_{W,t}^\pi(A_t, O_0, H_{t-1})\mathbb{E}\Big( \pi_t(A_t|O_t, H_{t-1})\sum_{a_t} b_{V,1}^\pi(a_t, O_t, H_{t-1}) - b_{V,1}^\pi(A_t, O_t, H_{t-1}) \,\Big|\, O_0, A_t, H_{t-1} \Big) \Bigg\}$$

$$(\text{by Assumption 4})$$

$$= \mathbb{E}\Bigg\{ b_{W,t}^\pi(A_t, O_0, H_{t-1})\Big( \pi_t(A_t|O_t, H_{t-1})\sum_{a_t} b_{V,1}^\pi(a_t, O_t, H_{t-1}) - b_{V,1}^\pi(A_t, O_t, H_{t-1}) \Big) \Bigg\}$$

$$= \mathbb{E}\Bigg\{ \frac{\omega_t(S_t, H_{t-1})}{\pi_t^b(A_t|S_t)}\mathbb{E}\Big\{ \sum_{a_t} b_{V,1}^\pi(a_t, O_t, H_{t-1})\pi_t(A_t|O_t, H_{t-1}) - b_{V,1}^\pi(A_t, O_t, H_{t-1}) \,\Big|\, S_t, A_t, O_t, H_{t-1} \Big\} \Bigg\}$$

(by Assumption 3)

$$=\mathbb{E}\Bigg\{\sum_a \pi_t^b(a|S_t)\frac{\omega_t(S_t,H_{t-1})}{\pi_t^b(a|S_t)}\mathbb{E}\Bigg\{\sum_{a_t} b_{V,1}^\pi(a_t,O_t,H_{t-1})\pi_t(a|O_t,H_{t-1})-b_{V,1}^\pi(a,O_t,H_{t-1})$$

$$\Big| S_t, A_t=a, O_t, H_{t-1}\Bigg\}\Bigg\}$$

$$=\mathbb{E}\Bigg\{\omega_t(S_t,H_{t-1})\mathbb{E}\Bigg\{\sum_{a_t}b_{V,1}^\pi(a_t,O_t,H_{t-1})-\sum_a b_{V,1}^\pi(a,O_t,H_{t-1})\Big| S_t,A_t=a,O_t,H_{t-1}\Bigg\}\Bigg\}$$

$$=0.$$

For the last equation, we further have

$$\mathbb{E}\Bigg\{\operatorname{Var}\Bigg[b_{W,1}^\pi(A_1,O_0)\pi_1(A_1|O_1)\Big[\sum_{a_1}b_{V,1}^\pi(a_1,O_1)-\Big(R_1+\sum_{a_2}b_{V,2}^\pi(a_2,O_2,H_1)\Big)\Big]\Big| O_1,S_1,A_1\Bigg]\Bigg\}$$

$$=\mathbb{E}\Bigg\{\operatorname{Var}\Bigg[\mathbb{E}\Big(b_{W,1}^\pi(A_1,O_0)\pi_1(A_1|O_1)\Big| O_1,S_1,A_1\Big)$$

$$\Big[\sum_{a_1}b_{V,1}^\pi(a_1,O_1)-\Big(R_1+\sum_{a_2}b_{V,2}^\pi(a_2,O_2,H_1)\Big)\Big]\Big| O_1,S_1,A_1\Bigg]\Bigg\}\quad \text{(by } O_0 \perp\!\!\!\perp O_1 \mid S_1,A_1)$$

$$\overset{(i)}{\le}C_W\,\mathbb{E}\Bigg\{\mathbb{E}\Big(b_{W,1}^\pi(A_1,O_0)\pi_1(A_1|O_1)\Big| O_1,S_1,A_1\Big)$$

$$\operatorname{Var}\Bigg[\sum_{a_1}b_{V,1}^\pi(a_1,O_1)-\Big(R_1+\sum_{a_2}b_{V,2}^\pi(a_2,O_2,H_1)\Big)\Big| O_1,S_1,A_1\Bigg]\Bigg\}\quad \text{(by Assumption 3 )}$$

$$=C_W\,\mathbb{E}\Bigg\{\frac{p_1^\pi(S_1)\pi_1(A_1|O_1)p(O_1|S_1)}{p_1^{\pi^b}(S_1)\pi_1^b(A_1|S_1)p(O_1|S_1)}\operatorname{Var}\Bigg[\sum_{a_1}b_{V,1}^\pi(a_1,O_1)-\Big(R_1+\sum_{a_2}b_{V,2}^\pi(a_2,O_2,H_1)\Big)\Big| O_1,S_1,A_1\Bigg]\Bigg\}$$

$$\Bigg(\text{by }\mathbb{E}\to\mathbb{E}^\pi:\quad \frac{p_t^\pi(a_t,o_t,s_t,h_{t-1})}{p_t^{\pi^b}(a_t,o_t,s_t,h_{t-1})}=\frac{p_t^\pi(s_t,h_{t-1})\pi_t(a_t|o_t,h_{t-1})p(o_t|s_t)}{p_t^{\pi^b}(s_t,h_{t-1})\pi_t^b(a_t|s_t)p(o_t|s_t)}\Bigg)$$

$$=C_W\,\mathbb{E}^\pi\Bigg\{\operatorname{Var}\Bigg[\sum_{a_1}b_{V,1}^\pi(a_1,O_1)-\Big(R_1+\sum_{a_2}b_{V,2}^\pi(a_2,O_2,H_1)\Big)\Big| O_1,S_1,A_1\Bigg]\Bigg\}$$

where the inequality $(i)$ follows from the Assumption 3, which states the ratio function satisfies

$$\sup_{t,o_t,s_t,a_t,h_{t-1}}\mathbb{E}\Big[b_{W,t}^\pi(a_t,h_{t-1},O_0)\pi_t(a_t\mid o_t,h_{t-1})\mid o_t,s_t,a_t,h_{t-1}\Big]\le C_W.$$

Thus, we have

$$\mathbb{E}\Bigg\{\operatorname{Var}\Bigg[\mathcal{E}\Big(\frac{\pi_1(A_1|O_1)}{\pi_1^b(A_1|S_1)}\psi_1^\top\mathbf{P}_{A_1}^\dagger\sum_{r_1,o_2}\widehat{\mathbf{P}}_{A_1,r_1,o_2}\psi_1\Big[\sum_{a_1}b_{V,1}^\pi(a_1,O_1)-\Big(r_1+\sum_{a_2}b_{V,2}^\pi(a_2,o_2,H_1)\Big)\Big]\Big)\Big| \mathcal{D}_1\Bigg]\Bigg\}$$

$$\le\frac{C_W C_P^2(1-\theta^*)^{-1}}{n}|\mathcal{O}||\mathcal{A}|\;\mathbb{E}^\pi\Bigg\{\operatorname{Var}\Bigg[\sum_{a_1}b_{V,1}^\pi(a_1,O_1)-\Big(R_1+\sum_{a_2}b_{V,2}^\pi(a_2,O_2,H_1)\Big)\Big| O_1,S_1,A_1\Bigg]\Bigg\}.$$

- For $t=2$, by applying the similar arguments, we have

$$\mathbb{E}\Bigg\{\operatorname{Var}\Bigg[\mathcal{E}\Big(\frac{\pi_1}{\pi_1^b}\frac{\pi_2}{\pi_2^b}\psi_2^\top\mathbf{P}_{A_2}^\dagger\sum_{r_2,o_3}\widehat{\mathbf{P}}_{A_2,r_2,o_3}\psi_2\Big[\sum_{a_2}b_{V,2}^\pi(a_2,O_2,H_1)-\Big(r_2+\sum_{a_3}b_{V,2}^\pi(a_3,o_3,H_2)\Big)\Big]\Big)\Big| \mathcal{D}_2\Bigg]\Bigg\}$$

$$= \mathbb{E}\Bigg\{ \mathrm{Var}\Bigg[ \mathcal{E}\Bigg( \frac{p_1^\pi(S_1)\pi_1(A_1|O_1)}{p_1^{\pi^b}(S_1)\pi_1^b(A_1|S_1)} \sum_{a_2} \pi_2(a_2|O_2,H_1)\psi_2^\top \mathbf{P}_{a_2}^\dagger \sum_{r_2,o_3} \widehat{\mathbf{P}}_{a_2,r_2,o_3}\psi_2$$

$$\Bigg[ \sum_{a_2} b_{V,2}^\pi(a_2,O_2,H_1) - \Big(r_2 + \sum_{a_3} b_{V,2}^\pi(a_3,o_3,(O_2,a_2,H_1))\Big)\Bigg]\Bigg) \Big| \mathcal{D}_2\Bigg]\Bigg\}$$

$$= \mathbb{E}\Bigg\{ \mathrm{Var}\Bigg[ \mathcal{E}\Bigg( \frac{p_2^\pi(S_2,H_1)\pi_2(A_2|O_2,H_1)}{p_2^{\pi^b}(S_2,H_1)\pi_2^b(A_2|S_2)} \psi_2^\top \mathbf{P}_{A_2}^\dagger \sum_{r_2,o_3} \widehat{\mathbf{P}}_{A_2,r_2,o_3}\psi_2$$

$$\Bigg[ \sum_{a_2} b_{V,2}^\pi(a_2,o_2,H_1) - \Big(r_2 + \sum_{a_3} b_{V,2}^\pi(a_3,o_3,(O_2,A_2,H_1))\Big)\Bigg]\Bigg) \Big| \mathcal{D}_2\Bigg]\Bigg\} \qquad \text{(by Lemma 4)}$$

$$= \mathbb{E}\Bigg\{ \mathrm{Var}\Bigg[ \mathcal{E}\Bigg( b_{W,2}^\pi(A_2,O_0,H_1)\pi_2(A_2|O_2,H_1)\psi_2^\top \mathbf{P}_{A_2}^\dagger \sum_{r_2,o_3} \widehat{\mathbf{P}}_{A_2,r_2,o_3}\psi_2$$

$$\Bigg[ \sum_{a_2} b_{V,2}^\pi(a_2,o_2,H_1) - \Big(r_2 + \sum_{a_3} b_{V,2}^\pi(a_3,o_3,H_2)\Big)\Bigg]\Bigg) \Big| \mathcal{D}_2\Bigg]\Bigg\}$$

$$\overset{(i)}{\le} \frac{C_P^2(1-\theta^*)^{-1}}{n}|\mathcal{O}||\mathcal{H}_1||\mathcal{A}|$$

$$\mathrm{Var}\Bigg\{ b_{W,2}^\pi(A_2,O_0,H_1)\pi_2(A_2|O_2,H_1)\Bigg[ \sum_{a_2} b_{V,2}^\pi(a_2,o_2,H_1) - \Big(R_2 + \sum_{a_3} b_{V,2}^\pi(a_3,O_3,H_2)\Big)\Bigg]\Bigg\}$$

$$= \frac{C_P^2(1-\theta^*)^{-1}}{n}|\mathcal{O}|^2|\mathcal{A}|^2\, \mathbb{E}\Bigg\{ \mathrm{Var}\Bigg[ \mathbb{E}\Big[ b_{W,2}^\pi(A_2,O_0,H_1)\pi_2(A_2|O_2,H_1) \,\Big|\, O_2,S_2,A_2,H_1\Big]$$

$$\Bigg[ \sum_{a_2} b_{V,2}^\pi(a_2,O_2,H_1) - \Big(R_2 + \sum_{a_3} b_{V,2}^\pi(a_3,O_3,H_2)\Big)\Bigg] \Big| O_2,S_2,A_2,H_1\Bigg]\Bigg\}$$

$$\overset{(ii)}{\le} \frac{C_W C_P^2(1-\theta^*)^{-1}}{n}|\mathcal{O}|^2|\mathcal{A}|^2\, \mathbb{E}\Bigg\{ \mathbb{E}\Big[ b_{W,2}^\pi(A_2,O_0,H_1)\pi_2(A_2|O_2,H_1) \,\Big|\, O_2,S_2,A_2,H_1\Big]$$

$$\mathrm{Var}\Bigg[ \sum_{a_2} b_{V,2}^\pi(a_2,O_2,H_1) - \Big(r_2 + \sum_{a_3} b_{V,2}^\pi(a_3,o_3,H_2)\Big) \,\Big|\, O_2,S_2,A_2,H_1\Bigg]\Bigg\}$$

$$= \frac{C_W C_P^2(1-\theta^*)^{-1}}{n}|\mathcal{O}|^2|\mathcal{A}|^2\, \mathbb{E}\Bigg\{ \frac{p_2^\pi(S_2,H_1)\pi_2(A_2\mid O_2,H_1)}{p_2^{\pi^b}(S_2,H_1)\pi_2^b(A_2\mid S_2)}$$

$$\mathrm{Var}\Bigg[ \sum_{a_2} b_{V,2}^\pi(a_2,O_2,H_1) - \Big(R_2 + \sum_{a_3} b_{V,2}^\pi(a_3,O_3,H_2)\Big) \,\Big|\, O_2,S_2,A_2,H_1\Bigg]\Bigg\}$$

$$= \frac{C_W C_P^2(1-\theta^*)^{-1}}{n}|\mathcal{O}|^2|\mathcal{A}|^2\, \mathbb{E}^\pi\Bigg\{ \mathrm{Var}\Bigg[ \sum_{a_2} b_{V,2}^\pi(a_2,O_2,H_1) - \Big(R_2 + \sum_{a_3} b_{V,2}^\pi(a_3,O_3,H_2)\Big) \,\Big|\, O_2,S_2,A_2,H_1\Bigg]\Bigg\}$$

The inequality $(i)$ is derived using the same arguments as in the proof for the case $t = 1$ and equation (23). The inequality $(ii)$ is a consequence of Assumption 3, as utilized in the proof for the case $t = 1$.

Therefore, for any $t \in [T]$, by the same reasoning, we have

$$
\mathbb{E}\left\{ \mathrm{Var}\left[ \mathcal{E}\left( \prod_{t'=1}^{t} \frac{\pi_{t'}}{\pi_{t'}^b} \psi_t^\top \mathbf{P}_{A_t}^\dagger \sum_{r_t, o_{t+1}} \widehat{\mathbf{P}}_{A_t, r_t, o_{t+1}} \psi_t \left[ \sum_{a_t} b_{V,t}^\pi - \left( r_t + \sum_{a_{t+1}} b_{V,t+1}^\pi \right) \right] \right) \Big| \mathcal{D}_t \right] \right\}
$$

$$
\leq \frac{C_W C_P^2 (1 - \theta^*)^{-1}}{n} |\mathcal{O}|^2 |\mathcal{A}|^2
$$

$$
\mathbb{E}^\pi \left\{ \mathrm{Var}\left[ \sum_{a_t} b_{V,t}^\pi(a_t, O_t, H_{t-1}) - \left( R_t + \sum_{a_{t+1}} b_{V,t+1}^\pi(a_{t+1}, O_{t+1}, H_t) \right) \Big| O_t, S_t, A_t, H_{t-1} \right] \right\}
$$

$$
\tag{24}
$$

### C.1.2 Proof of Lemma 1

*Proof.* let's suppress the target policy $\pi$ for simplicity. Denote $\widetilde{\mathcal{D}}_t = \{O_{1:t}, S_{1:t}, A_{1:t}, R_{1:t-1}\}$. First, note that

$$
\mathbb{E}^\pi \left[ R_t + \sum_{a'} b_{V,t+1}^\pi(a', O_{t+1}, H_t) \Big| O_t, S_t, A_t, H_{t-1} \right]
$$

$$
= \sum_a \mathbb{E}\left[ R_t + \sum_{a'} b_{V,t+1}^\pi(a', O_{t+1}, H_t) \Big| O_t, S_t, A_t = a, H_{t-1} \right] \pi_t(a \mid O_t, H_{t-1})
$$

$$
= \sum_a \mathbb{E}\left[ \left( R_t + \sum_{a'} b_{V,t+1}^\pi(a', O_{t+1}, H_t) \right) \pi_t(a \mid O_t, H_{t-1}) \Big| O_t, S_t, A_t = a, H_{t-1} \right]
$$

$$
= \mathbb{E}\left[ \mathbb{E}\left[ \sum_a \mathbb{E}\left[ \left( R_t + \sum_{a'} b_{V,t+1}^\pi(a', O_{t+1}, H_t) \right) \pi_t(a \mid O_t, H_{t-1}) \Big| O_t, S_t, A_t = a, H_{t-1} \right] \Big| S_t, A_t = a, H_{t-1} \right] \right]
$$

$$
= \mathbb{E}\left[ \sum_a \mathbb{E}\left[ \left( R_t + \sum_{a'} b_{V,t+1}^\pi(a', O_{t+1}, H_t) \right) \pi_t(a \mid O_t, H_{t-1}) \Big| S_t, A_t = a, H_{t-1} \right] \right]
$$

(by the law of total expectation)

$$
= \mathbb{E}\left[ \sum_a \mathbb{E}\left[ b_{V,t}^\pi(a, O_t, H_{t-1}) \Big| S_t, A_t = a, H_{t-1} \right] \right] \qquad \text{(by Equation (18))}
$$

$$
= \sum_a b_{V,t}^\pi(a, O_t, H_{t-1})
$$

$$
\tag{25}
$$

Then, by iteratively applying the law of total variance, we have

$$
\mathrm{Var}\left[ \sum_{t=1}^{h} R_t + \sum_a b_{V,h+1}^\pi(a, O_{h+1}, H_h) \right]
$$

$$
= \mathbb{E}\left[ \mathrm{Var}\left[ \sum_{t=1}^{h} R_t + \sum_a b_{V,h+1}^\pi(a, O_{h+1}, H_h) \Big| \widetilde{\mathcal{D}}_h \right] \right] + \mathrm{Var}\left[ \mathbb{E}\left[ \sum_{t=1}^{h} R_t + \sum_a b_{V,h+1}^\pi(a, O_{h+1}, H_h) \Big| \widetilde{\mathcal{D}}_h \right] \right]
$$

$$
= \mathbb{E}\left[ \mathrm{Var}\left[ R_h + \sum_a b_{V,h+1}^\pi(a, O_{h+1}, H_h) \Big| O_h, S_h, A_h, H_{h-1} \right] \right]
$$

$$
+ \mathrm{Var}\left[ \sum_{t=1}^{h-1} R_t + \mathbb{E}\left[ R_h + \sum_a b_{V,h+1}^\pi(a, O_{h+1}, H_h) \Big| O_h, S_h, A_h, H_{h-1} \right] \right]
$$

$$
= \mathbb{E}\left[ \mathrm{Var}\left[ R_h + \sum_a b_{V,h+1}^\pi(a, O_{h+1}, H_h) \Big| S_h, A_h, H_{h-1} \right] \right]
$$

$$
+ \mathrm{Var}\left[ \sum_{t=1}^{h-1} R_t + \sum_a b_{V,h}^\pi(a, O_h, H_{h-1}) \right] \qquad \text{(by Equation (25))}
$$

$$
\begin{aligned}
=&\mathbb{E}\left[\mathrm{Var}\left[R_h + \sum_a b_{V,h+1}^\pi(a, O_{h+1}, H_h) \,\Big|\, O_h, S_h, A_h, H_{h-1}\right]\right] \\
&+ \mathbb{E}\left[\mathrm{Var}\left[\sum_{t=1}^{h-1} R_t + \sum_a b_{V,h}^\pi(a, O_h, H_{h-1}) \,\Big|\, \widetilde{\mathcal{D}}_{h-1}\right]\right] \\
&+ \mathrm{Var}\left[\mathbb{E}\left[\sum_{t=1}^{h-1} R_t + \sum_a b_{V,h}^\pi(a, O_h, H_{h-1}) \,\Big|\, \widetilde{\mathcal{D}}_{h-1}\right]\right] \\
=&\mathbb{E}\left[\mathrm{Var}\left[R_h + \sum_a b_{V,h+1}^\pi(a, O_{h+1}, H_h) \,\Big|\, O_h, S_h, A_h, H_{h-1}\right]\right] \\
&+ \mathbb{E}\left[\mathrm{Var}\left[R_{h-1} + \sum_a b_{V,h}^\pi(a, O_h, H_{h-1}) \,\Big|\, O_{h-1}, S_{h-1}, A_{h-1}, H_{h-2}\right]\right] \\
&+ \mathrm{Var}\left[\sum_{t=1}^{h-2} R_t + \mathbb{E}\left[R_{h-1} + \sum_a b_{V,h}^\pi(a, O_h, H_{h-1}) \,\Big|\, O_{h-1}, S_{h-1}, A_{h-1}, H_{h-2}\right]\right] \\
=&\mathbb{E}\left[\mathrm{Var}\left[R_h + \sum_a b_{V,h+1}^\pi(a, O_{h+1}, H_h) \,\Big|\, O_h, S_h, A_h, H_{h-1}\right]\right] \\
&+ \mathbb{E}\left[\mathrm{Var}\left[R_{h-1} + \sum_a b_{V,h}^\pi(a, O_h, H_{h-1}) \,\Big|\, O_{h-1}, S_{h-1}, A_{h-1}, H_{h-2}\right]\right] \\
&+ \mathrm{Var}\left[\sum_{t=1}^{h-2} R_t + \sum_a b_{V,h-1}^\pi(a, O_{h-1}, H_{h-2})\right] \qquad \text{(by Equation (25))} \\
=&\cdots \\
\geq&\sum_{t=1}^{h} \mathbb{E}^\pi\left[\mathrm{Var}\left[R_t + \sum_a b_{V,t+1}^\pi(a, O_{t+1}, H_t) \,\Big|\, O_t, S_t, A_t, H_{t-1}\right]\right],
\end{aligned}
$$

Then, letting $h = T$ and noting that $b_{V,T+1}^\pi = 0$, we get the desired result. $\qquad\square$

### C.1.3 Proof of Lemma 3

**Lemma 3.** *The variance of $\mathcal{E}(E_1)$ can be decomposed as follows,*

$$
\begin{aligned}
&Var\left\{ \sum_{t=1}^{T} \mathcal{E}\left( \prod_{t'=1}^{t} \frac{\pi_{t'}}{\pi_{t'}^b} \psi_t^\top \mathbf{P}_{a_t}^\dagger \sum_{r_t, o_{t+1}} \widehat{\mathbf{P}}_{a_t, r_t, o_{t+1}} \psi_t \Big[ \sum_{a_t} b_{V,t}^\pi - \big(r_t + \sum_{a_{t+1}} b_{V,t+1}^\pi\big) \Big] \right) \right\} \\
=&\sum_{t=1}^{T} \mathbb{E}\left\{ Var\left[ \mathcal{E}\left( \prod_{t'=1}^{t} \frac{\pi_{t'}}{\pi_{t'}^b} \psi_t^\top \mathbf{P}_{a_t}^\dagger \sum_{r_t, o_{t+1}} \widehat{\mathbf{P}}_{a_t, r_t, o_{t+1}} \psi_t \Big[ \sum_{a_t} b_{V,t}^\pi - \big(r_t + \sum_{a_{t+1}} b_{V,t+1}^\pi\big) \Big] \right) \,\Big|\, \mathcal{D}_t \right] \right\}.
\end{aligned}
$$

*Proof.* By iteratively applying the law of total variance, we have

$$
\begin{aligned}
&\mathrm{Var}\left\{ \sum_{t=1}^{T} \mathcal{E}\left( \prod_{t'=1}^{t} \frac{\pi_{t'}}{\pi_{t'}^b} \psi_t^\top \mathbf{P}_{A_t}^\dagger \sum_{r_t, o_{t+1}} \widehat{\mathbf{P}}_{A_t, r_t, o_{t+1}} \psi_t \Big[ \sum_{a_t} b_{V,t}^\pi - \big(r_t + \sum_{a_{t+1}} b_{V,t+1}^\pi\big) \Big] \right) \right\} \\
=&\mathbb{E}\left\{ \mathrm{Var}\left[ \sum_{t=1}^{T} \mathcal{E}\left( \prod_{t'=1}^{t} \frac{\pi_{t'}}{\pi_{t'}^b} \psi_t^\top \mathbf{P}_{A_t}^\dagger \sum_{r_t, o_{t+1}} \widehat{\mathbf{P}}_{A_t, r_t, o_{t+1}} \psi_t \Big[ \sum_{a_t} b_{V,t}^\pi - \big(r_t + \sum_{a_{t+1}} b_{V,t+1}^\pi\big) \Big] \right) \,\Big|\, \mathcal{D}_T \right] \right\} \\
&+ \mathrm{Var}\left\{ \mathbb{E}\left[ \sum_{t=1}^{T} \mathcal{E}\left( \prod_{t'=1}^{t} \frac{\pi_{t'}}{\pi_{t'}^b} \psi_t^\top \mathbf{P}_{A_t}^\dagger \sum_{r_t, o_{t+1}} \widehat{\mathbf{P}}_{A_t, r_t, o_{t+1}} \psi_t \Big[ \sum_{a_t} b_{V,t}^\pi - \big(r_t + \sum_{a_{t+1}} b_{V,t+1}^\pi\big) \Big] \right) \,\Big|\, \mathcal{D}_T \right] \right\} \\
=&\mathbb{E}\left\{ \mathrm{Var}\left[ \sum_{t=1}^{T} \mathcal{E}\left( \prod_{t'=1}^{t} \frac{\pi_{t'}}{\pi_{t'}^b} \psi_t^\top \mathbf{P}_{A_t}^\dagger \sum_{r_t, o_{t+1}} \widehat{\mathbf{P}}_{A_t, r_t, o_{t+1}} \psi_t \Big[ \sum_{a_t} b_{V,t}^\pi - \big(r_t + \sum_{a_{t+1}} b_{V,t+1}^\pi\big) \Big] \right) \,\Big|\, \mathcal{D}_T \right] \right\}
\end{aligned}
$$

$$+ \text{Var}\left\{ \sum_{t=1}^{T-1} \mathcal{E}\left( \prod_{t'=1}^{t} \frac{\pi_{t'}}{\pi_{t'}^b} \psi_t^\top \mathbf{P}_{A_t}^\dagger \sum_{r_t,o_{t+1}} \widehat{\mathbf{P}}_{A_t,r_t,o_{t+1}} \psi_t \Big[ \sum_{a_t} b_{V,t}^\pi - \big( r_t + \sum_{a_{t+1}} b_{V,t+1}^\pi \big) \Big] \right) \right\}$$

$$= \mathbb{E}\left\{ \text{Var}\left[ \sum_{t=1}^{T} \mathcal{E}\left( \prod_{t'=1}^{t} \frac{\pi_{t'}}{\pi_{t'}^b} \psi_t^\top \mathbf{P}_{A_t}^\dagger \sum_{r_t,o_{t+1}} \widehat{\mathbf{P}}_{A_t,r_t,o_{t+1}} \psi_t \Big[ \sum_{a_t} b_{V,t}^\pi - \big( r_t + \sum_{a_{t+1}} b_{V,t+1}^\pi \big) \Big] \right) \Big| \mathcal{D}_T \right] \right\}$$

$$+ \mathbb{E}\left\{ \text{Var}\left[ \sum_{t=1}^{T-1} \mathcal{E}\left( \prod_{t'=1}^{t} \frac{\pi_{t'}}{\pi_{t'}^b} \psi_t^\top \mathbf{P}_{A_t}^\dagger \sum_{r_t,o_{t+1}} \widehat{\mathbf{P}}_{A_t,r_t,o_{t+1}} \psi_t \Big[ \sum_{a_t} b_{V,t}^\pi - \big( r_t + \sum_{a_{t+1}} b_{V,t+1}^\pi \big) \Big] \right) \Big| \mathcal{D}_{T-1} \right] \right\}$$

$$+ \text{Var}\left\{ \mathbb{E}\left[ \sum_{t=1}^{T-1} \mathcal{E}\left( \prod_{t'=1}^{t} \frac{\pi_{t'}}{\pi_{t'}^b} \psi_t^\top \mathbf{P}_{A_t}^\dagger \sum_{r_t,o_{t+1}} \widehat{\mathbf{P}}_{A_t,r_t,o_{t+1}} \psi_t \Big[ \sum_{a_t} b_{V,t}^\pi - \big( r_t + \sum_{a_{t+1}} b_{V,t+1}^\pi \big) \Big] \right) \Big| \mathcal{D}_{T-1} \right] \right\}$$

$$= \cdots$$

$$= \sum_{t=1}^{T} \mathbb{E}\left\{ \text{Var}\left[ \mathcal{E}\left( \prod_{t'=1}^{t} \frac{\pi_{t'}}{\pi_{t'}^b} \psi_t^\top \mathbf{P}_{A_t}^\dagger \sum_{r_t,o_{t+1}} \widehat{\mathbf{P}}_{A_t,r_t,o_{t+1}} \psi_t \Big[ \sum_{a_t} b_{V,t}^\pi - \big( r_t + \sum_{a_{t+1}} b_{V,t+1}^\pi \big) \Big] \right) \Big| \mathcal{D}_t \right] \right\}.$$

$\square$

### C.1.4 Proof of Lemma 4

**Lemma 4.** *For any function $f_t : \mathcal{A} \times \mathcal{O} \times \mathcal{H}_{t-1} \to \mathbb{R}^d$, the following holds:*

$$\mathbb{E}\left[ \frac{p_t^\pi(S_t, H_{t-1})}{p_t^{\pi^b}(S_t, H_{t-1})\pi_t^b(A_t \mid S_t)} f_t(A_t, O_t, H_{t-1}) \right]$$

$$= \mathbb{E}\left[ \frac{p_{t-1}^\pi(S_{t-1}, H_{t-2})\pi_{t-1}(A_{t-1} \mid O_{t-1}, H_{t-2})}{p_{t-1}^{\pi^b}(S_{t-1}, H_{t-2})\pi_{t-1}^b(A_{t-1} \mid S_{t-1})} \sum_a f_t(a, O_t, H_{t-1}) \right].$$

*Proof.* For the left-hand side of the equation, we have

$$\mathbb{E}\left[ \frac{p_t^\pi(S_t, H_{t-1})}{p_t^{\pi^b}(S_t, H_{t-1})\,\pi_t^b(A_t \mid S_t)} f_t(A_t, O_t, H_{t-1}) \right]$$

$$= \int \left[ \frac{p_t^\pi(s_t, h_{t-1})}{p_t^{\pi^b}(s_t, h_{t-1})\,\pi_t^b(a_t \mid s_t)} f_t(a_t, o_t, h_{t-1}) \right] p_t^{\pi^b}(a_t, o_t, s_t, h_{t-1})\,da_t do_t ds_t dh_{t-1}$$

$$= \int \left[ \frac{p_t^\pi(s_t, h_{t-1})}{p_t^{\pi^b}(s_t, h_{t-1})\,\pi_t^b(a_t \mid s_t)} f_t(a_t, o_t, h_{t-1}) \right] \pi_t^b(a_t \mid s_t)\,p(o_t \mid s_t)\,p_t^{\pi^b}(s_t, h_{t-1})\,da_t do_t ds_t dh_{t-1}$$

by $O_t \perp\!\!\!\perp A_t \mid S_t$

$$= \int_{o_t,s_t,h_{t-1}} \sum_a \frac{p_t^\pi(s_t, h_{t-1})}{p_t^{\pi^b}(s_t, h_{t-1})\,\pi_t^b(a \mid s_t)} f_t(a, o_t, h_{t-1})\,\pi_t^b(a \mid s_t)\,p(o_t \mid s_t)\,p_t^{\pi^b}(s_t, h_{t-1})\,do_t ds_t dh_{t-1}$$

$$= \int_{o_t,s_t,h_{t-1}} \sum_a f_t(a, o_t, h_{t-1})\,p(o_t \mid s_t)\,p_t^\pi(s_t, h_{t-1})\,do_t ds_t dh_{t-1}.$$

The third equality holds by taking expectation with respect to $A_t$.

Now we analyze the right-hand side term.

$$\mathbb{E}\left[\frac{p_{t-1}^{\pi}\left(S_{t-1}, H_{t-2}\right) \pi_{t-1}\left(A_{t-1} \mid O_{t-1}, H_{t-2}\right)}{p_{t-1}^{\pi^b}\left(S_{t-1}, H_{t-2}\right) \pi_{t-1}^{b}\left(A_{t-1} \mid S_{t-1}\right)} \sum_{a} f_t\left(a, O_t, H_{t-1}\right)\right]$$

$$=\int\left[\frac{p_{t-1}^{\pi}\left(s_{t-1}, h_{t-2}\right) \pi_{t-1}\left(a_{t-1} \mid o_{t-1}, h_{t-2}\right)}{p_{t-1}^{\pi^b}\left(s_{t-1}, h_{t-2}\right) \pi_{t-1}^{b}\left(a_{t-1} \mid s_{t-1}\right)} \sum_{a} f_t\left(a, o_t, h_{t-1}\right)\right]$$

$$p^{\pi^b}\left(o_t, a_{t-1}, o_{t-1}, s_{t-1}, h_{t-2}\right) do_t da_{t-1} do_{t-1} ds_{t-1} dh_{t-2}$$

$$=\int\left[\frac{p_{t-1}^{\pi}\left(s_{t-1}, h_{t-2}\right) \pi_{t-1}\left(a_{t-1} \mid o_{t-1}, h_{t-2}\right)}{p_{t-1}^{\pi^b}\left(s_{t-1}, h_{t-2}\right) \pi_{t-1}^{b}\left(a_{t-1} \mid s_{t-1}\right)} \sum_{a} f_t\left(a, o_t, h_{t-1}\right)\right]$$

$$\int p^{\pi^b}\left(o_t, s_t, a_{t-1}, o_{t-1}, s_{t-1}, h_{t-2}\right) ds_t do_t da_{t-1} do_{t-1} ds_{t-1} dh_{t-2} \quad \text{(marginalization over } S_t)$$

$$=\int\frac{p_{t-1}^{\pi}\left(s_{t-1}, h_{t-2}\right) \pi_{t-1}\left(a_{t-1} \mid o_{t-1}, h_{t-2}\right)}{p_{t-1}^{\pi^b}\left(s_{t-1}, h_{t-2}\right) \pi_{t-1}^{b}\left(a_{t-1} \mid s_{t-1}\right)} \sum_{a} f_t\left(a, o_t, h_{t-1}\right)$$

$$p^{\pi^b}\left(o_t, s_t, a_{t-1}, o_{t-1}, s_{t-1}, h_{t-2}\right) ds_t do_t da_{t-1} do_{t-1} ds_{t-1} dh_{t-2}$$

$$=\int p_{t-1}^{\pi}\left(s_{t-1}, h_{t-2}\right) \pi_{t-1}\left(a_{t-1} \mid o_{t-1}, h_{t-2}\right) \sum_{a} f_t\left(a, o_t, h_{t-1}\right)$$

$$p\left(o_t \mid s_t\right) p\left(s_t \mid a_{t-1}, s_{t-1}\right) p\left(o_{t-1} \mid s_{t-1}\right) do_t ds_t da_{t-1} do_{t-1} ds_{t-1} dh_{t-2}$$

$$=\int_{O_{o_t}, s_t, h_{t-1}} \sum_{a} f_t\left(a, o_t, h_{t-1}\right) p\left(o_t \mid s_t\right) p_t^{\pi}\left(s_t, h_{t-1}\right) do_t ds_t dh_{t-1}.$$

The last equality is by noticing that

$$p_t^{\pi}\left(s_t, h_{t-1}\right) = p(s_t \mid s_{t-1}, a_{t-1})\pi_t(a_{t-1} \mid o_{t-1}, h_{t-2})p(o_{t-1} \mid s_{t-1})p_{t-1}^{\pi}(s_{t-1}, h_{t-2}).$$

Then the proof is completed by comparing these two terms. $\qquad\square$

### C.1.5 Bounding $\mathcal{E}(E_2)$

Similar as before, we treat $\mathcal{E}(E_2) = \sum_{t=1}^{T} \mathcal{E}(I_t)$ as the sum of $T$ terms where

$$\mathcal{E}(I_1) =\mathcal{E}\left\{\left(\sum_{a_1} \pi_1(a_1|O_1)\psi_1^{\top}(O_1)\widehat{\mathbf{P}}_{a_1}^{\dagger} \sum_{r_1, o_2} \widehat{\mathbf{P}}_{a_1, r_1, o_2}\psi_1(O_1) - \sum_{a_1} \pi_1(a_1|O_1)\psi_1^{\top}(O_1)\mathbf{P}_{a_1}^{\dagger} \sum_{r_1, o_2} \widehat{\mathbf{P}}_{a_1, r_1, o_2}\psi_1(O_1)\right)\right.$$
$$\left.\left[\sum_{a_1} b_{V,1}^{\pi}(a_1, O_1) - \left(r_1 + \sum_{a_2} b_{V,2}^{\pi}(a_2, o_2, (O_1, a_1))\right)\right]\right\},$$

$$\mathcal{E}(I_2) =\mathcal{E}\left\{\left(\sum_{a_1} \pi_1(a_1|O_1)\psi_1^{\top}(O_1)\widehat{\mathbf{P}}_{a_1}^{\dagger} \sum_{r_1, o_2} \widehat{\mathbf{P}}_{a_1, r_1, o_2}\psi_1(O_1) \cdot \sum_{a_2} \pi_2(a_2|o_2, (O_1, a_1))\psi_2^{\top}\widehat{\mathbf{P}}_{a_2}^{\dagger} \sum_{r_2, o_3} \widehat{\mathbf{P}}_{a_2, r_2, o_3}\psi_2\right.\right.$$
$$\left.- \sum_{a_1} \pi_1(a_1|O_1)\psi_1^{\top}(O_1)\mathbf{P}_{a_1}^{\dagger} \sum_{r_1, o_2} \mathbf{P}_{a_1, r_1, o_2}\psi_1(O_1) \cdot \sum_{a_2} \pi_2(a_2|o_2, (O_1, a_1))\psi_2^{\top}\mathbf{P}_{a_2}^{\dagger} \sum_{r_2, o_3} \widehat{\mathbf{P}}_{a_2, r_2, o_3}\psi_2\right)$$
$$\left.\left[\sum_{a_2} b_{V,2}^{\pi}(a_2, o_2, (O_1, a_1)) - \left(r_2 + \sum_{a_3} b_{V,3}^{\pi}(a_3, o_3, (O_1, a_1, o_2, a_2))\right)\right]\right\},$$
$$\cdots$$

$$\mathcal{E}(I_T) =\mathcal{E}\left\{\left(\sum_{a_1} \pi_1\psi_1^{\top}\widehat{\mathbf{P}}_{a_1}^{\dagger} \sum_{r_1, o_2} \widehat{\mathbf{P}}_{a_1, r_1, o_2}\psi_1 \sum_{a_2} \pi_2\psi_2^{\top}\widehat{\mathbf{P}}_{a_2}^{\dagger} \sum_{r_2, o_3} \widehat{\mathbf{P}}_{a_2, r_2, o_3}\psi_2 \cdots \sum_{a_T} \pi_T\psi_T^{\top}\mathbf{P}_{a_T}^{\dagger} \sum_{r_T} \widehat{\mathbf{P}}_{a_T, r_T}\psi_T\right.\right.$$
$$\left.- \sum_{a_1} \pi_1\psi_1^{\top}\mathbf{P}_{a_1}^{\dagger} \sum_{r_1, o_2} \mathbf{P}_{a_1, r_1, o_2}\psi_1 \sum_{a_2} \pi_2\psi_2^{\top}\mathbf{P}_{a_2}^{\dagger} \sum_{r_2, o_3} \mathbf{P}_{a_2, r_2, o_3}\psi_2 \cdots \sum_{a_T} \pi_T\psi_T^{\top}\mathbf{P}_{a_T}^{\dagger} \sum_{r_T} \widehat{\mathbf{P}}_{a_T, r_T}\psi_T\right)$$
$$\left.\left[\sum_{a_T} b_{V,T}^{\pi}(a_T, o_T, (O_1, a_1, \cdots, a_{T-1})) - r_T\right]\right\}.$$

Next, we aim to derive the bound of $\mathrm{Var}[\mathcal{E}(E_2)]$. We first split $\mathcal{E}(E_2)$ into two parts:

$$\mathcal{E}(E_2)$$

$$=\mathcal{E}\Bigg\{\sum_{t=1}^{T}\Big(\sum_{a_1}\pi_1\psi_1^\top\widehat{\mathbf{P}}_{a_1}^\dagger\cdots\sum_{r_t,o_{t+1}}\widehat{\mathbf{P}}_{a_t,r_t,o_{t+1}}\psi_t-\sum_{a_1}\pi_1\psi_1^\top\mathbf{P}_{a_1}^\dagger\cdots\sum_{r_t,o_{t+1}}\widehat{\mathbf{P}}_{a_t,r_t,o_{t+1}}\psi_t\Big)$$

$$\Big[\sum_{a_t}b_{V,t}^\pi(a_t,o_t,(O_1,A_1,\ldots,o_{t-1},a_{t-1}))-\Big(r_t+\sum_{a_{t+1}}b_{V,t+1}^\pi(a_{t+1},o_{t+1},(O_1,A_1,\ldots,o_t,a_t))\Big)\Big]\Bigg\}$$

$$=\mathcal{E}\Bigg\{\sum_{t=1}^{T}\Big(\big(\prod_{t'=1}^{t}\frac{\pi_{t'}}{\pi_{t'}^b}\big)\psi_1^\top\widehat{\mathbf{P}}_{A_1}^\dagger\cdots\sum_{r_t,o_{t+1}}\widehat{\mathbf{P}}_{A_t,r_t,o_{t+1}}\psi_t-\big(\prod_{t'=1}^{t}\frac{\pi_{t'}}{\pi_{t'}^b}\big)\psi_1^\top\mathbf{P}_{A_1}^\dagger\cdots\sum_{r_t,o_{t+1}}\widehat{\mathbf{P}}_{A_t,r_t,o_{t+1}}\psi_t\Big)$$

$$\Big[\sum_{a_t}b_{V,t}^\pi(a_t,o_t,(O_1,A_1,\ldots,o_{t-1},A_{t-1}))-\Big(r_t+\sum_{a_{t+1}}b_{V,t+1}^\pi(a_{t+1},o_{t+1},(O_1,A_1,\ldots,o_t,A_t))\Big)\Big]\Bigg\}$$

$$=\mathcal{E}\Bigg\{\sum_{t=1}^{T}\big(\prod_{t'=1}^{t}\frac{\pi_{t'}}{\pi_{t'}^b}\big)\psi_1^\top\mathbf{P}_{A_1}^\dagger\cdots\sum_{r_{t-1},o_t}\mathbf{P}_{A_{t-1},r_{t-1},o_t}\psi_{t-1}\big(\psi_t^\top\widehat{\mathbf{P}}_{A_t}^\dagger-\psi_t^\top\mathbf{P}_{A_t}^\dagger\big)\sum_{r_t,o_{t+1}}\widehat{\mathbf{P}}_{A_t,r_t,o_{t+1}}\psi_t$$

$$\Big[\sum_{a_t}b_{V,t}^\pi(a_t,o_t,(O_1,A_1,\ldots,o_{t-1},A_{t-1}))-\Big(r_t+\sum_{a_{t+1}}b_{V,t+1}^\pi(a_{t+1},o_{t+1},(O_1,A_1,\ldots,o_t,A_t))\Big)\Big]\Bigg\}$$

$$+\mathcal{E}\Bigg\{\sum_{t=1}^{T}\Big(\big(\prod_{t'=1}^{t}\frac{\pi_{t'}}{\pi_{t'}^b}\big)\psi_1^\top\widehat{\mathbf{P}}_{A_1}^\dagger\cdots\sum_{r_{t-1},o_t}\widehat{\mathbf{P}}_{A_{t-1},r_{t-1},o_t}\psi_{t-1}-\big(\prod_{t'=1}^{t}\frac{\pi_{t'}}{\pi_{t'}^b}\big)\psi_1^\top\mathbf{P}_{A_1}^\dagger\cdots\sum_{r_{t-1},o_t}\mathbf{P}_{A_{t-1},r_{t-1},o_t}\psi_{t-1}\Big)$$

$$\psi_t^\top\widehat{\mathbf{P}}_{A_t}^\dagger\sum_{r_t,o_{t+1}}\widehat{\mathbf{P}}_{A_t,r_t,o_{t+1}}\psi_t\Big[\sum_{a_t}b_{V,t}^\pi(a_t,o_t,(O_1,A_1,\ldots,o_{t-1},A_{t-1}))-\Big(r_t+\sum_{a_{t+1}}b_{V,t+1}^\pi(a_{t+1},o_{t+1},(O_1,A_1,\ldots,o_t,A_t))\Big)\Big]\Bigg\}$$

$$=E_{2,1}+E_{2,2}$$

**(1) Bounding $\mathcal{E}(E_{2,1})$** We first verify that $\mathbb{E}[\mathcal{E}(E_{21})]=0$. For each $t\in[T]$, it holds that

$$\mathbb{E}\Bigg\{\mathbb{E}\Bigg\{\mathcal{E}\Bigg\{\sum_{t=1}^{T}\big(\prod_{t'=1}^{t}\frac{\pi_{t'}}{\pi_{t'}^b}\big)\psi_1^\top\mathbf{P}_{A_1}^\dagger\cdots\sum_{r_{t-1},o_t}\mathbf{P}_{A_{t-1},r_{t-1},o_t}\psi_{t-1}\big(\psi_t^\top\widehat{\mathbf{P}}_{A_t}^\dagger-\psi_t^\top\mathbf{P}_{A_t}^\dagger\big)\sum_{r_t,o_{t+1}}\widehat{\mathbf{P}}_{A_t,r_t,o_{t+1}}\psi_t$$

$$\Big[\sum_{a_t}b_{V,t}^\pi(a_t,o_t,(O_1,A_1,\ldots,o_{t-1},A_{t-1}))-\Big(r_t+\sum_{a_{t+1}}b_{V,t+1}^\pi(a_{t+1},o_{t+1},(O_1,A_1,\ldots,o_t,A_t))\Big)\Big]\Bigg\}\Big|\mathcal{D}_t\Bigg\}\Bigg\}$$

$$=\mathbb{E}\Bigg\{\mathcal{E}\Bigg\{\sum_{t=1}^{T}\big(\prod_{t'=1}^{t}\frac{\pi_{t'}}{\pi_{t'}^b}\big)\psi_1^\top\mathbf{P}_{A_1}^\dagger\cdots\sum_{r_{t-1},o_t}\mathbf{P}_{A_{t-1},r_{t-1},o_t}\psi_{t-1}\big(\psi_t^\top\widehat{\mathbf{P}}_{A_t}^\dagger-\psi_t^\top\mathbf{P}_{A_t}^\dagger\big)\sum_{r_t,o_{t+1}}\mathbf{P}_{A_t,r_t,o_{t+1}}\psi_t$$

$$\Big[\sum_{a_t}b_{V,t}^\pi(a_t,o_t,(O_1,A_1,\ldots,o_{t-1},A_{t-1}))-\Big(r_t+\sum_{a_{t+1}}b_{V,t+1}^\pi(a_{t+1},o_{t+1},(O_1,A_1,\ldots,o_t,A_t))\Big)\Big]\Bigg\}\Bigg\}$$

$$=\mathbb{E}\Bigg\{\mathcal{E}\Bigg\{\sum_{t=1}^{T}\big(\prod_{t'=1}^{t}\frac{\pi_{t'}}{\pi_{t'}^b}\big)\psi_1^\top\mathbf{P}_{A_1}^\dagger\cdots\sum_{r_{t-1},o_t}\mathbf{P}_{A_{t-1},r_{t-1},o_t}\psi_{t-1}\big(\psi_t^\top\mathbf{P}_{A_t}^\dagger-\psi_t^\top\mathbf{P}_{A_t}^\dagger\big)\sum_{r_t,o_{t+1}}\mathbf{P}_{A_t,r_t,o_{t+1}}\psi_t$$

$$\Big[\sum_{a_t}b_{V,t}^\pi(a_t,o_t,(O_1,A_1,\ldots,o_{t-1},A_{t-1}))-\Big(r_t+\sum_{a_{t+1}}b_{V,t+1}^\pi(a_{t+1},o_{t+1},(O_1,A_1,\ldots,o_t,A_t))\Big)\Big]\Bigg\}\Bigg\}$$

$$=0.$$

With probability at least $1 - T|\mathcal{A}|\delta$, we have the upper bound such that

$$
\mathrm{Var}\Big[\mathcal{E}(E_{2,1})\Big]
$$

$$
\overset{(i)}{=}\sum_{t=1}^{T}\mathbb{E}\Bigg\{\mathrm{Var}\Bigg[\mathcal{E}\Bigg(\Big(\prod_{t'=1}^{t}\frac{\pi_{t'}}{\pi_{t'}^{b}}\Big)\psi_1^{\top}\mathbf{P}_{A_1}^{\dagger}\cdots\sum_{r_{t-1},o_t}\mathbf{P}_{A_{t-1},r_{t-1},o_t}\psi_{t-1}\Big(\psi_t^{\top}\widehat{\mathbf{P}}_{A_t}^{\dagger}-\psi_t^{\top}\mathbf{P}_{A_t}^{\dagger}\Big)\sum_{r_t,o_{t+1}}\widehat{\mathbf{P}}_{A_t,r_t,o_{t+1}}\psi_t
$$

$$
\Big[\sum_{a_t}b_{V,t}^{\pi}(a_t,o_t,(O_1,A_1,\ldots,o_{t-1},A_{t-1}))-\Big(r_t+\sum_{a_{t+1}}b_{V,t+1}^{\pi}(a_{t+1},o_{t+1},(O_1,A_1,\ldots,o_t,A_t))\Big)\Big]\Bigg)\Bigg]\Big|\,\mathcal{D}_t\Bigg]\Bigg\}
$$

$$
\leq\frac{T^2}{n}(1-\theta^*)^{-1}|\mathcal{A}|\sum_{t=1}^{T}
$$

$$
\mathbb{E}\Bigg\{\mathcal{E}\Bigg(\Big(\prod_{t'=1}^{t}\frac{\pi_{t'}}{\pi_{t'}^{b}}\Big)\psi_1^{\top}\mathbf{P}_{A_1}^{\dagger}\cdots\sum_{r_{t-1},o_t}\mathbf{P}_{A_{t-1},r_{t-1},o_t}\psi_{t-1}\Big(\psi_t^{\top}\widehat{\mathbf{P}}_{A_t}^{\dagger}-\psi_t^{\top}\mathbf{P}_{A_t}^{\dagger}\Big)\sum_{r_t,o_{t+1}}\mathrm{diag}\Big(\mathbf{P}_{A_t,r_t,o_{t+1}}\psi_t\Big)\Bigg)^2\Bigg\}
$$

$$
=\frac{T^2}{n}(1-\theta^*)^{-1}|\mathcal{A}|\sum_{t=1}^{T}\mathbb{E}\Bigg\{\mathcal{E}\Bigg(\Big(\prod_{t'=1}^{t}\frac{\pi_{t'}}{\pi_{t'}^{b}}\Big)\psi_1^{\top}\mathbf{P}_{A_1}^{\dagger}\cdots\mathbf{P}_{A_{t-1}}\psi_{t-1}\psi_t^{\top}\Big(\widehat{\mathbf{P}}_{A_t}^{\dagger}-\mathbf{P}_{A_t}^{\dagger}\Big)\mathrm{diag}\Big(\mathbf{P}_{A_t}\psi_t\Big)\Bigg)^2\Bigg\}
$$

$$
=\frac{T^2}{n}(1-\theta^*)^{-1}|\mathcal{A}|\sum_{t=1}^{T}\mathbb{E}\Bigg\{\mathcal{E}\Bigg(\Big(\prod_{t'=1}^{t}\frac{\pi_{t'}}{\pi_{t'}^{b}}\Big)\psi_t^{\top}\Big(\widehat{\mathbf{P}}_{A_t}^{\dagger}-\mathbf{P}_{A_t}^{\dagger}\Big)\mathrm{diag}\Big(\mathbf{P}_{A_t}\psi_t\Big)\Bigg)^2\Bigg\}
$$

$$
\leq\frac{T^2}{n}(1-\theta^*)^{-1}|\mathcal{A}|\sum_{t=1}^{T}\mathbb{E}\Bigg[\prod_{t'=1}^{t}\frac{\pi_{t'}}{\pi_{t'}^{b}}\Bigg]^2\max_{a_t\in\mathcal{A}}\|\psi_t\|_2^2\big\|\mathbf{P}_{a_t}^{\dagger}-\widehat{\mathbf{P}}_{a_t}^{\dagger}\big\|_2^2\big\|\mathrm{diag}\big(\mathbf{P}_{a_t}\psi_t\big)\big\|_2^2
$$

$$
\overset{(ii)}{\leq}\frac{T^3}{n}(1-\theta^*)^{-1}|\mathcal{A}|C_{\pi^b}\sum_{t=1}^{T}\max_{a_t\in\mathcal{A}}\big\|\widehat{\mathbf{P}}_{a_t}^{\dagger}\big\|_2^2\big\|\mathbf{P}_{a_t}-\widehat{\mathbf{P}}_{a_t}\big\|_2^2\big\|\mathbf{P}_{a_t}^{\dagger}\big\|_2^2\qquad\text{(by Assumption 2(a))}
$$

$$
\overset{(iii)}{\lesssim}\frac{T^2}{n}(1-\theta^*)^{-1}C_{\pi^b}C_P^4|\mathcal{O}|^2|\mathcal{H}_{T-1}|^2\sum_{t=1}^{T}\max_{a_t\in\mathcal{A}}\big\|\mathbf{P}_{a_t}-\widehat{\mathbf{P}}_{a_t}\big\|_2^2\qquad\text{(by Equation (26))}
$$

$$
\leq\frac{T^2}{n}(1-\theta^*)^{-1}C_{\pi^b}C_P^4|\mathcal{A}|^2|\mathcal{O}|^2|\mathcal{H}_{T-1}|^2\cdot|\mathcal{O}||\mathcal{H}_{T-1}|\sum_{t=1}^{T}\frac{\log(|\mathcal{O}||\mathcal{H}_{t-1}|/\delta)}{n(1-\theta^*)}\quad\text{(by Lemma 5)}
$$

$$
\lesssim\frac{T^3}{n^2}(1-\theta^*)^{-2}C_{\pi^b}C_P^4|\mathcal{O}|^{3T}|\mathcal{A}|^{3T-1}\log(|\mathcal{O}|^T|\mathcal{A}|^{T-1}/\delta)
$$

The equality $(i)$ holds by recursively applying the law of total variance (same as Lemma 3) and $\mathbb{E}\big[\mathcal{E}(I_t)\mid\mathcal{D}_t\big]=0$. The inequality $(ii)$ uses the fact that $A^{-1}-B^{-1}=A^{-1}(B-A)B^{-1}$, $\|\psi_t\|_2=1$, and $\big\|\mathrm{diag}\big(\mathbf{P}_{A_t}\psi_t\big)\big\|_2\leq1$. For inequality $(iii)$, by Lemma 11, for any $a_t\in\mathcal{A}$, we have

$$
\sigma_{min}(\widehat{\mathbf{P}}_{a_t})\geq\sigma_{min}(\mathbf{P}_{a_t})-\|\widehat{\mathbf{P}}_{a_t}-\mathbf{P}_{a_t}\|_2.
$$

Thus, for any $a_t\in\mathcal{A}$, we have

$$
\begin{aligned}
\|\widehat{\mathbf{P}}_{a_t}^{\dagger}\|_2&=\frac{1}{\sigma_{min}(\widehat{\mathbf{P}}_{a_t})}\\
&\leq\frac{1}{\sigma_{min}(\mathbf{P}_{a_t})-\|\widehat{\mathbf{P}}_{a_t}-\mathbf{P}_{a_t}\|_2}\\
&\sim\frac{1}{\sigma_{min}(\mathbf{P}_{a_t})}\\
&\leq C_P\sqrt{|\mathcal{O}||\mathcal{H}_{t-1}|}\quad\text{(by Assumption 2).}
\end{aligned}
\tag{26}
$$

Here, we assume a sufficiently large sample size $n$, ensuring $\|\widehat{\mathbf{P}}_{a_t}-\mathbf{P}_{a_t}\|_2$ converge to 0.

Therefore, with probability at least $1-T|\mathcal{A}|\delta$, we obtain

$$
\mathcal{E}(E_{2,1})\lesssim\frac{T^{1.5}}{n}(1-\theta^*)^{-1}C_{\pi^b}^{\frac{1}{2}}C_P^2|\mathcal{O}|^{\frac{3T}{2}}|\mathcal{A}|^{\frac{3T-1}{2}}\sqrt{\log(|\mathcal{O}|^T|\mathcal{A}|^{T-1}/\delta)}
\tag{27}
$$

**(2) Bounding** $\mathcal{E}(E_{2,2})$  Next, we derive the bound for $\mathcal{E}(E_{2,2})$.

For ease of expression, we denote

$$
\begin{aligned}
\mathcal{P}_t &:= \psi_t^\top \mathbf{P}_{A_t}^\dagger \sum_{r_t, o_{t+1}} \mathbf{P}_{A_t, r_t, o_{t+1}} \psi_t, \\
\widehat{\mathcal{P}}_t &:= \psi_t^\top \widehat{\mathbf{P}}_{A_t}^\dagger \sum_{r_t, o_{t+1}} \widehat{\mathbf{P}}_{A_t, r_t, o_{t+1}} \psi_t.
\end{aligned}
$$

Notice that, for each $t \in [T]$, with probability at least $1 - |\mathcal{A}|\delta$, $\left|\mathcal{P}_t - \widehat{\mathcal{P}}_t\right|$ can be bounded by

$$
\begin{aligned}
\left|\mathcal{P}_t - \widehat{\mathcal{P}}_t\right| &= \left| \psi_t^\top \mathbf{P}_{A_t}^\dagger \mathbf{P}_{A_t} \psi_t - \psi_t^\top \widehat{\mathbf{P}}_{A_t}^\dagger \widehat{\mathbf{P}}_{A_t} \psi_t \right| \\
&= \left| \psi_t^\top \mathbf{P}_{A_t}^\dagger \mathbf{P}_{A_t} \psi_t - \psi_t^\top \widehat{\mathbf{P}}_{A_t}^\dagger \mathbf{P}_{A_t} \psi_t + \psi_t^\top \widehat{\mathbf{P}}_{A_t}^\dagger \mathbf{P}_{A_t} \psi_t - \psi_t^\top \widehat{\mathbf{P}}_{A_t}^\dagger \widehat{\mathbf{P}}_{A_t} \psi_t \right| \\
&= \left| \psi_t^\top \left( \mathbf{P}_{A_t}^\dagger - \widehat{\mathbf{P}}_{A_t}^\dagger \right) \mathbf{P}_{A_t} \psi_t + \psi_t^\top \widehat{\mathbf{P}}_{A_t}^\dagger \left( \mathbf{P}_{A_t} - \widehat{\mathbf{P}}_{A_t} \right) \psi_t \right| \\
&\leq \|\psi_t\|_2^2 \left\| \mathbf{P}_{A_t}^\dagger - \widehat{\mathbf{P}}_{A_t}^\dagger \right\|_2 \|\mathbf{P}_{A_t}\|_2 + \|\psi_t\|_2^2 \left\| \widehat{\mathbf{P}}_{A_t}^\dagger \right\|_2 \left\| \mathbf{P}_{A_t} - \widehat{\mathbf{P}}_{A_t} \right\|_2 \\
&\leq \left\| \widehat{\mathbf{P}}_{A_t}^\dagger \right\|_2 \left\| \widehat{\mathbf{P}}_{A_t} - \mathbf{P}_{A_t} \right\|_2 + \left\| \widehat{\mathbf{P}}_{A_t}^\dagger \right\|_2 \left\| \mathbf{P}_{A_t} - \widehat{\mathbf{P}}_{A_t} \right\|_2 \\
&\leq 2 \max_{a_t \in \mathcal{A}} \left\| \widehat{\mathbf{P}}_{a_t}^\dagger \right\|_2 \left\| \widehat{\mathbf{P}}_{a_t} - \mathbf{P}_{a_t} \right\|_2 \\
&\lesssim C_P |\mathcal{A}| \sqrt{|\mathcal{O}||\mathcal{H}_{t-1}|} \frac{(1-\theta^*)^{-\frac{1}{2}}}{\sqrt{n}} \sqrt{|\mathcal{O}||\mathcal{H}_{t-1}|} \sqrt{\log(|\mathcal{O}||\mathcal{H}_{t-1}|/\delta)} \\
&\qquad \text{(by Equation (26) and Lemma 5)} \\
&= \frac{(1-\theta^*)^{-\frac{1}{2}}}{\sqrt{n}} C_P |\mathcal{O}|^t |\mathcal{A}|^t \sqrt{\log(|\mathcal{O}|^t |\mathcal{A}|^{t-1}/\delta)} \\
&:= \xi_{t,n,\delta}
\end{aligned}
$$

$$(28)$$

Notice that, for each $t \in [T]$, with probability at least $1 - |\mathcal{A}|\delta$, we have

$$\left| \psi_t^\top \widehat{\mathbf{P}}_{A_t}^\dagger \sum_{r_t, o_{t+1}} \widehat{\mathbf{P}}_{A_t, r_t, o_{t+1}} \psi_t \left[ \sum_{a_t} b_{V,t}^\pi - \left( r_t + \sum_{a_{t+1}} b_{V,t+1}^\pi \right) \right] \right|$$

$$= \left| \psi_t^\top \widehat{\mathbf{P}}_{A_t}^\dagger \mathbf{P}_{A_t} \mathbf{P}_{A_t}^\dagger \sum_{r_t, o_{t+1}} \widehat{\mathbf{P}}_{A_t, r_t, o_{t+1}} \psi_t \left[ \sum_{a_t} b_{V,t}^\pi - \left( r_t + \sum_{a_{t+1}} b_{V,t+1}^\pi \right) \right] \right|$$

$$\leq \max_{a_t \in \mathcal{A}} \left| \psi_t^\top \widehat{\mathbf{P}}_{a_t}^\dagger \mathbf{P}_{a_t} \mathbf{P}_{a_t}^\dagger \sum_{r_t, o_{t+1}} \widehat{\mathbf{P}}_{a_t, r_t, o_{t+1}} \psi_t \left[ \sum_{a_t} b_{V,t}^\pi - \left( r_t + \sum_{a_{t+1}} b_{V,t+1}^\pi \right) \right] \right|$$

$$\leq \max_{a_t \in \mathcal{A}} \left\| \psi_t \widehat{\mathbf{P}}_{a_t}^\dagger \mathbf{P}_{a_t} \right\|_2 \left\| \mathbf{P}_{a_t}^\dagger \sum_{r_t, o_{t+1}} \widehat{\mathbf{P}}_{a_t, r_t, o_{t+1}} \psi_t \left[ \sum_{a_t} b_{V,t}^\pi - \left( r_t + \sum_{a_{t+1}} b_{V,t+1}^\pi \right) \right] \right\|_2$$

$$\overset{(i)}{\leq} \max_{a_t \in \mathcal{A}} \left\| \widehat{\mathbf{P}}_{a_t}^\dagger \mathbf{P}_{a_t} \right\|_2 \frac{T}{\sqrt{n}} (1 - \theta^*)^{-\frac{1}{2}} C_P \sqrt{|\mathcal{O}||\mathcal{H}_{t-1}||\mathcal{A}|}$$

$$\overset{(ii)}{\leq} \max_{a_t \in \mathcal{A}} \left\| \mathbf{I} + \widehat{\mathbf{P}}_{a_t}^\dagger \left( \mathbf{P}_{a_t} - \widehat{\mathbf{P}}_{a_t} \right) \right\|_2 \frac{T}{\sqrt{n}} (1 - \theta^*)^{-\frac{1}{2}} C_P \sqrt{|\mathcal{O}||\mathcal{H}_{t-1}||\mathcal{A}|}$$

$$\leq \max_{a_t \in \mathcal{A}} \left( \left\| \mathbf{I} \right\|_2 + \left\| \widehat{\mathbf{P}}_{a_t}^\dagger \right\|_2 \left\| \mathbf{P}_{a_t} - \widehat{\mathbf{P}}_{a_t} \right\|_2 \right) \frac{T}{\sqrt{n}} (1 - \theta^*)^{-\frac{1}{2}} C_P \sqrt{|\mathcal{O}||\mathcal{H}_{t-1}||\mathcal{A}|}$$

$$\leq \left( 1 + C_P \frac{(1 - \theta^*)^{-\frac{1}{2}}}{\sqrt{n}} |\mathcal{O}||\mathcal{H}_{t-1}||\mathcal{A}| \sqrt{\log(|\mathcal{O}||\mathcal{H}_{t-1}|/\delta)} \right) \frac{T}{\sqrt{n}} (1 - \theta^*)^{-\frac{1}{2}} C_P \sqrt{|\mathcal{O}||\mathcal{H}_{t-1}||\mathcal{A}|}$$

(by Equation (26) and Lemma 5)

$$= \left( 1 + \frac{(1 - \theta^*)^{-\frac{1}{2}}}{\sqrt{n}} C_P |\mathcal{O}|^t |\mathcal{A}|^t \sqrt{\log(|\mathcal{O}|^t |\mathcal{A}|^{t-1}/\delta)} \right) \frac{T}{\sqrt{n}} (1 - \theta^*)^{-\frac{1}{2}} C_P |\mathcal{O}|^{\frac{t}{2}} |\mathcal{A}|^{\frac{t}{2}}$$

$$= T \left( 1 + \xi_{t,n,\delta} \right) \zeta_{t,n}.$$

(29)

Here, we denote

$$\zeta_{t,n} := \frac{(1 - \theta^*)^{-\frac{1}{2}}}{\sqrt{n}} C_P |\mathcal{O}|^{\frac{t}{2}} |\mathcal{A}|^{\frac{t}{2}}.$$

For $(i)$, by using the similar argument in the proof of bounding $\mathcal{E}(E_1)$, we have

$$\max_{a_t \in \mathcal{A}} \left\| \mathbf{P}_{a_t}^\dagger \sum_{r_t, o_{t+1}} \widehat{\mathbf{P}}_{a_t, r_t, o_{t+1}} \psi_t \left[ \sum_{a_t} b_{V,t}^\pi - \left( r_t + \sum_{a_{t+1}} b_{V,t+1}^\pi \right) \right] \right\|_2 \leq \mathcal{O}_P \left( \frac{T}{\sqrt{n}} (1 - \theta^*)^{-\frac{1}{2}} C_P \sqrt{|\mathcal{O}||\mathcal{H}_{t-1}||\mathcal{A}|} \right).$$

The inequality $(ii)$ uses the fact that $A^{-1}B = I + A^{-1}(B - A)$.

Then, for $\mathcal{E}(E_{2,2})$, we have

$$\mathcal{E}(E_{2,2})$$

$$= \mathcal{E} \left( \sum_{t=1}^{T} \left( \left( \prod_{t'=1}^{t-1} \frac{\pi_{t'}}{\pi_{t'}^b} \right) \psi_1^\top \widehat{\mathbf{P}}_{A_1}^\dagger \cdots \sum_{r_{t-1}, o_t} \widehat{\mathbf{P}}_{A_{t-1}, r_{t-1}, o_t} \psi_{t-1} - \left( \prod_{t'=1}^{t-1} \frac{\pi_{t'}}{\pi_{t'}^b} \right) \psi_1^\top \mathbf{P}_{A_1}^\dagger \cdots \sum_{r_{t-1}, o_t} \mathbf{P}_{A_{t-1}, r_{t-1}, o_t} \psi_{t-1} \right) \right.$$

$$\frac{\pi_t}{\pi_t^b} \psi_t^\top \widehat{\mathbf{P}}_{A_t}^\dagger \sum_{r_t, o_{t+1}} \widehat{\mathbf{P}}_{A_t, r_t, o_{t+1}} \psi_t$$

$$\left. \left[ \sum_{a_t} b_{V,t}^\pi(a_t, o_t, (O_1, A_1, \ldots, o_{t-1}, A_{t-1})) - \left( r_t + \sum_{a_{t+1}} b_{V,t+1}^\pi(a_{t+1}, o_{t+1}, (O_1, A_1, \ldots, o_t, A_t)) \right) \right] \right)$$

$$= \mathcal{E} \left( \sum_{t=1}^{T} \left( \prod_{t'=1}^{t-1} \frac{\pi_{t'}}{\pi_{t'}^b} \widehat{\mathcal{P}}_{t'} - \prod_{t'=1}^{t-1} \frac{\pi_{t'}}{\pi_{t'}^b} \mathcal{P}_{t'} \right) \frac{\pi_t}{\pi_t^b} \psi_t^\top \widehat{\mathbf{P}}_{A_t}^\dagger \sum_{r_t, o_{t+1}} \widehat{\mathbf{P}}_{A_t, r_t, o_{t+1}} \psi_t \right.$$

$$\left. \left[ \sum_{a_t} b_{V,t}^\pi(a_t, o_t, (O_1, A_1, \ldots, o_{t-1}, A_{t-1})) - \left( r_t + \sum_{a_{t+1}} b_{V,t+1}^\pi(a_{t+1}, o_{t+1}, (O_1, A_1, \ldots, o_t, A_t)) \right) \right] \right)$$

$$\leq \mathcal{E}\left(\sum_{t=1}^{T}\left(\prod_{t'=1}^{t}\frac{\pi_{t'}}{\pi_{t'}^b}\right)\left(\prod_{t'=1}^{t-1}\widehat{\mathcal{P}}_{t'}-\prod_{t'=1}^{t-1}\mathcal{P}_{t'}\right)\psi_t^\top\widehat{\mathbf{P}}_{A_t}^\dagger\sum_{r_t,o_{t+1}}\widehat{\mathbf{P}}_{A_t,r_t,o_{t+1}}\psi_t\right.$$

$$\left.\left[\sum_{a_t}b_{V,t}^\pi(a_t,o_t,(O_1,A_1,\ldots,o_{t-1},A_{t-1}))-\left(r_t+\sum_{a_{t+1}}b_{V,t+1}^\pi(a_{t+1},o_{t+1},(O_1,A_1,\ldots,o_t,A_t))\right)\right]\right)$$

$$\leq C_{\pi^b}^{\frac{1}{2}}\sum_{t=1}^{T}\left\{\mathcal{E}\left(\left(\left(\prod_{t'=1}^{t-1}\left(\widehat{\mathcal{P}}_{t'}-\mathcal{P}_{t'}+\mathcal{P}_{t'}\right)\right)-\prod_{t'=1}^{t-1}\mathcal{P}_{t'}\right)\psi_t^\top\widehat{\mathbf{P}}_{A_t}^\dagger\sum_{r_t,o_{t+1}}\widehat{\mathbf{P}}_{A_t,r_t,o_{t+1}}\psi_t\right.\right.$$

$$\left.\left.\left[\sum_{a_t}b_{V,t}^\pi(a_t,o_t,(O_1,A_1,\ldots,o_{t-1},A_{t-1}))-\left(r_t+\sum_{a_{t+1}}b_{V,t+1}^\pi(a_{t+1},o_{t+1},(O_1,A_1,\ldots,o_t,A_t))\right)\right]\right)^2\right\}^{\frac{1}{2}}$$

(by Assumption 2(a) and Cauchy-Schwarz inequality)

$$\leq C_{\pi^b}^{\frac{1}{2}}\sum_{t=1}^{T}\left\{\mathcal{E}\left(\left|\prod_{t'=1}^{t-1}\left(\widehat{\mathcal{P}}_{t'}-\mathcal{P}_{t'}+\mathcal{P}_{t'}\right)-\prod_{t'=1}^{t-1}\mathcal{P}_{t'}\right|\cdot\left|\psi_t^\top\widehat{\mathbf{P}}_{A_t}^\dagger\sum_{r_t,o_{t+1}}\widehat{\mathbf{P}}_{A_t,r_t,o_{t+1}}\psi_t\right.\right.\right.$$

$$\left.\left.\left.\left[\sum_{a_t}b_{V,t}^\pi(a_t,o_t,(O_1,A_1,\ldots,o_{t-1},A_{t-1}))-\left(r_t+\sum_{a_{t+1}}b_{V,t+1}^\pi(a_{t+1},o_{t+1},(O_1,A_1,\ldots,o_t,A_t))\right)\right]\right|\right)^2\right\}^{\frac{1}{2}}$$

$$= C_{\pi^b}^{\frac{1}{2}}\sum_{t=1}^{T}\left\{\mathcal{E}\left(\left|\sum_{(\delta_1,\ldots,\delta_{t-1})\in\{0,1\}^{t-1}/\{1\}^{t-1}}\left(\mathcal{P}_1\right)^{\delta_1}\left(\widehat{\mathcal{P}}_1-\mathcal{P}_1\right)^{1-\delta_1}\cdots\left(\mathcal{P}_{t-1}\right)^{\delta_{t-1}}\left(\widehat{\mathcal{P}}_{t-1}-\mathcal{P}_{t-1}\right)^{1-\delta_{t-1}}\right|\right.\right.$$

$$\left.\left.\cdot\left|\psi_t^\top\widehat{\mathbf{P}}_{A_t}^\dagger\sum_{r_t,o_{t+1}}\widehat{\mathbf{P}}_{A_t,r_t,o_{t+1}}\psi_t\left[\sum_{a_t}b_{V,t}^\pi-\left(r_t+\sum_{a_{t+1}}b_{V,t+1}^\pi\right)\right]\right|\right)^2\right\}^{\frac{1}{2}}$$

$$\leq C_{\pi^b}^{\frac{1}{2}}\sum_{t=1}^{T}\sum_{(\delta_1,\ldots,\delta_{t-1})\in\{0,1\}^{t-1}/\{1\}^{t-1}}\left|\mathcal{P}_1\right|^{\delta_1}\left|\widehat{\mathcal{P}}_1-\mathcal{P}_1\right|^{1-\delta_1}\cdots\left|\mathcal{P}_{t-1}\right|^{\delta_{t-1}}\left|\widehat{\mathcal{P}}_{t-1}-\mathcal{P}_{t-1}\right|^{1-\delta_{t-1}}\cdot T\left(1+\xi_{t,n,\delta}\right)\zeta_{t,n}$$

(by Equation (29))

$$\leq C_{\pi^b}^{\frac{1}{2}}\sum_{t=1}^{T}\left\{\left(\left|\mathcal{P}_1\right|^{\delta_1}+\left|\widehat{\mathcal{P}}_1-\mathcal{P}_1\right|^{1-\delta_1}\right)\cdots\left(\left|\mathcal{P}_{t-1}\right|^{\delta_{t-1}}+\left|\widehat{\mathcal{P}}_{t-1}-\mathcal{P}_{t-1}\right|^{1-\delta_{t-1}}\right)-\left|\mathcal{P}_1\right|^{\delta_1}\cdots\left|\mathcal{P}_{t-1}\right|^{\delta_{t-1}}\right\}$$

$$\cdot T\left(1+\xi_{t,n,\delta}\right)\zeta_{t,n}$$

$$\leq C_{\pi^b}^{\frac{1}{2}}\sum_{t=1}^{T}\left[\left(|1+|\widehat{\mathcal{P}}_1-\mathcal{P}_1|\right)\cdots\left(|1+|\widehat{\mathcal{P}}_{t-1}-\mathcal{P}_{t-1}|\right)-1\right]\cdot T\left(1+\xi_{t,n,\delta}\right)\zeta_{t,n}$$

$$\leq TC_{\pi^b}^{\frac{1}{2}}\sum_{t=1}^{T}\left[\prod_{t'=1}^{t-1}\left(1+\xi_{t',n}\right)-1\right]\left(1+\xi_{t,n,\delta}\right)\zeta_{t,n}\qquad\text{(by Equation (28))}$$

$$\leq TC_{\pi^b}^{\frac{1}{2}}\sum_{t=1}^{T}\left[\left(1+\xi_{t,n,\delta}\right)^t-1\right]\left(1+\xi_{t,n,\delta}\right)\zeta_{t,n}$$

$$\overset{(i)}{\lesssim} TC_{\pi^b}^{\frac{1}{2}}\sum_{t=1}^{T}t\,\xi_{t,n,\delta}\left(1+\xi_{t,n,\delta}\right)\zeta_{t,n}$$

$$\leq T^3 C_{\pi^b}^{\frac{1}{2}}\left(1+\xi_{T,n,\delta}\right)\xi_{T,n,\delta}\zeta_{T,n}.$$

The inequality $(i)$ holds since $\left(1+\xi_{t,n,\delta}\right)^t \lesssim t\xi_{t,n,\delta}$ with $\xi_{t,n,\delta} < 1$, which is ensured by assuming a sufficiently large sample size $n$. Therefore, with the probability at least $1 - |\mathcal{A}|T^2\delta$, we obtain

$$
\begin{aligned}
\mathcal{E}(E_{2,2}) &\lesssim T^3 C_{\pi^b}^{\frac{1}{2}}\left(1+\xi_{T,n,\delta}\right)\xi_{T,n,\delta}\zeta_{T,n} \\
&= \frac{T^3}{n}(1-\theta^*)^{-1}C_{\pi^b}^{\frac{1}{2}}C_P^2|\mathcal{O}|^{\frac{3T}{2}}|\mathcal{A}|^{\frac{3T}{2}}\sqrt{\log(|\mathcal{O}|^T|\mathcal{A}|^{T-1}/\delta)} \\
&\quad + \frac{T^3}{n^{\frac{3}{2}}}(1-\theta^*)^{-\frac{3}{2}}C_{\pi^b}^{\frac{1}{2}}C_P^3|\mathcal{O}|^{\frac{5T}{2}}|\mathcal{A}|^{\frac{5T}{2}}\log(|\mathcal{O}|^T|\mathcal{A}|^{T-1}/\delta)
\end{aligned}
\tag{30}
$$

Combining with (27) and (30), with the probability of at least $1 - |\mathcal{A}|(T^2+T)\delta$, the upper bound of $\mathcal{E}(E_2)$ is

$$
\begin{aligned}
\mathcal{E}(E_2) &\lesssim \frac{T^{1.5}}{n}(1-\theta^*)^{-1}C_{\pi^b}^{\frac{1}{2}}C_P^2|\mathcal{O}|^{\frac{3T}{2}}|\mathcal{A}|^{\frac{3T-1}{2}}\sqrt{\log(|\mathcal{O}|^T|\mathcal{A}|^{T-1}/\delta)} \\
&\quad + \frac{T^3}{n}(1-\theta^*)^{-1}C_{\pi^b}^{\frac{1}{2}}C_P^2|\mathcal{O}|^{\frac{3T}{2}}|\mathcal{A}|^{\frac{3T}{2}}\sqrt{\log(|\mathcal{O}|^T|\mathcal{A}|^{T-1}/\delta)} \\
&\quad + \frac{T^3}{n^{\frac{3}{2}}}(1-\theta^*)^{-\frac{3}{2}}C_{\pi^b}^{\frac{1}{2}}C_P^3|\mathcal{O}|^{\frac{5T}{2}}|\mathcal{A}|^{\frac{5T}{2}}\log(|\mathcal{O}|^T|\mathcal{A}|^{T-1}/\delta).
\end{aligned}
$$

### C.1.6   Proof of Lemma 5

**Lemma 5.** *Let $P$ be the $d_1 \times d_2$ matrix whose rows each sum to one. Let $\widehat{P}$ denote its entry-wise estimated counterpart, where each entry $\widehat{p}_{ij} \in (0,1)$ is an unbiased estimator of $p_{ij} \in (0,1)$ based on $n_{ij}$ independent samples satisfying $\sum_{j=1}^{d_2} n_{ij} = N$ for each $i = 1,\ldots,d_1$. With probability at least $1 - \delta$, the spectral norm bound holds that*

$$
\left\|\widehat{P} - P\right\|_2 \lesssim \sqrt{\frac{d_1 \log(d_1/\delta)}{N}}.
$$

*Proof.* Without loss of generality, we assume $d_1 > d_2$.

We begin by expressing $\widehat{P} - P$ as a sum of independent random matrices

$$
\widehat{P} - P = \frac{1}{N}\sum_{k=1}^{N} E_k, \quad E_k = \begin{pmatrix} (E_k^{(1)})^\top \\ \vdots \\ (E_k^{(d_1)})^\top \end{pmatrix}, \quad E_k^{(i)} = X_k^{(i)} - P_i.
$$

Here, $\widehat{P}_i = \frac{1}{N}\sum_{k=1}^{N} X_k^{(i)}$ is an empirical average of $N$ independent multinomial trials, $P_i$ denotes the $i$-th row of $P$, and $X_k^{(i)} \in \mathbb{R}^{d_2}$ is a one-hot vector drawn independently from a multinomial distribution with mean $P_i$.

Then, we apply the matrix Bernstein inequality to the sum $\sum_{k=1}^{N} E_k/N$. Note that $\mathbb{E}[E_k] = 0$, $\|E_k\| \le 2\sqrt{d_1}$, thus it is easy to check that $\sigma^2 = \frac{d_1}{N}$. By applying Lemma 6, for any $t > 0$, we have

$$
\mathbb{P}\left(\left\|\widehat{P} - P\right\|_2 \ge t\right) = \mathbb{P}\left(\left\|\sum_{k=1}^{N} E_k/N\right\|_2 \ge t\right) \le (d_1 + d_2)\exp\left(\frac{-t^2/2}{\frac{d_1}{N} + \frac{2\sqrt{d_1}}{N}t/3}\right).
$$

Setting $t = C\sqrt{\frac{d_1 \log(d_1/\delta)}{N}}$ for a constant, with probability at least $1 - \delta$, the spectral norm of the deviation

$$
\left\|\widehat{P} - P\right\|_2 \lesssim \sqrt{\frac{d_1 \log(d_1/\delta)}{N}}.
$$

$\square$

## C.2 Bounding $\mathbb{E}[\sum_a \widehat{b}_{V,1}(a, O_1)] - \widehat{\mathbb{E}}[\sum_a \widehat{b}_{V,1}(a, O_1)]$

We first introduce the concept of Rademacher complexity, which is used to measure the size of function classes. Given any real-valued function class $\mathcal{G}$ defined over a random variable $X$ and any radius $\delta > 0$, the population Rademacher complexity is defined as

$$\mathcal{R}(\mathcal{G}) = \mathbb{E}_{\epsilon, X}\left[\sup_{g \in \mathcal{G}} \left|\frac{1}{n}\sum_{i=1}^n \epsilon_i g(X_i)\right|\right],$$

where $\{X_i\}_{i=1}^n$ are i.i.d. copies of $X$ and $\{\epsilon_i\}_{i=1}^n$ are i.i.d. Rademacher random variables taking values in $\{-1, +1\}$ with equal probability. The empirical Rademacher complexity is given by

$$\mathcal{R}_n(\mathcal{G}) = \mathbb{E}_{\epsilon}\left[\sup_{g \in \mathcal{G}} \left|\frac{1}{n}\sum_{i=1}^n \epsilon_i g(X_i)\right|\right].$$

If the function class $\mathcal{G}$ is bounded by $T$, by applying Lemma 10, the empirical Rademacher complexity can be bounded by

$$\mathcal{R}_n(\mathcal{G}) \leq T\sqrt{\frac{2\log|\mathcal{G}|}{n}}. \tag{31}$$

Note that we can separate $\mathbb{E}[\sum_a \widehat{b}_{V,1}(a, O_1)] - \widehat{\mathbb{E}}[\sum_a \widehat{b}_{V,1}(a, O_1)]$ into two parts,

$$\left|\mathbb{E}\left[\sum_a \widehat{b}_{V,1}(a, O_1)\right] - \widehat{\mathbb{E}}\left[\sum_a \widehat{b}_{V,1}(a, O_1)\right]\right|$$

$$= \left|\mathbb{E}\left[\sum_a \widehat{b}_{V,1}(a, O_1) - \sum_a b_{V,1}^\pi(a, O_1)\right] - \widehat{\mathbb{E}}\left[\sum_a \widehat{b}_{V,1}(a, O_1) - \sum_a b_{V,1}^\pi(a, O_1)\right]\right.$$

$$\left. + \mathbb{E}\left[\sum_a b_{V,1}^\pi(a, O_1)\right] - \widehat{\mathbb{E}}\left[\sum_a b_{V,1}^\pi(a, O_1)\right]\right|$$

$$\leq \left|\mathbb{E}\left[\sum_a \widehat{b}_{V,1}(a, O_1) - \sum_a b_{V,1}^\pi(a, O_1)\right] - \widehat{\mathbb{E}}\left[\sum_a \widehat{b}_{V,1}(a, O_1) - \sum_a b_{V,1}^\pi(a, O_1)\right]\right|$$

$$+ \left|\mathbb{E}\left[\sum_a b_{V,1}^\pi(a, O_1)\right] - \widehat{\mathbb{E}}\left[\sum_a b_{V,1}^\pi(a, O_1)\right]\right|$$

$$= (a) + (b).$$

For $(a)$, note that $\sum_a \widehat{b}_{V,1} - \sum_a b_{V,1}^\pi \in [-T, T]$. By applying Lemma 8, with probability at least $1 - \delta$, we have

$$(a) = \left|\mathbb{E}\left[\sum_a \widehat{b}_{V,1}(a, O_1) - \sum_a b_{V,1}^\pi(a, O_1)\right] - \widehat{\mathbb{E}}\left[\sum_a \widehat{b}_{V,1}(a, O_1) - \sum_a b_{V,1}^\pi(a, O_1)\right]\right|$$

$$\leq \sup_{b \in \mathcal{B}_1}\left(\mathbb{E}\left[\sum_a b(a, O_1) - \sum_a b_{V,1}^\pi(a, O_1)\right] - \widehat{\mathbb{E}}\left[\sum_a b(a, O_1) - \sum_a b_{V,1}^\pi(a, O_1)\right]\right) \tag{32}$$

$$\leq \mathcal{R}_n(\mathcal{B}_1) + T\sqrt{\frac{\log(2/\delta)}{2n}}$$

$$\leq 2c_0 T\sqrt{\frac{\log(c_1/\delta)}{n}} \qquad \text{(by Equation (31))}$$

where $c_0$ and $c_1$ are constants independent of $n, T$.

For $(b)$, note that $\sum_a b_{V,1}^\pi \in [0, T]$. By using Hoeffding's inequality, with probability at least $1 - \delta$, we have

$$(b) = \left|\mathbb{E}\left[\sum_a b_{V,1}^\pi(a, O_1)\right] - \widehat{\mathbb{E}}\left[\sum_a b_{V,1}^\pi(a, O_1)\right]\right| \leq c_0' T\sqrt{\frac{\log(c_1'/\delta)}{n}}, \tag{33}$$

where $c_0$ and $c_1$ are constants independent of $n, T$.

Combining (32) and (33), with probability at least $1 - 2\delta$, it holds that

$$\left| \mathbb{E}\Big[ \sum_a \widehat{b}_{V,1}(a, O_1) \Big] - \widehat{\mathbb{E}}\Big[ \sum_a \widehat{b}_{V,1}(a, O_1) \Big] \right| \lesssim T\sqrt{\frac{\log 1/\delta}{n}}.$$

## D   Additional Lemmas

**Lemma 6** (Matrix Bernstein inequality, Tropp (2012)). *Suppose $X_1, \ldots, X_n$ are mean-zero, $d_1 \times d_2$ random matrices such that $\|X_i\| \leq C$ almost surely for all $i \in \{1, \ldots, n\}$. Then for any $t \geq 0$,*

$$\mathbb{P}\left( \left\| \sum_{i=1}^n X_i \right\| \geq t \right) \leq (d_1 + d_2) \exp\left\{ \frac{-t^2/2}{\sigma^2 + Ct/3} \right\}$$

*where $\sigma^2 = \max\left\{ \| \sum_{i=1}^n \mathbb{E}[X_i^\top X_i]\|, \| \sum_{i=1}^n \mathbb{E}[X_i X_i^\top]\| \right\}$.*

**Lemma 7** (McDiarmid's inequality). *Suppose $X_1, \cdots, X_n \in \mathcal{X}$ are independent random variables and the function $f : \mathcal{X}^n \to \mathbb{R}$ is a mapping. For all $i \in \{1, \cdots, n\}$, and for all $x_1, \cdots, x_n, x_i' \in \mathcal{X}$, the function $f$ satisfies*

$$|f(x_1, \cdots, x_{i-1}, x_i, x_{i+1}, \cdots, x_n) - f(x_1, \cdots, x_{i-1}, x_i', x_{i+1}, \cdots, x_n)| \leq c_i,$$

*then, we have*

$$\mathbb{P}\left( \left| f(X_1, \cdots, X_n) - \mathbb{E}f(X_1, \cdots, X_n) \right| \geq t \right) \leq 2\exp\left( \frac{-2t^2}{\sum_{i=1}^n c_i^2} \right).$$

**Lemma 8.** *Suppose $\mathcal{F}$ is a finite function class with $\|f\|_\infty \leq M$, with probability at least $1 - \delta$, we have*

$$\sup_{f \in \mathcal{F}} \left| \frac{1}{n} \sum_{i=1}^n f(X_i) - \mathbb{E}[f(X)] \right| \leq 2\mathcal{R}_n(\mathcal{F}) + M\sqrt{\frac{\log(2/\delta)}{2n}}.$$

*Proof.* Using the symmetrization technique and McDiarmid's inequality (Lemma 7), for any $\delta > 0$, we have

$$\mathbb{P}\left( \sup_{f \in \mathcal{F}} \left| \frac{1}{n} \sum_{i=1}^n f(X_i) - \mathbb{E}[f(X)] \right| > 2\mathcal{R}_n(\mathcal{F}) + \epsilon \right) \leq 2\exp(-\frac{2n\epsilon^2}{M^2}).$$

Setting $2\exp\left( -\frac{2n\epsilon^2}{M^2} \right) = \delta$ and solving $\epsilon$ completes this probability inequality. $\square$

**Lemma 9** (Hoeffding's inequality). *Suppose $X_1, X_2, \ldots, X_n$ are independent random variables where $a \leq X_i \leq b$ almost surely. For any $t > 0$, we have*

$$\mathbb{P}\left( \left| \frac{1}{n} \sum_{i=1}^n X_i - \mathbb{E}[\frac{1}{n} \sum_{i=1}^n X_i] \right| > t \right) \leq 2 \cdot \exp\left\{ -\frac{2nt^2}{(b-a)^2} \right\}.$$

**Lemma 10** (Massart's finite class lemma). *Let $\mathcal{G}$ be a finite set of functions. Suppose that all functions in $\mathcal{G}$ are bounded, i.e., $\sup_{g \in \mathcal{G}} \|g\|_\infty \leq B$. Then, the empirical Rademacher complexity is bounded by*

$$\mathcal{R}_n(\mathcal{G}) \leq B\sqrt{\frac{2\log|\mathcal{G}|}{n}}.$$

**Lemma 11** (Theorem 4.11 in Stewart and Sun (1990)). *Let $A, B \in \mathbb{R}^{m \times n}$ be the same-dimensional matrix with singular value*

$$\sigma_1(A) \geq \sigma_2(A) \geq \sigma_r(A),$$
$$\sigma_1(B) \geq \sigma_2(B) \geq \sigma_r(B).$$

*Then for any unitarily invariant norm $\|\cdot\|$,*

$$diag(\sigma_i(A) - \sigma_i(B)) \leq \|A - B\|.$$

**Lemma 12** (Unbiased empirical transition probability). *The empirical transition probability estimator $\widehat{P}(s'|s,a) = \frac{N(s,a,s')}{N(s,a)}$ is unbiased, that $\mathbb{E}[\widehat{P}(s'|s,a)] = P(s'|s,a)$, where $N(s,a,s') = \sum_{i=1}^{N} \mathbb{1}\{s_i = s, a_i = a, s_{i+1} = s'\}$ and $N(s,a) = \sum_{i=1}^{N} \mathbb{1}\{s_i = s, a_i = a\}$.*

*Proof.* Given $N(s,a)$, $N(s,a,s')$ is a binomial variable such that

$$N(s,a,s')|N(s,a) \sim \text{Binomial}\big(N(s,a), P(s'|s,a)\big).$$

Thus, the expectation of this binomial variable is

$$\mathbb{E}\left[\frac{N(s,a,s')}{N(s,a)}\bigg|N(s,a)\right] = P(s'|s,a).$$

By applying the law of total expectation, we have

$$\mathbb{E}\left[\frac{N(s,a,s')}{N(s,a)}\right] = \mathbb{E}\left[\mathbb{E}\left[\frac{N(s,a,s')}{N(s,a)}\bigg|N(s,a)\right]\right] = P(s'|s,a).$$

$\square$

