# OpenReview forum: "Breaking the Order Barrier: Off-Policy Evaluation for Confounded POMDPs"
_NeurIPS.cc/2025/Conference — NeurIPS 2025 poster_

### Official Review · Reviewer_o3rs · 2025-06-06

**Clarity:** 3
**Significance:** 2
**Originality:** 3
**Rating:** 4
**Confidence:** 3

**Summary:**

The paper addresses off‐policy evaluation in finite‐horizon tabular POMDPs with unobserved states by using invertibility assumptions to treat observable histories as proxies. They propose a two‐step estimator—learning conditional probabilities from data and solving a Bellman‐like recursion over proxy value functions—and derive finite‐sample error bounds showing an $O(T^{1.5}/\sqrt{n})$ rate (improving to $O(T/\sqrt{n})$ under a concentrability condition).

**Questions:**

- I would suggest the authors to provide a more detiailed comparison with previous work in terms of assumptions and complexity, e.g., [1][2]. The current is not clear how th newly proposed method improves the previous results, and whether these better results come from algorithm improvement or benefits from stronger assumption, i.e., full-rank of $P,\hat P$, history-dependent full coverage.
- The claim ``sufficent large $n$" in Theorem 2  is not rigorous, what's the required scale?
- Since avoiding exponential dependence on the horizon is a crucial challenge in POMDPs—and prior work has sought the right complexity measures—can the proposed method leverage low-rank or lower-noise structure to improve its dependence?

[1] Shi, C., Uehara, M., Huang, J., & Jiang, N. (2022, June). A minimax learning approach to off-policy evaluation in confounded partially observable markov decision processes. In International Conference on Machine Learning (pp. 20057-20094). PMLR.

[2] Uehara, M., Kiyohara, H., Bennett, A., Chernozhukov, V., Jiang, N., Kallus, N., ... & Sun, W. (2023). Future-dependent value-based off-policy evaluation in pomdps. Advances in neural information processing systems, 36, 15991-16008.

[3] Zhang, Y., & Jiang, N. (2024). On the curses of future and history in future-dependent value functions for off-policy evaluation. arXiv preprint arXiv:2402.14703.

**Ethical Concerns:**

["NO or VERY MINOR ethics concerns only"]

**Final Justification:**

Author's response have solved my main concerns, and thus I increase my score accordingly.

**Limitations:**

- Minor: The expression in Theorem 1 is confusing because it mixes uppercase and lowercase characters. It would be clearer to write \$O\_i = o\_i\$ consistently rather than using \$O\_i\$ directly.

**Paper Formatting Concerns:**

/

**Quality:**

3

**Strengths And Weaknesses:**

- This proposed a new mothod for OPE in the context of confounded POMDP for tabular case, which is novel.
- The method is clearly motivated and well-presented, and the avoidance of minimax learning approach makes it more computationally efficient
- The exponential dependence $(|O||A|)^{T/2}$ can be high since this paper considers the ``worst" case where the underlying information structure is full-rank such that the estimation can not benefit from any low-rank architecture.
- The method closely parallels standard OPE in tabular MDPs by treating the entire history as an aggregated state for value estimation and using empirical distributions for that estimation.

---

> ### Author Rebuttal · Authors · 2025-07-31
>
> We thank the reviewer for the suggestive feedback. Our one-to-one response to the reviewer’s questions and concerns is given below.
>
>
>
> > Suggest the authors provide a more detailed comparison with previous work in terms of assumptions and complexity, e.g., [1, 2]. ...  whether these better results come from algorithm improvement or benefits from stronger assumptions.
>
> We appreciate the suggestion to provide a more detailed comparison with previous work. We would like to first clarify the differences in the settings compared to the relevant literature. Paper [1] considers a confounded POMDP setting similar to ours, while papers [2] and [3] focus on unconfounded POMDPs, where the behavior policy depends on the one-step observation. All three relevant papers [1,2,3] consider general function approximation, while we focus on the tabular setting.
>
> - **Assumptions:**  The comparison is primarily centered around the coverage assumption. [2] assumes the bounded importance sampling weight of  $\frac{\pi(A_t|O_t)}{\pi^b(A_t|O_t)}\leq \infty$, which is a standard assumption in the OPE literature. [3]  introduces a novel coverage assumption, belief coverage, which is tailored to the structure of POMDPs. This coverage assumption ensures that the data collected under the behavior policy provides sufficient information about the belief states encountered under the evaluation policy, ultimately enabling polynomial bounds on the horizon. [1] assumes the existence of weight bridge functions $b_W:\mathcal{O}\times\mathcal{A}\to\mathbb{R}$, which is similar to the weight function we define in Assumption 3. However, we do not require the estimation of the weight function.
> - **Complexity:**  Omitting the terms related to the size of the observation space, action space, and function class, the error bound in [3] and our work scale with $\mathcal{O}(\frac{T}{\sqrt{n}})$, while the results in [2] scale with  $\mathcal{O}(\frac{T^2}{\sqrt{n}})$ and [1] focuses on the discounted setting and there is no finite-sample results for finite-horizon in [1]. However, [3] focuses on the unconfounded POMDP setting, which is easier than ours.
>
> - **Source of improvement:**  The improved results stem from a fine-grained finite-sample analysis in the context of confounded POMDPs. First, we are the first to explicitly account for dependencies on the horizon $T$ , sample size $n$, the sizes of the observation space $|\mathcal{O}|$,  and the sizes of action spaces $|\mathcal{A}|$.  We carefully separate the policy value error into first-order term and higher-order terms.  Second, under bounded concentrability conditions (i.e. the weight function in assumption 3 is bounded), the dependence in Theorem 3  can be improved to $\mathcal{O}(\frac{T}{\sqrt{n}}\sqrt{|\mathcal{O}||\mathcal{A}|})$ for the memoryless case, which matches the optimal rate for OPE in MDPs. **This improvement is achieved through Lemma 1, which first establishes a novel variance decomposition specific to confounded POMDPs.**
>
>
>
> > The claim "sufficiently large $n$" in Theorem 2 is not rigorous
>
> We apologize for this confusion and will remove this claim from Theorem 2. We have assumed a sufficiently large $n$ in Assumption 2(c), where we require a sufficient number of samples for each triple $(o_0,h_{t−1},a_t)$, ensuring consistent estimation of the conditional probability matrices.  Specifically, we require $n_{o_0,h_{t-1},a_t}\ge n p_t^{\pi^b}\left(o_0, h_{t-1},a_t\right)(1-\theta_{t,ij})$, where $n_{o_0,h_{t−1},a_t}$ represents the count of the triple $(o_0, h_{t-1}, a_t)$ in the data, and $p_t^{\pi^b}(o_0, h_{t-1},a_t)$ is the probability density under the behavior policy $\pi^b$. Define the event $E :=\\{\exists ~t,o_0, h_{t-1},a_t ~ \text{s.t.} n_{o_0,h_{t-1},a_t}\ge n p_t^{\pi^b}\left(o_0, h_{t-1},a_t\right)(1-\theta_{t,ij})\\}$. Then, combining the multiplicative Chernoff bound and a union bound over each $t, o_0, h_{t-1}$, and $a_t$, we have
> $$
> \mathbb{P}[E^c] \leq \sum_t \sum_{o_0}\sum_{h_{t-1}} \sum_{a_t} \mathbb{P}\left[n_{o_0,h_{t-1},a_t}<n p_t^{\pi^b}\left(o_0, h_{t-1},a_t\right)(1-\theta_{t,ij})\right] \\
> \leq T |\mathcal{O}|^T  |\mathcal{A}|^T e^{-\frac{{\theta^*}^2 n \min_{t, o_0,h_{t-1},a_t} p_t^{\pi^b}(o_0, h_{t-1},a_t)}{2}}.
> $$
> To ensure the number of samples is sufficiently large, the $\mathbb{P}[E^c]$ should be sufficiently small. This requires $n \geq \frac{\operatorname{polylog}(|\mathcal{O}|^T, |\mathcal{A}|^T, T)}{\min_{t, o_0,h_{t-1},a_t} p_t^{\pi^b}(o_0, h_{t-1},a_t)}$.
>
> > Since avoiding exponential dependence on the horizon is a crucial challenge in POMDPs—and prior work has sought the right complexity measures—can the proposed method leverage low-rank or lower-noise structure to improve its dependence?
>
>
>
> Thank you for raising this important point.  We completely agree that avoiding exponential dependence on the horizon is a critical challenge in POMDPs. Our proposed method could indeed benefit from leveraging low-rank structures, as motivated by recent efforts.
>
> Recent work [4] assumes **latent low-rank transitions**, where the transition dynamics of the hidden state space have a low-dimensional structure. Additionally, they consider *M-step decodable POMDPs* [5], where an unknown decoder can perfectly decode the latent state from the latest $M$-step memory. The approach first learn a representation of latent states that captures only the most relevant features based on the history, and then uses the representation to learn an $M$-length memory-based policy. They demonstrate that the sample complexity can avoid exponential dependence on $T$ with respect to the sizes of the observation and action spaces, reducing to $(|\mathcal{O}||\mathcal{A}|)^{M}$, where $M$ is a fixed length that can be determined theoretically.  The paper [6] also assumes the latent low-rank transition and structure of M-step decodable POMDPs, enabling a more computationally efficient algorithm and avoiding the exponential dependence on horizon.
>
> **Incorporating these assumptions into our tabular confounded POMDP setting could potentially improve the current results**. For example, we could approximate the belief state with a lower-dimensional representation that captures the most important information. In a tabular setting, the belief state can be formulated as a matrix, where each entry is a probability over hidden states based on the history of observations and actions, such as $\mathbb{P}(S_t=s|O_o=o,H_{t-1}=h_{t-1},A_{t}=a)$. We can approximate the matrix using low-rank approximations such as SVD and PCA to reduce dimensionality, which can help keep the matrix size manageable, even for long horizons.
>
> We believe our work can naturally incorporate these structures and derive the corresponding bounds. However, given the lack of a comprehensive finite-sample analysis of confounded POMDPs, our focus has primarily been on deriving a fine-grained error bound. In future work, we will explore incorporating additional structural conditions to extend the current results.
>
>
>
> >notations in Theorem 1
>
> Thanks for this suggestion. The notation $\mathbb{P}(X)$ represents a $n$-length vector with $\mathbb{P}_ {i}(X):=p(X=x_i)$ and $\mathbb{P}(X | Y)$ denotes a $n \times m$ matrix with $\mathbb{P}_ {i, j}(X | Y):=p(X=x_j | Y=y_i)$. The lowercase letters denote realizations of a random variable, such as $X = x_i$. For clarity,  we will write $\mathbb{P}(O_{t+1}, O_t = o_t \mid A_t = a_t, O_0,H_{t-1})$ consistently, rather than $\mathbb{P}(O_{i+1}, o_i \mid a_i, O_0,H_{i-1})$ for simplicity.
>
>
>
> **Ref:**
>
> [1]  A minimax learning approach to off-policy evaluation in confounded partially observable markov decision processes. ICML, 2022.
>
> [2] Future-dependent value-based off-policy evaluation in pomdps. NIPS, 2022
>
> [3] On the curses of future and history in future-dependent value functions for off-policy evaluation. NIPS, 2024.
>
> [4] Provably Efficient Reinforcement Learning in Partially Observable Dynamical Systems. NIPS, 2022
>
> [5] Provable reinforcement learning with a short-term memory. ICML, 2022.

---

### Official Review · Reviewer_6MWe · 2025-06-15

**Clarity:** 4
**Significance:** 3
**Originality:** 3
**Rating:** 6
**Confidence:** 4

**Summary:**

This paper studies the off-policy evaluation for confounded POMDPs, which is argulably the most complicated dynamic decision-making paradigm. It designs an estimator that is based on a proxy value function, which satisfies a Bellman-type recursive equation. It also provides the finite-sample bound for the value function of the behavioral policy.

**Questions:**

- The paper explores two extremes of behavioral policies: memoryless policies that only depend on the observation; and confounded policies that can depend on the whole history. Usually there may be something in the middle. For example, in the Bayesian formulation of POMDP, one can formulate a belief state of the hidden state and then use the belief state to devide an MDP-like policy. For such policies, it can be seen that the history is first transformed to a low-dimensional representation (alhtough unobserved) and then the action depends on this reprentation. I believe the unobserved state $S_t$ can be used to model this. But because $S_t$ is not observed, it is not used in the estimator. My question is, if we know a functional form of $S_t$ (such as Bayesian updating), with some unknown parameters, is it possible to design more efficient estimators?

**Ethical Concerns:**

["NO or VERY MINOR ethics concerns only"]

**Final Justification:**

My score remains.

**Quality:**

4

**Strengths And Weaknesses:**

Strength:
- The paper is very clearly written. Although the topic is highly technical and the notation is heavy, I find it easy to navigate through the contents. The results are also explained well without getting into the technical details.
- The comparison to the literature, including the assumptions and the estimators, is thorough.

I don't find any major weaknesses of the work.

---

> ### Author Rebuttal · Authors · 2025-07-31
>
> We appreciate the reviewer for their thoughtful comments and will improve the quality accordingly!
>
> In our approach, the latent state $S_t$ is unobservable, so we do not directly include it in the estimator. However, when we have access to a functional form of $S_t$,  e.g. $S_t = f(O_{t-1},A_{t-1},\dots,)$, this can be viewed as a belief state. The form represents the history of observations and actions, allowing us to use a low-dimensional representation to substitute the latent state to model the system's dynamics. In this case, the problem reduces to an MDP-like setting, where the complexity no longer grows exponentially with the horizon.
>
>  In essence, we can leverage additional structural conditions of the POMDPs to improve the efficiency of the estimator.  Recent work [1] assumes **latent low-rank transitions** where the transition dynamics of the hidden state space have a low-dimensional structure, and assumes *M-step decodable POMDPs* [2] where there is a decoder that can perfectly decode the latent state using the latest M-step memory. The approach first learn a predictive state representation of latent states that captures only the most relevant features based on the history, and then use this representation to learn a $M$-length memory-based policy. They demonstrate that the sample complexity can avoid exponential dependence on $T$ with respect to the sizes of the observation and action spaces, reducing to $(|\mathcal{O}||\mathcal{A}|)^{M}$ , where $M$ is a fixed length that can be determined theoretically. We believe our work can naturally incorporate these structures and derive the corresponding bounds. So in future work, we will explore incorporating additional structural conditions into our confounded POMDPs.
>
>
>
>
>
> **Ref:**
>
> [1] Provably Efficient Reinforcement Learning in Partially Observable Dynamical Systems. NIPS, 2022
>
> [2] Provable reinforcement learning with a short-term memory. ICML, 2022.

---

> > ### Comment · Reviewer_6MWe · 2025-08-05
> >
> > Thank you for your clarification. I don't have further comments.

---

> > > ### Author Response · Authors · 2025-08-05
> > > **Thank You for Your Review and Feedback**
> > >
> > > Thank you once again for your careful and thoughtful review. We sincerely appreciate your positive comments!

---

### Official Review · Reviewer_mBzT · 2025-07-01

**Clarity:** 2
**Significance:** 2
**Originality:** 2
**Rating:** 4
**Confidence:** 2

**Summary:**

This paper studies off-policy evaluation (OPE) in tabular Partially Observable Markov Decision Processes (POMDPs) with unobserved confounding. The main result is finite-sample statistical error bounds of a model-based OPE estimator. For history-dependent target policies, the error scales exponentially with the horizon $T$, while for memoryless target policies, the error bound improves to polynomial horizon dependency. Under further conditions on the concentrability coefficients, the rates are improved, especially for the memoryless target policy case the rate matches the optimal rate for OPE in tabular MDPs.

**Questions:**

1. In Line 153, why is it a *sufficient* condition that $|\mathcal{O}|>|\mathcal{S}|$ to ensure the rank condition (Assumption 1)? It seems that it is only a necessary condition.
2. The statement of Theorem 1 is confusing. Are the random variables (in capital letters) taken expectation? What's their relationship with the realizations (in lowercase)?

**Ethical Concerns:**

["NO or VERY MINOR ethics concerns only"]

**Final Justification:**

The author’s response partially resolved my concerns, including clarity issues and the exponential dependency on the horizon parameter. However, I am still concerned with the technical significance of the paper, as well as the lacking of the fundamental limit of the problem, i.e., the lower bounds. Given that the paper stands as an initial step towards studying the explicit dependencies of the sample complexity on all problem parameters for tabular confounded POMDPs, I appreciate the main contributions of the paper. Overall, I am willing to raise my score to 4.

**Limitations:**

Please see the Questions part.

**Quality:**

2

**Strengths And Weaknesses:**

**Strengths:**
1. This paper is the first to study the explicit dependencies of tabular confounded POMDP OPE error bounds on the key variables including horizon $T$ and observation-action space sizes $|\mathcal{O}|$ and $|\mathcal{A}|$.
2. The improved error bounds under further concentrability conditions using variance decomposition are interesting.

**Weaknesses:**
1. The lower bound of this exact problem (OPE in tabular confounded POMDP) is not provided.
2. From the reviewer's perspective, the motivation of showing the exponential dependency on the time horizon $T$ is not convincing. On the one hand, it is conceivable that the exponential dependency does exist. On the other, real world problems of practical interests may exhibit certain structures that allow reasonable efficiency to learn a good policy, without falling into such a worst-case paradigm.
3. The techniques to obtain the identification result and the error bounds are relatively standard given existing literature on (i) OPE in confounded POMDPs; (ii) OPE in tabular MDPs.
4. Some unclear statements, please see the Questions part.

---

> ### Author Rebuttal · Authors · 2025-07-31
>
> We thank the reviewer for their valuable comments and will improve the quality accordingly! Below we respond to the weaknesses and questions.
>
>
>
>  **1. Lower bound is not provided.**
>
> The error bound scale as $\mathcal{O}(\frac{T}{\sqrt{n}})$ in Theorem 3, omitting the dependencies on the size of observation and action spaces, which matches the lower bound for tabular MDPs [1].  While no existing work has established the minimax lower bound for confounded POMDPs, due to the added complexity introduced by confounders, the lower bound for confounded POMDPs should at least be $\mathcal{O}(\frac{T}{\sqrt{n}}|\mathcal{O}|^{\alpha_1}|\mathcal{A}|^{\alpha_2})$, where $\alpha_1$ and $\alpha_2$ remain unclear and require more sophisticated methods to determine. We acknowledge that deriving a lower bound in confounded POMDPs is difficult due to the challenges of partial observability, confounders, and complex statistical dependencies. We leave this challenging direction as future work.
>
>
>
> **2. Concerns about the exponential dependency on horizon $T$**
>
> Thank you for raising this important question. In general, for POMDPs, an exponential dependence on the horizon in sample complexity is unavoidable without specific information about POMDPs.  However, as you mentioned, real-world problems may exhibit certain structures that allow the algorithm to learn efficiently without the exponential dependency on $T$. **However, achieving this requires additional structural conditions of the POMDPs.**
>
> In paper [2], under the conditions where the *latent transitions are deterministic* (i.e. $P_t(\cdot|S_t,A_t)$ is a Dirac) and the existence of a gap in optimal action values, the algorithm shows that the algorithm achieves polynomial rather than exponential growth in $T$;  Paper [3] assumes the *latent low-rank transition* where the transition dynamics of the hidden state space have a low-dimensional structure,  and consider *M-step decodable POMDPs* [4] assuming the existence of a decoder that can perfectly decode the latent state by looking at the latest $M$-memory. The approach allows learning a representation of latent states that captures only the most relevant features based on the history, showing that the sample complexity can avoid exponential dependence on $T$, reducing to $(|\mathcal{O}||\mathcal{A}|)^{M}$ , where $M$ is a fixed length that can be determined theoretically.
>
> In summary, these works show that, under additional assumptions about the structure of the POMDPs (e.g., low-rank transitions or deterministic latent dynamics), polynomial scaling on $T$ is achievable.  We believe our work can naturally incorporate these structures and derive the corresponding bounds. However, given the already lack of a comprehensive finite-sample analysis of confounded POMDPs, our focus has primarily been on deriving a fine-grained error bound. In future work, we will explore incorporating these structural conditions to extend the current results.
>
>
>
> **3. The techniques to obtain the identification result and the error bounds are relatively standard**
>
> We would like to clarify that this paper is the first to systematically analyze the finite-sample error bounds in confounded POMDPs.
>
>  Specifically, we use two main techniques to derive fine-grained error bounds in the context of confounded POMDPs. First, we are the first to explicitly account for dependencies on the horizon  $T$ , sample size $n$, the sizes of the observation space $|\mathcal{O}|$,  and the sizes of action spaces $|\mathcal{A}|$.  The key technique is the error decomposition shown in Section C.1, where we carefully separate the policy value error into a first-order term and other higher-order terms.  Second, under bounded concentrability conditions, the dependence in the memoryless case can be improved to $\mathcal{O}(\frac{T}{\sqrt{n}}\sqrt{|\mathcal{O}||\mathcal{A}|})$, which matches the optimal rate for OPE in MDPs. **This improvement is achieved through Lemma 1, which first establishes a novel variance decomposition specific to confounded POMDPs.**
>
>
>
> **4. Condition $|\mathcal{O}|>|\mathcal{S}|$ in Assumption 1**
>
>  We apologize for the problematic expression.  $|\mathcal{O}|\ge|\mathcal{S}|$ is the necessary condition. Assumption 1 implies that the observation contains sufficient information about the state.
>
>
>
> **5. Some confusing notations in Theorem 1**
>
> The random variables in capital letters do not take expectation, since the notation $\mathbb{P}(X)$ represents a $n$-length vector with $\mathbb{P}_ {i}(X):=p(X=x_i)$ and $\mathbb{P}(X | Y)$ denotes a $n \times m$ matrix with $\mathbb{P}_ {i, j}(X | Y):=p(X=x_j | Y=y_i)$. The lowercase letters denote realizations of a random variable, such as $X = x_i$. For clarity,  we will write $\mathbb{P}(O_{t+1}, O_t = o_t \mid A_t = a_t, O_0,H_{t-1})$ consistently, rather than $\mathbb{P}(O_{i+1}, o_i \mid a_i, O_0,H_{i-1})$ for simplicity.
>
>
>
>
>
> **ref:**
>
> [1] Doubly Robust Off-policy Value Evaluation for Reinforcement Learning. ICML, 2016.
>
> [2] Computationally Efficient PAC RL in POMDPs with Latent Determinism and Conditional Embeddings. ICML, 2023.
>
> [3] Provably Efficient Reinforcement Learning in Partially Observable Dynamical Systems. NIPS, 2022
>
> [4] Provable reinforcement learning with a short-term memory. ICML, 2022.

---

> > ### Comment · Reviewer_mBzT · 2025-08-04
> >
> > The author’s response partially resolved my concerns, including clarity issues and the exponential dependency on the horizon parameter. However, I am still concerned with the technical significance of the paper, as well as the lacking of the fundamental limit of the problem, i.e., the lower bounds. Given that the paper stands as an initial step towards studying the explicit dependencies of the sample complexity on all problem parameters for tabular confounded POMDPs, I appreciate the main contributions of the paper. Overall, I am willing to raise my score to 4.

---

> > > ### Author Response · Authors · 2025-08-05
> > > **Thank You for Your Review and Feedback**
> > >
> > > I sincerely appreciate the raised score and the constructive feedback that you provided. We will carefully consider these comments in our future work. Thank you once again for your thoughtful review.

---

### Official Review · Reviewer_4S6u · 2025-07-02

**Clarity:** 3
**Significance:** 3
**Originality:** 3
**Rating:** 4
**Confidence:** 2

**Summary:**

This paper investigates the problem of OPE in finite-horizon, tabular POMDPs with unmeasured confounding, focusing on the explicit statistical dependencies on trajectory length and the sizes of the observation and action spaces. The authors address the challenging scenario where the behavior policy depends on unobserved states. The core contribution is a two-step, model-based estimator for policy value, which relies on observable histories as proxies for hidden states and uses invertibility conditions on conditional probability matrices for identification. The paper systematically analyzes the finite-sample error bounds of this estimator, showing that for fully history-dependent policies, the error scales exponentially with the horizon and observation/action space, while for memoryless policies, the bound improves and matches the best-known result for tabular MDPs. Furthermore, under bounded concentrability conditions, the dependence on the horizon can be improved. Simulation studies confirm that empirical trends align with the theoretical analysis.

**Questions:**

1. How reasonable are Assumptions 1 (invertibility) and 2 (coverage) in practical offline RL datasets? Can the authors propose computationally feasible diagnostics or practical criteria for practitioners to check these conditions before applying the method?
2. How sensitive is the estimator to violations of invertibility or coverage? Could the authors numerically or empirically investigate near-rank-deficient matrices or scarce data for certain histories to characterize when estimation breaks down?
3. Can the authors provide empirical comparisons with alternative OPE methods in the tabular regime? How do the variance and robustness properties compare in practice?
4. How do the proposed error bounds and empirical results compare to those from spectral methods or recent future-dependent estimators for confounded POMDPs?
5. How does the computational cost of the two-step estimator scale with horizon and observation/action space sizes?
6. Does the analysis accommodate time-varying or nonstationary behavior policies, or policies depending on longer histories? What is the rationale for the data assumptions made (e.g., as in line 121 differing from prior works), and how does this choice affect practical applicability?
7. The simulation experiments lack sensitivity analysis with respect to key parameters, such as $\epsilon$.
8. In Figure 2(b), performance shows a sharp drop at $n=1000, T=4$. Is this due to randomness from a single trial, or does it indicate a more systematic issue?

**Ethical Concerns:**

["NO or VERY MINOR ethics concerns only"]

**Final Justification:**

This work combines algorithms, theory, and simulation, and I believe it has the potential to be accepted.

**Limitations:**

Not sufficient.

**Quality:**

3

**Strengths And Weaknesses:**

**Strengths**: The paper makes a significant theoretical advancement by providing the first explicit finite-sample error bounds for OPE in confounded POMDPs, clarifying how sample size, horizon length, observation/action space cardinalities, and policy type jointly influence statistical difficulty. The results directly address three open questions regarding horizon dependence, the effect of history-based policies, and the role of concentrability coefficients. For memoryless policies, the bounds match the sharpest known results from tabular MDPs, effectively bridging the gap between POMDP and MDP settings even in the presence of confounding. The analysis transparently distinguishes between history-dependent and memoryless policies, making clear when exponential versus polynomial sample complexity arises. The technical development is mathematically transparent, with well-stated assumptions and constructive derivations, and simulation results empirically confirm the predicted scaling relationships between estimation error, horizon, and sample size.

**Weaknesses**
1. The main results hinge on invertibility and full-rank conditions of certain conditional probability matrices, as well as sufficient coverage/visitation, which may be difficult to satisfy or verify in practice, especially in high-dimensional, sparse, or continuous settings.
2. For history-dependent policies, the sample complexity and estimator scale exponentially with the horizon and observation/action space sizes, making the method impractical for long horizons or large tabular problems. The computational burden of estimating and inverting large matrices may be significant even for $T$.
3. Experiments are restricted to a single, simple synthetic setup. There are no tests of robustness to key assumption violations (e.g., low rank, reduced coverage, mis-specified proxies), and no empirical evaluation on more realistic, high-dimensional, or classic POMDP benchmarks.
4. The empirical results only showcase the proposed estimator; there is no direct comparison to other OPE approaches.
5. Some technical sections, particularly around estimation and recursion in Section 4.2, use terse or overloaded notation that may hinder accessibility for a broader audience. Definitions and notation for distributions are sometimes implicit or scattered.
6. Important details for replication (e.g., code, random seeds, parameter sweeps) are missing.

---

> ### Author Rebuttal · Authors · 2025-07-31
>
> We appreciate the reviewer for their valuable comments! Below, we respond to the weaknesses and questions.
>
> **1. Clarification of assumption:**
>
> The invertibility and coverage conditions are standard and widely used in POMDP literature [1, 2, 3].
>
> - **invertibility:** The invertibility condition is essential for identifying policy value in tabular OPE  for POMDPs [1]. Checking the invertibility of $\mathbb{P}(O_t | a_t, S_t)$ is straightforward when both $S_t$ and $O_t$ are binary. In this case, invertibility is ensured if $\mathbb{P}(O_t = 1 | a_t, S_t = 0) \neq \mathbb{P}(O_t = 1 | a_t, S_t = 1)$ for all $a_t$. For larger $S_t$ and $O_t$, if $|\mathcal{O}|>|\mathcal{S}|$, the conditional probability matrix $\mathbb{P}(O_t | a_t, S_t)$ is more likely to satisfy full row rank,  as it is rare to encounter linearly dependent columns in real-world tasks.  This scenario is reasonable and common in practice. For instance, in an autonomous vehicle scenario, the true state may include position and velocity, while the observation could come from multiple sensors, such as camera images,  GPS data, accelerometer readings, and LIDAR scans. Additionally, $\mathbb{P}(O_t|a_t, O_0,H_{t-1})$  is more likely to have full row rank, as $|\mathcal{O}||\mathcal{H}_{t-1}|\gg|\mathcal{O}|$  over time. This shows the advantage of using the full history as a proxy for the hidden state.
>
> - **coverage:**  The coverage condition is a standard assumption for RL theory. We adopt a **more relaxed condition** that imposes a bounded second moment condition on the cumulative importance ratio in Assumption 2, compared to directly bounding the importance weight used in literature [1,4].
>
> - **violations of assumption**:  The key to avoiding violations of invertibility lies in estimating the empirical probability matrix. Two main factors influence this estimation:  sample size $n$ and Horizon $T$. A longer horizon introduces more action-observation-history combinations that increase zero entries, while a larger sample size helps reduce these zero entries.  In the tabular setting, it is feasible to check the invertibility condition using singular value decomposition (SVD). If the smallest eigenvalue of $\widehat{\mathbf{P}}_{a_t}$ is close to 0 (or the condition number is very large),  the matrix is nearly singular and almost non-invertible, which can be checked using `np.linalg.svd()` in NumPy. The coverage requires a bounded second moment condition on the importance ratio, and a sufficient condition to ensure this is to check the behavior policy $\pi^b(a_t | s_t) > 0$ for all $a_t$ and $s_t$.
> - **check violation:** Due to time constraints, we do not conduct a comprehensive experiment to illustrate these points, but we outline an experimental plan to examine invertibility. By fixing the sample size $n$ and varying the horizon $T$, we will compute the smallest eigenvalue of $\widehat{\mathbf{P}}_{a_t}$ to observe when invertibility is violated, or fix $T$ and vary $n$.  However, the numerical stability (`np.linalg.pinv()`) may degrade with matrices having large or small singular values, and computing the pseudoinverse for large matrices can be expensive. Future work will explore using low-rank approximations to improve computational efficiency.
> - **computational cost scales with $T,|\mathcal{O}|,|\mathcal{A}|$:** The computational cost increases with  $T,|\mathcal{O}|,|\mathcal{A}|$.  As mentioned previously, computing the pseudoinverse using `np.linalg.pinv()` can be computationally expensive with large matrices. For empirical matrix $\widehat{\mathbf{P}}_ {a_t}$, which has dimension $|\mathcal{O}||\mathcal{H}_ {t-1}|\times|\mathcal{O}|$, the size of $\widehat{\mathbf{P}}_{a_t}$ increases as the horizon and the observation space size grow, thereby escalating the computational cost.
>
>
>
> **2 Experiments:**
>
> > a single synthetic setup; check assumption violations; consider more complex settings
>
> Our primary focus is on deriving statistical error bounds for tabular confounded POMDPs, so the experiments are designed to validate the theoretical results in a simple synthetic setup. To evaluate the violation of the coverage assumption, we add the experiments to investigate their impact on error performance under the memoryless policy case ($T=6$). We compare 4 different data generation strategies, where the ratio indicates the proportion of data collected from a random policy. The results show that the error increases as the distribution shift grows, which validates the error bounds.
>
> |        | without mixture policy | mixture(0.2) | mixture (0.5) | mixture(0.6) |
> | ------ | ---------------------- | ------------ | ------------- | ------------ |
> | n=500  | 1.65                   | 2.59         | 3.62          | 4.31         |
> | n=1000 | 1.45                   | 2.05         | 2.93          | 3.65         |
> | n=2000 | 1.21                   | 1.94         | 3.16          | 3.37         |
>
> To extend the evaluation to high-dimensional and more complex settings, function approximation is necessary (such as neural networks, kernel methods). However, since our theoretical results are specifically derived in tabular settings, this is beyond the scope of our main focus, and we leave it as future work.
>
> > empirical comparisons with alternative methods.
>
> Appreciate the suggestion. Since the simulation studies are primarily focused on verifying the proposed error bound and there is no publicly available code for other confounded POMDP methods, we did not include the comparison. But we will include other tabular OPE methods in the future.
>
> > lack sensitivity analysis for key parameters, such as $\epsilon$.
>
>  Thanks for this nice suggestion. We compare 3 settings with $\epsilon=0,0.4, 0.8$ for memoryless case. When $\epsilon=0$, $P_t(O_t|S_t)=\mathbb{1}\\{O_t=S_t\\}$, meaning no confounding exists, and larger $\epsilon$ indicates more noise within the emission probability. The results show that error increases as $\epsilon$ grows, which further illustrates that the less information the observation contains about the latent state, the more challenging learning confounded POMDPs become.
>
> |               | $\epsilon=0$ | $\epsilon=0.4 $ | $\epsilon= 0.8$ |
> | ------------- | ------------ | --------------- | --------------- |
> | $n=1000, T=6$ | 1.41         | 3.41            | 4.60            |
> | $n=2000, T=6$ | 1.38         | 3.36            | 4.59            |
> | $n=1000, T=8$ | 1.73         | 4.34            | 5.01            |
> | $n=2000, T=8$ | 1.69         | 4.16            | 4.88            |
>
> > the sharp drop in Fig 2(b).
>
> This drop is due to the unreliable pseudoinverse solution of empirical matrices in certain repeated runs,  caused by the relatively small sample size leading to sparse empirical matrices. In future work, we may consider better methods to address these practical challenges.
>
>
>
> **3. Other questions:**
>
> > Does the analysis accommodate time-varying behavior policies,...?
>
> In our analysis, the behavior policy, defined as $\pi^{b}_t:\mathcal{S}\to\Delta(\mathcal{A})$, is time-varying, and we consider both fully history-dependent and memoryless policy. Unlike previous work where the behavior policy depends only on the one-step observation, i.e., $\pi^{b}_t:\mathcal{O}\to\Delta(\mathcal{A})$, our setting is more challenging due to unobserved states acting as unmeasured confounders,  affecting actions, rewards, and transitions simultaneously. This is more aligned with real-world scenarios, as in autonomous driving,  where the behavior policy of the vehicle’s control may depend on the unobserved driver’s mental state, with observed variables (e.g., position, speed, traffic conditions) used for OPE.
>
> > compare to other methods
>
> - **compared to spectral methods [4]:**  The key difference is that the paper [4] aims to relax the invertibility condition using spectral method, and establish the identification results for confounded POMDPs. However, their work does not provide finite-sample error bounds. In contrast, our focus is on establishing the finite-sample error bounds of a model-based OPE estimator for confounded POMDPs, and we provide the first explicit statistical dependencies on horizon and the sizes of the observation and action spaces.
>
> - **compared to future-dependent methods [2,3]:** Recent work [2, 3] use the future proxy to infer the hidden state,  implicitly assuming that the future proxy contains sufficient information about hidden states. However, they focus on **unconfounded POMDPs**, which restricts the behavior policy to be memoryless and dependent only on the current observation. In contrast, our work focus on more complex and realistic confounded POMDPs, where the behavior policy depends on the latent state. Our error bounds, scale as  $\mathcal{O}(\frac{T}{\sqrt{n}})$, are sharper than  $\mathcal{O}(\frac{T^2}{\sqrt{n}})$ in [2,3] in terms of horizon $T$, omitting dependencies on the sizes of the observation and action spaces, as work [2, 3] focuses on general function approximation.
>
> > Important details for replication are missing
>
> All simulations are performed using a single NVIDIA GeForce RTX 3090 GPU, and each data point in the figure is average of the results from 5 random seeds. Due to the prohibition on using any links this year, we will release the code if this manuscript is accepted.
>
> > overloaded notation used in Section 4.2
>
> We will remove the overloaded notations $\mathbf{B}^{\pi}$ and $\mathbf{C}_{r,o}$ and clarify the hidden symbols to improve readability.
>
> Ref:
>
> [1] Off-policy evaluation in partially observable environments. AAAI.
>
> [2] Future-dependent value-based off-policy evaluation in pomdps. NIPS
>
> [3] On the curses of future and history in future-dependent value functions for off-policy evaluation. NIPS.
>
> [4] A spectral approach to off-policy evaluation for pomdps. ICML.
>
> [5] A minimax learning approach to off-policy evaluation in confounded partially observable markov decision processes. ICML.

---

> > ### Comment · Reviewer_4S6u · 2025-08-04
> >
> > Thanks for the rebuttal. I will keep my score for potential acceptance.

---

> > > ### Author Response · Authors · 2025-08-05
> > > **Thank You for Your Review and Feedback**
> > >
> > > I sincerely thank you for your positive feedback, and your constructive and thoughtful review.

---

### Decision · Program_Chairs · 2025-09-17

**Decision:**

Accept (poster)

**Comment:**

**(a) Summary of Scientific Claims and Findings**

The paper addresses of off-policy evaluation (OPE) in finite-horizon, tabular POMDPs with unmeasured confounding, where the behavior policy may depend on hidden states. The authors propose a two-step model-based estimator that leverages observable histories as proxies for hidden states under invertibility assumptions on conditional probability matrices. They establish finite-sample error bounds that reveal exponential scaling with horizon and observation/action space size for history-dependent policies, and improved (polynomial) bounds for memoryless policies. With additional concentrability assumptions, the horizon dependence is further reduced, achieving rates comparable to tabular MDPs. Synthetic simulations corroborate the theoretical scaling trends.

**(b) Strengths**

- Novelty: Provides the first explicit finite-sample error bounds for OPE in confounded POMDPs.

- Theoretical contribution: Results characterize the statistical difficulty as a function of horizon, policy type, and problem dimensions.

- Bridging gap to MDPs: Shows that for memoryless policies, the bounds match the stricter known MDP results

- Empirical alignment: Simulation experiments, while limited, validate the theoretical findings

**(c) Weaknesses**

- Strong assumptions: Relies on invertibility and full-rank conditions of conditional probability matrices, as well as broad coverage, which may be unrealistic or unverifiable in practical scenarios

- Limited experiments: Evaluation is restricted to simple synthetic settings, lacks robustness checks on violating assumptions, comparative baselines, or tests on more realistic benchmarks.

- No lower bounds are provided, leaving unclear whether the derived upper bounds are tight

**(d) Reasons for Decision**

After the discussion, the paper remains with some weaknesses, especially related to the technical significance since no lower bounds are provided. On the other side, this represents an initial work on this topic. For this reason, along with the soundness of the paper, I am recommending acceptance.

**(e) Rebuttal and Discussion**

The authors' rebuttal and discussion have clarified some issues. Although, as mentioned above, some issues regarding significance remain open.